# Selective remodelling of the adipose niche in obesity and weight loss

Antonio M. A. Miranda[1,2], Liam McAllan[1,2], Guianfranco Mazzei[1,2], Ivan Andrew[1,2], Iona Davies[3], Meryem Ertugrul[1,2], Julia Kenkre[3,4], Hiromi Kudo[5], Joana Carrelha[6], Bhavik Patel[2,7], Sophie Newton[1,2], Weihua Zhang[8,9], Alice Pollard[1], Amy Cross[10], Oliver McCallion[10], Mikyung Jang[1,2], Ka Lok Choi[1,2], Scarlett Brown[1,2], Yasmin Rasool[1,2], Marco Adamo[11], Mohamed Elkalaawy[11], Andrew Jenkinson[11], Borzoueh Mohammadi[11], Majid Hashemi[11], Robert Goldin[4,5], Laurence Game[1,2], Joanna Hester[10], Fadi Issa[10], Dylan G. Ryan[12], Patricia Ortega[4], Ahmed R. Ahmed[4,13], Rachel L. Batterham[11,14,15], John C. Chambers[4,8,9,16,17], Jaspal S. Kooner[4,9,16,18], Damir Baranasic[1,2,19], Michela Noseda[18], Tricia Tan[3,4] & William R. Scott[1,2,4 ✉]

Weight loss significantly improves metabolic and cardiovascular health in people with obesity[1–3]. The remodelling of adipose tissue (AT) is central to these varied and important clinical effects[4]. However, surprisingly little is known about the underlying mechanisms, presenting a barrier to treatment advances. Here we report a spatially resolved single-nucleus atlas (comprising 171,247 cells from 70 people) investigating the cell types, molecular events and regulatory factors that reshape human AT, and thus metabolic health, in obesity and therapeutic weight loss. We discover selective vulnerability to senescence in metabolic, precursor and vascular cells and reveal that senescence is potently reversed by weight loss. We define gene regulatory mechanisms and tissue signals that may drive a degenerative cycle of senescence, tissue injury and metabolic dysfunction. We find that weight loss reduces adipocyte hypertrophy and biomechanical constraint pathways, activating global metabolic flux and bioenergetic substrate cycles that may mediate systemic improvements in metabolic health. In the immune compartment, we demonstrate that weight loss represses obesity-induced macrophage infiltration but does not completely reverse activation, leaving these cells primed to trigger potential weight regain and worsen metabolic dysfunction. Throughout, we map cells to tissue niches to understand the collective determinants of tissue injury and recovery. Overall, our complementary single-nucleus and spatial datasets offer unprecedented insights into the basis of obese AT dysfunction and its reversal by weight loss and are a key resource for mechanistic and therapeutic exploration.

Obesity affects more than one billion people worldwide[5]. Increased AT mass, which is the defining feature of obesity, is one of the main risk factors for type 2 diabetes, cardiovascular disease, certain cancers and early death[6]. Reduction in AT mass through weight loss (WL) significantly improves obesity-induced comorbidities and can reduce mortality[1–3]. A synergistic and detailed understanding of the biology underpinning these contrasting clinical effects is central to improving treatment options and health outcomes.

ATs have a unique capacity to adapt their structure and functions to maintain metabolic homeostasis as energy demands change[4,7]. In obesity, excess expansion limits this flexibility and induces pathological remodelling changes, notably adipocyte hypertrophy, immune cell infiltration, pro-inflammatory cytokine release, impaired angiogenesis and fibrosis, that contribute to multiorgan inflammation, insulin resistance, metabolic dysfunction and disease[4,7]. However, despite extensive investigation, the molecular triggers, cellular phenotypes

[1]Institute of Clinical Sciences, Faculty of Medicine, Imperial College London, London, UK. [2]MRC Laboratory of Medical Sciences, London, UK. [3]Department of Metabolism, Digestion and Reproduction, Imperial College London, London, UK. [4]Imperial College Healthcare NHS Trust, London, UK. [5]Section for Pathology, Division of Digestive Diseases, Department of Metabolism, Digestion and Reproduction, Faculty of Medicine, Imperial College London, London, UK. [6]Department of Immunology and Inflammation, Imperial College London, London, UK. [7]Department of Biochemistry, Oxford University, Oxford, UK. [8]Department of Epidemiology and Biostatistics, School of Public Health, Imperial College London, London, UK. [9]Department of Cardiology, Ealing Hospital, London North West University Healthcare NHS Trust, Middlesex, UK. [10]Translational Research and Immunology Group, Nuffield Department of Surgical Sciences, University of Oxford, Oxford, UK. [11]UCLH Bariatric Centre for Weight Loss, Weight Management and Metabolic and Endocrine Surgery, University College London Hospitals, London, UK. [12]MRC Mitochondrial Biology Unit, Keith Peter's building, Cambridge Biomedical Campus, University of Cambridge, Cambridge, UK. [13]Department of Surgery and Cancer, Imperial College London, London, UK. [14]Centre for Obesity Research, Rayne Institute, Department of Medicine, University College London, London, UK. [15]National Institute of Health Research, University College London Hospitals Biomedical Research Centre, London, UK. [16]MRC Centre for Environment and Health, School of Public Health, Imperial College London, London, UK. [17]Lee Kong Chian School of Medicine, Nanyang Technological University, Singapore, Singapore. [18]National Heart and Lung Institute, Imperial College London, London, UK. [19]Division of Electronics, Ruđer Bošković Institute, Zagreb, Croatia. ✉e-mail: w.scott@imperial.ac.uk

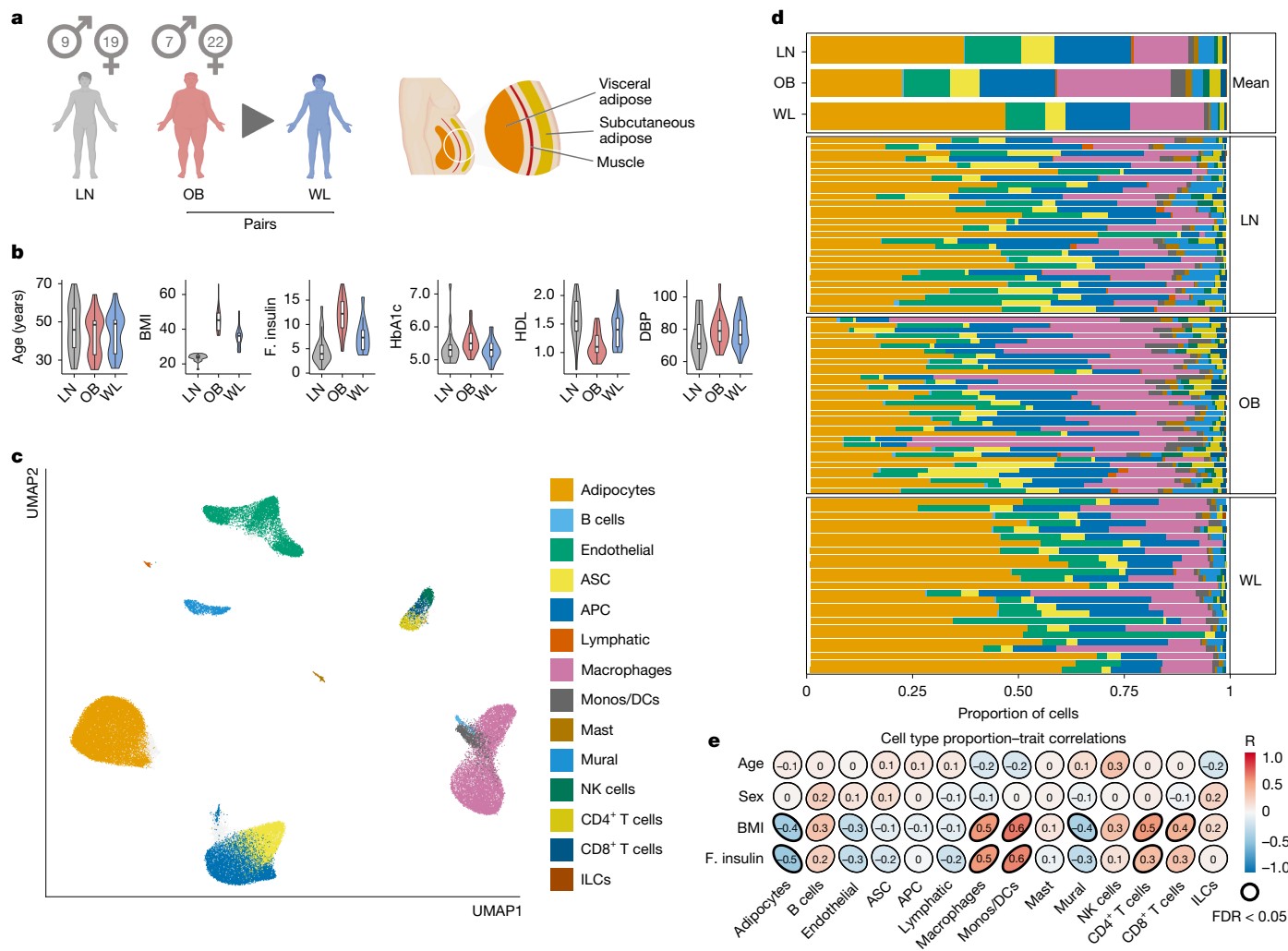

**Fig. 1 | A single-cell atlas of human AT in leanness, obesity and WL.**
**a**, Graphical representation of the primary study cohort (left; single-nucleus analyses in *n* = 25 obese (OB) people before and after WL and *n* = 24 lean (LN) people, with spatial analyses in *n* = 4 people per group) and AT anatomical location (right). **b**, Clinical characteristics of the primary cohort (*n* = 24 LN and 25 paired OB–WL donors). Boxplot, median interquartile range minimum and maximum. BMI, body mass index (kg m⁻²); F insulin, fasting insulin (mIU L⁻¹); HbA1c, haemoglobin A1c (%); HDL, high-density lipoprotein cholesterol (mM); DBP, diastolic blood pressure (mm Hg). **c**, Uniform manifold approximation

and projection (UMAP) of 145,452 human AT cells (*n* = 74 samples of the primary cohort and *n* = 13 samples of the Emont published cohort[11], single nucleus). ASC, adipocyte stem cells; APC, adipocyte progenitor cells; Mono, monocytes; DCs, dendritic cells; ILCs, innate lymphoid cells. **d**, Cell-type proportions (for the cell types in **c**) in the combined cohort, mean per group, and for each sample (single nucleus). **e**, Correlations between cell types and clinical traits (Pearson, LN and OB samples only, single nucleus). Illustration in **a** created using BioRender (Scott, W., https://BioRender.com/rtmnzaj; 2025).

and signalling pathways underlying obese AT dysfunction, particularly in humans, are only partly understood.

Therapeutic WL leads to a reduction in AT mass, systemic inflammation and insulin resistance, as well as subsequent improvements in obesity-related comorbidities[3,4,8]. Although this strongly suggests that WL ameliorates AT dysfunction and its harmful physiological effects, surprisingly little is known about the underlying mechanisms. Indeed, certain AT responses to WL may be maladaptive and predispose to weight regain[9].

Defining the cell types, regulatory mechanisms and signalling pathways responsible for pathological and therapeutic AT remodelling is needed to guide therapy development for the harmful health consequences of obesity.

## Mapping AT remodelling dynamics

To better understand obese AT dysfunction and its reversal after WL, we carried out single-nucleus RNA sequencing of approximately 100,000

cells from men and women with extreme obesity (*n* = 25) before and after WL surgery and from healthy lean controls (*n* = 24; Fig. 1a). WL significantly improved metabolic parameters, although not to the lean baseline (Fig. 1b and Extended Data Table 1). We focused on abdominal subcutaneous AT because of its contribution to central obesity and its adverse metabolic effects relative to other subcutaneous AT depots[10]. This cohort formed the basis for between-group exploratory analyses. Our results were integrated with a further 50,000 cells (nuclei) from the largest published human subcutaneous adipose atlas to improve cell annotation[11] (*n* = 9 obese and *n* = 4 lean samples; Extended Data Fig. 1a–c). Spatial transcriptomics in equivalent cohorts (approximately 25,000 cells, *n* = 4 per group; Fig. 1a, Extended Data Fig. 1b,c and Extended Data Table 1) enabled us to orient and contextualize cell phenotypes within the organizational hierarchy of healthy and dysfunctional AT.

This captured a rich representation of the cellular, structural and functional dynamics of the subcutaneous AT niche in human weight gain and WL. Tissue-wide clustering (Fig. 1c) and compositional

analyses demonstrated extensive immune cell (mainly macrophage but also lymphocyte) infiltration in obese AT (Fig. 1d,e and Extended Data Fig. 1d,e). Obese AT also showed a deficit in mature adipocytes, suggesting increased cell death and/or a failure to replenish mature adipocytes. WL mitigated these typically deleterious effects[4,7].

## Persistent macrophage activation

Immune cell infiltration is a pathognomonic feature of obese AT[7] but the impact of WL on inflammatory remodelling is unclear, with studies indicating opposing anti- and pro-inflammatory effects[9]. We clustered myeloid cells (n = 34,280; Fig. 2a, Extended Data Fig. 2a and Supplementary Table 2) into heterogeneous subclasses of AT macrophages, monocytes and dendritic cells (MYE1–10)[12].

The increase in AT macrophages (mean from 14% to 31%) primarily comprised lipid-associated macrophages (LAMs; mature MYE2 and immature MYE3) expressing lysosomal, lipid metabolism and metabolic activation markers (*CD9*, *TREM2*, *LPL* and *LIPA*; Fig. 2a,b and Extended Data Fig. 2a–c). Classical monocytes (MYE5) expressing *VCAN* also increased, indicating constitutive trafficking from blood. Visualization and marker gene patterns supported a differentiation continuum from monocytes, to immature and then mature LAMs (Fig. 2a and Extended Data Fig. 2a). Proportional analyses revealed lower fractions of tissue-resident macrophages (MYE1 and TRMs) expressing homeostatic markers (*LYVE1*, *FOLR2* and *MRC1*; Extended Data Fig. 2c). Neighbourhood graphs confirmed that this represented a relative (not absolute) TRM reduction (Extended Data Fig. 2b). Proliferative macrophages expressed *MCP-1* (*CCL2*), TRM and LAM markers, supporting low-level *MCP-1*-dependent expansion of both populations in human obesity[13] (Extended Data Fig. 2a).

Independent of adiposity, LAM abundance increased with metabolic dysfunction (Extended Data Fig. 2b). This led us to hypothesize that LAMs might have pleiotropic adaptive and maladaptive features. LAM subclustering revealed two main subpopulations that separated on lysosomal or metabolic (LAM ST1, adaptive) and inflammatory (LAM ST2, maladaptive; MHC II, *NLRP3*) signatures (Fig. 2b). Inflammatory LAMs expressed higher *TLR2* and *TREM1* (Fig. 2b, Extended Data Fig. 2d and Supplementary Table 5), cooperative receptors that initiate and amplify inflammation in the pathogen-recognition response[14,15]. In keeping with a deleterious role, inflammatory LAM numbers increased in obesity in association with metabolic dysfunction (Fig. 2c). Spatial and protein analyses indicated context-dependent orientations and functions, with adaptive LAMs aggregating at crown-like structures (CLS; around transcriptionally devoid adipocytes) and inflammatory LAMs being more abundant in isolation or pairs (Fig. 2d and Extended Data Fig. 2e–g).

To provide an unbiased understanding of macrophage metabolic reprogramming, we used gene expression to model metabolic flux systematically. This revealed a global activation state exclusive to obese macrophages, encompassing known[16] and previously unrecognized metabolic changes (1,495 of 1,895 reactions, binomial test, $P = 3.1 \times 10^{-148}$; Fig. 2e, Extended Data Figs. 1f and 3a,b and Supplementary Table 6). Specifically, we found a shift to a high-glycolysis (pro-inflammatory), high-respiratory (anti-inflammatory) profile consistent with extracellular flux analyses in obese mice[16]; corresponding changes in the pentose phosphate pathway and TCA cycle; pervasive activation of cholesterol, lipid and fatty acid synthesis, and oxidation pathways; obligatory upregulation of cellular transport (Fig. 2e,f and Extended Data Fig. 3a). Taking fatty acids as an example, flux modelling uncovered significant activation of fatty acid desaturation (*FADS1* and *SCD*) and mitochondrial β-oxidation (Fig. 2e,f), consistent with buffering and utilization of potentially toxic microenvironmental fatty acids for energy. Global bioactivation was greatest in, but was not limited to, LAMs (Extended Data Fig. 2h), establishing that diverse myeloid classes undergo extensive metabolic priming in obese AT. Experimental energetic profiling confirmed the higher basal activity and glycolytic

capacity of LAMs over TRMs, substantiating our transcriptome-based flux results (Fig. 2g and Extended Data Fig. 2i).

WL led to marked reductions in myeloid cell numbers (mean from 31% to 18%) across subclasses (Extended Data Fig. 2b). Proportional and density analyses showed that myeloid-cell fractions did not differ between obesity and WL (Fig. 2a and Extended Data Fig. 2c), and we verified this in situ (Extended Data Fig. 2j). WL did, however, shift LAMs towards less inflammatory subtypes (Fig. 2c). Overall, this indicated that obesity-induced myeloid cell states persist despite extensive WL. Transcriptomic flux analyses confirmed that global metabolic activation did not fully reverse with WL (Fig. 2e and Extended Data Fig. 3b). But WL did significantly reduce some aspects of fatty acid synthesis and oxidation (mainly desaturases and acyl-CoA synthetases; Fig. 2e,f), temporally linking these pathways to microenvironmental lipid availability. By contrast, glycolysis, respiratory capacity and pentose phosphate pathway flux increased (Fig. 2e,f), implying a need to requisition energy from other sources as fatty acid levels diminish. Differential expression analyses demonstrated widespread reductions in inflammasome, proinflammatory cytokine and chemotaxis genes (Fig. 2f, Extended Data Fig. 2k and Supplementary Tables 7 and 8). Network analyses implicated specific transcription factors (TFs) in TRM and LAM specification and revealed patterns reinforcing the finding that WL improves inflammatory and homeostatic networks, but not LAM transcriptional reprogramming (Fig. 2h and Supplementary Tables 9 and 10). Together, these results demonstrate a complex activation response in obese AT dominated by monocyte recruitment and persistent metabolic reprogramming.

## Reduced lymphocyte infiltration

Low overall numbers (6,222 cells (4%); Extended Data Fig. 2l–n) meant that we were unable to evaluate lymphoid subclass-level variations. Nevertheless, obese AT had higher proportions of CD4+ and CD8+ T cells, NK cells and B cells, remodelling effects ameliorated by WL (Extended Data Fig. 2n). WL also downregulated the lymphocyte activation and cytotoxicity genes (*ETS1* and *SYTL3* (refs. 17,18); Supplementary Tables 7 and 8), further supporting decreased inflammation.

## Enhanced adipocyte metabolic flexibility

Mature adipocytes undergo profound phenotypic changes in obesity and WL, expanding and shrinking to fit evolving energy needs[7]. How this affects their molecular characteristics and diverse metabolic functions is largely unclear. Subclustering revealed 8 mature adipocyte subpopulations (AD1–AD8, n = 44,583 cells; Fig. 3a, Extended Data Fig. 4a,b and Supplementary Table 2). Two subtypes exhibited 'stressed' (AD3, *JUN/NFKBIZ*-hi) and 'fibrotic' (AD6, *NOX4/LOX*-hi) profiles. Stressed and fibrotic adipocytes increased with obesity, indicating that there is selective vulnerability and pathogenicity to the tissue microenvironment (Fig. 3b). Another subpopulation with a lipid biosynthetic profile (AD5, *PNPLA3/MOGAT1*-hi) unexpectedly decreased in obesity (Fig. 3b). WL led to a marked reduction in stressed adipocytes (mean from 55% to 14%), a shift towards lower fibrotic numbers, and relative increases in lipid biosynthetic cells (Fig. 3b). Compositional changes in stressed and lipid biosynthetic populations were verified in situ (Extended Data Fig. 4c). Beige adipocytes were rare (AD8 *GATM*-hi, 1%) and invariant between conditions.

Expression-based metabolic flux analyses detected significant defects in fatty acids and branched-chain amino acid (BCAA) breakdown in obese compared with lean adipocytes, mirroring previous results[19,20] and together suggesting that metabolic flexibility was impaired (Fig. 3c and Extended Data Fig. 3a). By contrast, WL led to a marked global increase in adipocyte metabolic flux (1,485 of 1,895 reactions; binomial test, $P = 1.4 \times 10^{-142}$; Fig. 3c, Extended Data Figs. 1f and 3b and Supplementary Table 6) probably reflecting a negative energy balance.

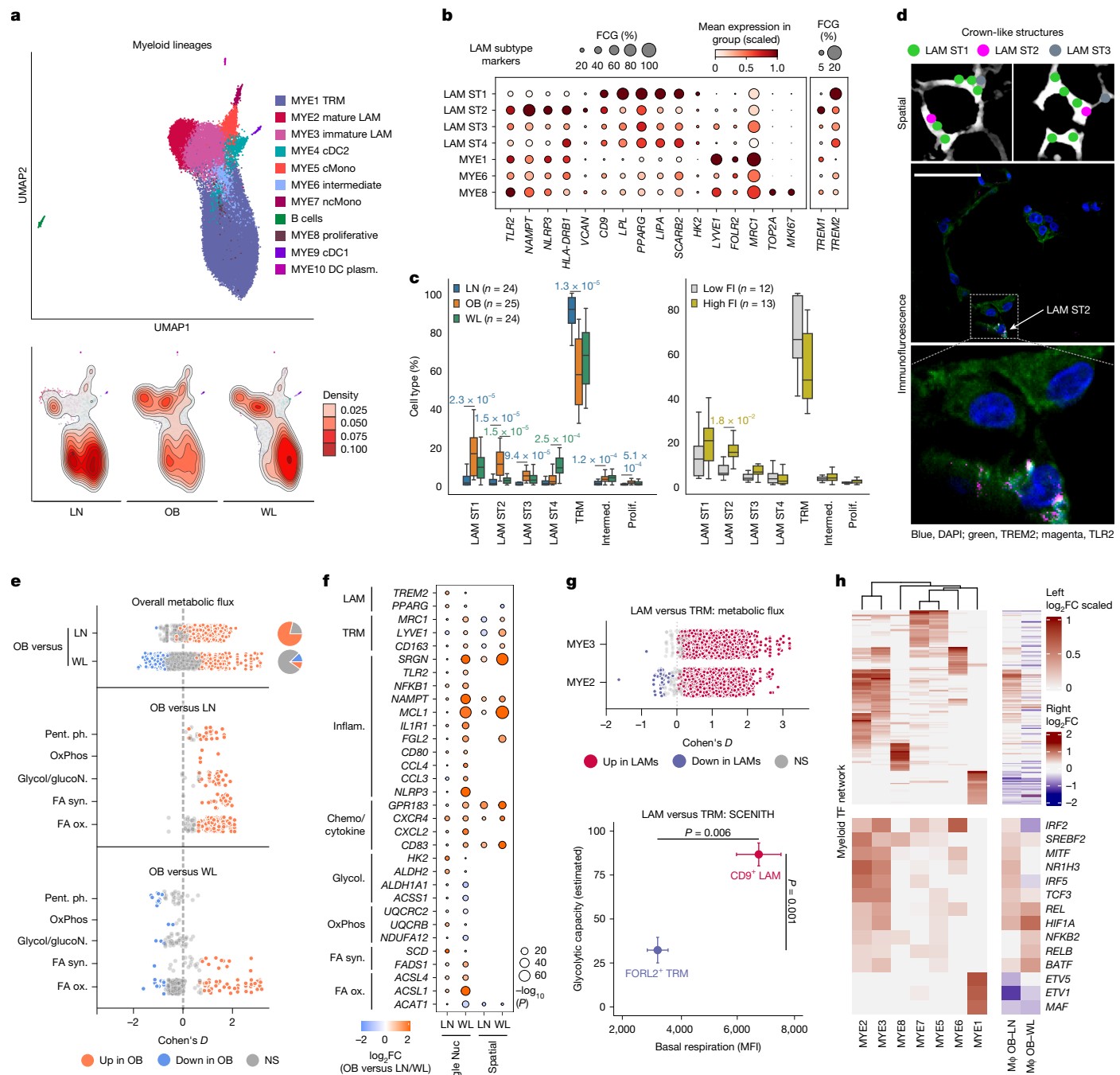

**Fig. 2 | Immune cell infiltration, activation and reprogramming in obesity and WL. a**, UMAP embedding of myeloid (MYE) cell classes (top) and densities (bottom). cDC1 and cDC2, dendritic cells 1 and 2; cMono and ncMono, classical and non-classical monocytes; plasm., plasmacytoid. **b**, LAM subtype (ST) marker genes relative to the main macrophage classes. FCG, fraction of cells in the group. **c**, LAM subtype proportions in LN, OB and WL (left) and OB split into low and high fasting insulin (FI, right), relative to total macrophages/sample. Boxplot, median IQR minimum and maximum; *n*, number of donors. Wilcoxon paired (OB–WL) and unpaired (OB–LN, FI) two-tailed, FDR adjusted *P*-values. Intermed., intermediate; Prolif., proliferative. **d**, CellTypist predicted LAM subtypes in spatial datasets at CLS (top). Immunohistochemistry of TREM2 (pan-LAM marker) and TLR2 (ST2 marker) at CLS (middle, bottom). Scale bar, 50 µm. **e**, Transcriptomic flux-based analyses showing global (top) and pathway-specific (middle and bottom) metabolic activation in OB compared with LN and WL macrophages. Cohen's *D*, coloured at FDR < 0.05 (Wilcoxon): red, obese high; blue, obese low; grey, non-significant. Pie charts show the proportions of significant reactions (*n* = 24 LN; *n* = 25 OB; *n* = 24 WL donors). Pent. ph, pentose

phosphate pathway; OxPhos, oxidative phosphorylation; Glycol/glucoN, glycolysis/glucogenesis; FA syn., fatty acid synthesis; FA ox., fatty acid oxidation. **f**, Differentially expressed genes in macrophages in LN–OB and OB–WL comparisons, separated by datasets. Coloured by log₂-transformed fold change (log₂FC): red, obese high; blue, obese low; sized by adjusted −log₁₀*P*-value; negative binomial mixed-effects model. Circled dots represent comparisons with absolute log₂FC > 0.3 and adjusted *P* < 0.05. **g**, Transcriptomic flux-based analyses (top) showing global metabolic activation in LAMs compared with TRMs. Cohen's *D*, coloured at FDR < 0.05 (Wilcoxon); red, LAM high; blue, LAM low; grey, non-significant (*n* = 86 MYE1, *n* = 74 MYE2 and *n* = 80 MYE3 samples). SCENITH (bottom) basal respiration (HPG incorporation) and glycolytic capacity (change in HPG incorporation) in LAMs and TRMs from OB donors (*n* = 7, mean ± s.e.m., paired Student's *t*-test). MFI, mean fluorescence intensity. **h**, Differential gene regulatory networks in: left, macrophage subtypes (scaled log₂FC > 0.3, subtype versus all other subtypes, Wilcoxon, FDR < 0.05); and right, all macrophages (Mφ) in LN–OB and OB–WL comparisons (log₂FC, Wilcoxon, red, OB high).

Unexpected anabolic activity led us to investigate whether triglyceride mobilization, which is a physiological response to caloric restriction, might initiate lipid cycling (repetitive degradation and resynthesis). To verify flux models, we compared enzymatic activity scores and pathway-limiting enzymes in important substrate pathways across groups[21] (Fig. 3d,e and Extended Data Fig. 4d–f). Obese adipocytes had consistently lower metabolic activities (scores and enzymes), again indicating metabolic inflexibility. WL systematically increased opposing lipid biosynthesis and breakdown pathways (Fig. 3d). Consistent with this, we found significant changes in canonical enzymes in sequential cycle steps (Fig. 3e), including *DGAT2*, which encodes an acyltransferase that catalyses triglyceride synthesis and mediates lipid cycling in vitro[22]. Because enzymatic expression is a crucial determinant of catalytic competence, this indicates that WL may initiate triglyceride cycling, a highly bioenergetic process with important lipid-diversifying, toxic fatty acid-quenching metabolic benefits[22]. WL also reversed defects in BCAA catabolism (pathway flux and canonical enzymes; Fig. 3c–e and Extended Data Fig. 4f), the predicted consequences of which are systemic BCAA clearance and improved insulin sensitivity[23]. Lipid cycling was a feature of *PNPLA3*-hi adipocytes (AD5), whereas stressed (AD3) adipocytes were characterized by lower metabolic turnover (Fig. 3f). These typical catabolic and previously unrecognized anabolic effects of WL suggest that substrate mobilization engages cell-autonomous cycling pathways that may underlie widespread improvements in metabolic homeostasis.

To see which TFs were explicitly responsible for WL-induced metabolic activation, we carried out network analyses limited to metabolic pathway genes (Extended Data Fig. 5g and Supplementary Table 11). *MLXILP* and *SREBF1* ranked highly in triglyceride synthesis, validating our approach and implicating them in control of WL-induced lipid cycling. Other notable findings were TFs linked to redox biology and BCAA catabolism. Many of the leading TFs (38 of 53, $P < 0.05$ Bonferroni adjusted, more than 50 metabolic target genes) overlapped human metabolic disease genome-wide association study (GWAS) loci[24] (Extended Data Fig. 4g), causally implicating specific TFs and the respective metabolic pathways in pathophysiology and treatment response.

Differential expression analyses identified altered biomechanics as a potential driver of adipocyte stress and metabolic dysfunction that was mitigated by WL. Specifically, obesity increased and WL decreased expression of key cytoskeletal tension, mechanotransduction, extracellular matrix (ECM) formation and fibrosis genes (*ACTA2*, *LOX*, *LOXL2* and *VGLL3*)[25,26], effects we verified in unbiased pathway analyses and in situ (Fig. 3g, Extended Data Fig. 4h and Supplementary Tables 7 and 8). Biomechanical genes were enriched in stressed and fibrotic AD3 and AD6 cells (Fig. 3g). We therefore evaluated whether adipocyte hypertrophy and mechanical strain might initiate these maladaptive changes, and whether adipocyte shrinkage during WL might reverse them. As expected, adipocyte sizes increased in obesity and reduced with WL (Fig. 3h and Extended Data Fig. 4i). Despite intrasample heterogeneity, adipocyte size correlated positively with mechanosensitive, stressed and fibrotic gene expression and negatively with homeostatic genes (Fig. 3g, exemplified by the stress marker *JUN* in Fig. 3h). The levels of correlation indicated that this may be one of several factors eliciting tissue stress and fibrosis, or perhaps it is a driver event in a degenerative cycle.

## Reversal of multicellular stress

Adipocyte progenitor cells (APCs) regenerate mature adipocytes and maintain tissue stroma, crucial homeostatic functions that may become impaired in obesity[7]. APCs clustered into: 'multipotent' *DPP4-CD55*-hi progenitors (ASC/APC1); 'committed' preadipocytes (APC2 and APC3) expressing canonical differentiation genes; adipogenesis-regulatory cells (APC4, *KCNIP*-hi and *CD142/F3*-hi); and profibrotic precursors (APC5, *ADAM12*-hi and *POSTN*-hi) (Extended Data Fig. 5a–c and

Supplementary Table 2). APC3 exhibited a stressed profile similar to that observed in mature adipocytes, as well as higher expression of *NOCT* (Fig. 4a and Extended Data Fig. 5c), a potentially restrictive gatekeeper to preadipocyte commitment[27]. In support of this, APC2 selectively expressed late-stage adipocyte maturation genes within a localized subregion (Extended Data Fig. 5c). Stressed and profibrotic cell numbers again reflected adiposity and reduced significantly with WL (Fig. 4b and Extended Data Fig. 6a,b). Both populations had higher expression of hypoxia-inducible factor 1A (*HIF1A*; Fig. 4a and Extended Data Fig. 6a,c), which promotes fibrosis and suppresses adipogenesis in mice (through PPARG phosphorylation)[28]. Correspondingly, WL downregulated hypoxia, profibrotic (TGFβ) and anti-adipogenic (WNT) genes (Extended Data Fig. 6c and Supplementary Tables 7 and 8). Thus, WL may attenuate hypoxia-induced impairment of differentiation competency and profibrotic signalling in certain human APC subpopulations.

Coordinated growth of the vascular network is essential for healthy AT expansion. Vascular cell subclustering recapitulated the endothelial (arterial, capillary and venous) and mural (smooth muscle and pericyte) zonations observed in other tissue types (Extended Data Fig. 5d–g and Supplementary Table 2). As with mature adipocytes and APCs, capillary endothelia and mural cells each showed 'basal' and 'stressed' profiles (Fig. 4a), which changed reciprocally with adiposity (Extended Data Fig. 6a,b). Stressed endothelia overexpressed *APOLD1* and *SNAI1* (Fig. 4a and Supplementary Table 4), highlighting potential pathological neovascularization and endothelial-to-mesenchymal transition[29,30]. Stressed mural cells enriched for *ADAMTS1* (Fig. 4a), an anti-angiogenic protein linked to pericyte detachment, fibrotic transition and capillary rarefaction[31]. In distinct single-nucleus and spatial datasets, WL markedly reduced stressed vascular cell content and markers (Fig. 4c and Extended Data Fig. 6a), implying the reversal of this pathological transformation.

All stressed cell states upregulated a common gene signature (188 genes; Extended Data Fig. 6d,e and Supplementary Table 12). Multicellular stress, although highest in obesity, was a feature of lean tissues, where it increased with age and metabolic dysfunction (Fig. 4c and Extended Data Fig. 6f,g). Gene and pathway analysis revealed putative mediators of multicellular stress (hypoxia, mechanical and oxidative stress, Gp130-mediated cytokines, DNA damage and cell cycle arrest; Extended Data Fig. 7a,b). In vitro induction of DNA damage (using Etoposide) recapitulated the in vivo effects on stress marker proteins and impaired ASPC differentiation capacity (Extended Data Fig. 7c–e). WL led to a marked reduction in multicellular stress genes (Extended Data Fig. 6c), overall emphasizing the importance of multicellular stress pathways in tissue injury and repair.

## Altered tissue niches and cell crosstalk

We used our spatial datasets to investigate the orientation and impact of stressed cells in tissues. To define the cells most associated with stress signals, we quantified the cellular composition of low- and high-stress regions (50-μm bins; Fig. 4d and Extended Data Fig. 8a,b). Stressed cell states were generally enriched in high-stress zonations, apart from stressed capillaries (EC2), which were spread throughout the tissue. We also found a strong association between regions of stress and immune cells, except TRM and NK cells, and an unexpected connection to arterial ECs (EC4; Fig. 4d).

Although this localized individual cell states to stressed zonations, it did not address the non-random grouping of cells in microenvironmental compartments. To evaluate this, we used spatially resolved proximity enrichment (within 300 μm, to capture adipocytes) to search for tissue niches based on cell state neighbourhoods. This identified five distinct cellular communities, termed arterial, venous, adipocyte, stem and stress niches (Fig. 4d,e and Extended Data Fig. 8c). No cell type was niche exclusive, indicating that these patterns reflect tissue gradations. Stem niches were enriched for multipotent ASC/APC1 and

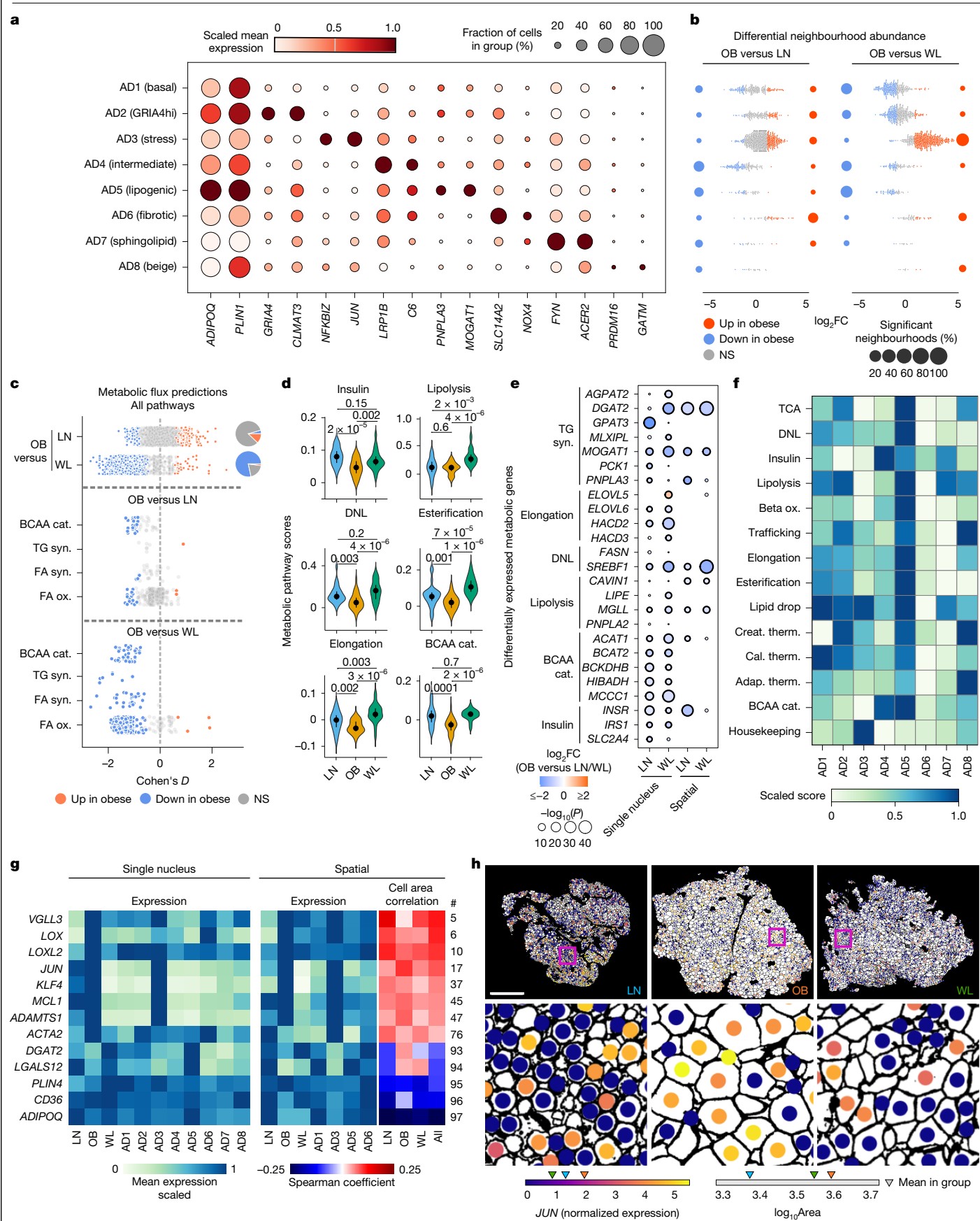

**Fig. 3 | See next page for caption.**

**Fig. 3 | Dynamic regulation of adipocyte cellular and molecular profiles in obesity and WL. a**, Marker-gene expression in mature adipocyte subpopulations. **b**, Beeswarm plots showing changes in neighbourhood abundance in LN–OB and OB–WL comparisons in adipocyte subpopulations. Log$_2$FC, coloured by spatial FDR < 0.1: red, OB high; blue, OB low. The circles show the percentage of significant neighbourhoods. **c**, Transcriptomic flux-based analyses of global (top) and example (middle and bottom) metabolic pathways in OB compared with LN and WL adipocytes. Reaction level, Cohen's *D*, coloured by FDR < 0.05 (Wilcoxon): red, OB high; blue, OB low; grey, non-significant; cat., catabolism; syn., synthesis; ox., oxidation. Pie charts show the proportion of significant reactions. **d**, Scores measuring overall activity in major metabolic pathways in individual adipocytes, averaged by participant (density, median IQR), then compared between conditions. DNL, de novo lipogenesis. Two-tailed Wilcoxon test unpaired (LN–OB and LN–WL) and paired (OB–WL) FDR-adjusted *P*-values are shown (*n* = 24 LN; *n* = 25 paired OB–WL donors). **e**, Differential expression of enzymatic genes in lipid and BCAA

metabolism pathways in OB compared with LN and WL adipocytes, separated by datasets. Coloured by log$_2$FC: red, OB high; blue, OB low; sized by adjusted −log$_{10}$*P*-value, negative binomial mixed-effects model. Circles represent comparisons with absolute log$_2$FC > 0.3 and adjusted *P* < 0.05. **f**, Overall activity in metabolic pathways in adipocyte subpopulations (scaled mean scores). Therm., thermogenesis; Creat., creatine; Cal., calcium; Adap., adaptive. **g**, Mean expression of mechanosensitive, stress, fibrotic and homeostatic genes across conditions and adipocyte subpopulations, in single nucleus (left) and spatial (middle) datasets (limited to genes in both datasets, nucleus segmentation). Spearman correlation (right) of genes with adipocyte areas in each condition and across all conditions combined (spatial dataset, boundary segmentation). The # denotes rank (high-to-low) across 97 genes (*P*-value threshold less than 1 × 10$^{-5}$ in more than one correlation). **h**, Representative spatial sections showing altered adipocyte sizes (WGA segmented) and *JUN* (stress marker) expression across conditions. Bottom bars, mean *JUN* expression and mean log$_{10}$ area in adipocytes across all spatial samples for each condition. Scale bar, 1 mm.

homeostatic TRMs. Stress niches were enriched for AD3, APC3, LAMs, other innate (cMono and cDC2) and adaptive (T cells) immune cells, implicating these states in stress induction and/or response. Arterial endothelial cells formed their own niche, associating with stressed precursors (APC3) and stressed mural cells (Mu4). Direct cell–cell colocalization uncovered immune cell proximity to large venous vessels and LAMs (Extended Data Fig. 8d), potentially reflecting extravasation and transmigration to CLS.

The identification of tissue zonations enabled us to investigate intra- and inter-niche signalling patterns. Ligand–receptor inference analyses in the spatial dataset revealed a complex network of communications. Adipokines and neurotrophic factors were enriched in the adipocyte niche (*ADIPOQ*, *LEP* and *NRXN3*; Extended Data Fig. 8e,f). Canonical WNT and ECM components (FN1, collagens and laminins) were prominent components of the stem niche (Extended Data Fig. 8e). The stress and arterial niches were enriched for proinflammatory chemo-cytokines (*CXCL2*, *CCL2* and *IL6*) and presumptive stress cues (*TGFB1*, *AREG*, *NAMPT* and *THBS1*), several of which overlapped (Fig. 4f and Extended Data Fig. 8e,f). Parallel intercellular communication analyses in the larger single-nucleus dataset linked diverse niche signals to source and target cells, as well as disease pathobiology (Fig. 4g,h and Extended Data Fig. 8g,h). For example, *THBS1* (stressed AD3), *ADGRE5* (pan-immune) and *NAMPT* (multicellular), which are emergent triggers of insulin resistance[32], immune glycolytic metabolism[33] and inflammation[34], were all amplified in obesity and reversed by WL (Fig. 4h and Extended Data Fig. 8h). This showed that stressed niches have a high concentration of signals implicated in pathological and restorative tissue remodelling.

## Repression of senescence

Differential expression analyses to define AT remodelling pathways established that WL broadly reverses the effects of obesity on gene regulation (Extended Data Fig. 9a,b). Many of the strongest transcriptional changes associated with WL were conserved across cell types (Extended Data Fig. 9c,d), indicating that these genes and their underlying pathways might represent important WL mechanisms. Genes altered by WL in multiple cell types (three or more cell types; FC > 0.5, *P* < 0.05 Bonferroni corrected) showed systematic downregulation (213 of 333 genes; binomial test, *P* = 3.9 × 10$^{-7}$). Downregulated genes were grouped into hallmark pathways of AT dysfunction: inflammation (*TNFA* and *IFNG*); hypoxia; fibrosis; immune cell recruitment and activation; and oxidative stress[4,7,26] (Extended Data Fig. 9e). WL also led to downregulation of cell cycle arrest genes (Extended Data Fig. 9e), together indicating that the reversal of cellular senescence might underlie the beneficial effects of WL on inflammation and metabolism.

To examine this, we tested and confirmed the repression of diverse senescent signatures (Fig. 5a and Extended Data Fig. 10a). In multiple cell types, WL led to the downregulation of *CDKN1A* (p21), which is one

of the main cell cycle inhibitors in senescence, and the upregulation of cell cycle progression genes repressed by p21 (ref. 35). Correspondingly, WL markedly decreased the expression of principal senescence markers and unbiased senescence scores (Fig. 5a and Extended Data Fig. 10a–d). We found that p21-positive cells, which had transcriptional characteristics of senescent cells (Extended Data Fig. 10e), were most prevalent among stressed adipocyte, precursor and vascular cell states (Extended Data Fig. 10f), indicating that the shared stress profile reflects vulnerability and transition to senescence. Lean AT also contained substantial (albeit significantly lower) numbers of p21-positive cells (Fig. 5b). By contrast, WL almost completely eliminated p21-positive cells from the tissue (Fig. 5b), a finding that we verified in situ using spatial transcriptomics (Extended Data Fig. 10g) and protein quantification (Fig. 5c). The repression of senescence mirrored enhanced adipocyte bioenergetics, indicating that these effects may be mechanistically coupled (Extended Data Fig. 10h). We therefore established that human WL has previously undescribed potent senolytic effects.

Tissue-wide gene regulatory network analyses revealed a tightly conserved transcriptional nexus in stressed, senescent cells that increased in obesity and decreased in WL (Fig. 5d). The identified TFs grouped into several classes (Fig. 5e and Extended Data Fig. 10i): the AP-1 superfamily, which primes the senescence genome; hallmark signal-dependent TFs that activate inflammation and the senescence-associated secretory phenotype (SASP); Krüppel-Like TFs, which are implicated in cell cycle arrest; TFs that control ciliogenesis (*RFX2/RFX3*), which is a putative senescence regulator; orphan nuclear receptor TFs that are induced by DNA damage and oxidative stress, key senescence triggers; and multiple candidate TFs not previously linked to senescence[36–40]. Individual TFs exhibited autoregulatory effects and shared multiple target genes (Fig. 5e and Extended Data Fig. 10i), including *CDKN1A*, indicating that these TFs may cooperate to potentiate a degenerative cycle of cell stress, senescence, SASP release, inflammation and tissue injury. This transcriptional cascade is turned off by WL.

Because of its importance in reinforcing senescence, we sought to further define the signatures of the AT SASP by systematically comparing the expression of secretory proteins[41] across stressed (high senescence) and basal cell states. This revealed changes in diverse mediators of senescence, tissue injury and metabolic dysfunction, including signalling peptides implicated in multicellular stress and intra- and inter-niche communication (AREG, ADAMTS1, OSMR, IL6ST and CXCL2; refs. 37,42) (Extended Data Fig. 10j and Supplementary Table 12). Presumptive SASP components systematically replicated in situ and localized to stressed and arterial niches (Fig. 5f and Extended Data Fig. 10k,l). Senescent cells strongly upregulated NAMPT, an intracellular driver of the SASP (through enzymatic activity in the NAD salvage pathway) and an extracellular adipocytokine (visfatin) with pleiotropic, context-dependent, predominantly pro-inflammatory effects[34,37] (Fig. 5a and Extended Data Fig. 10j). *NAMPT* expression was

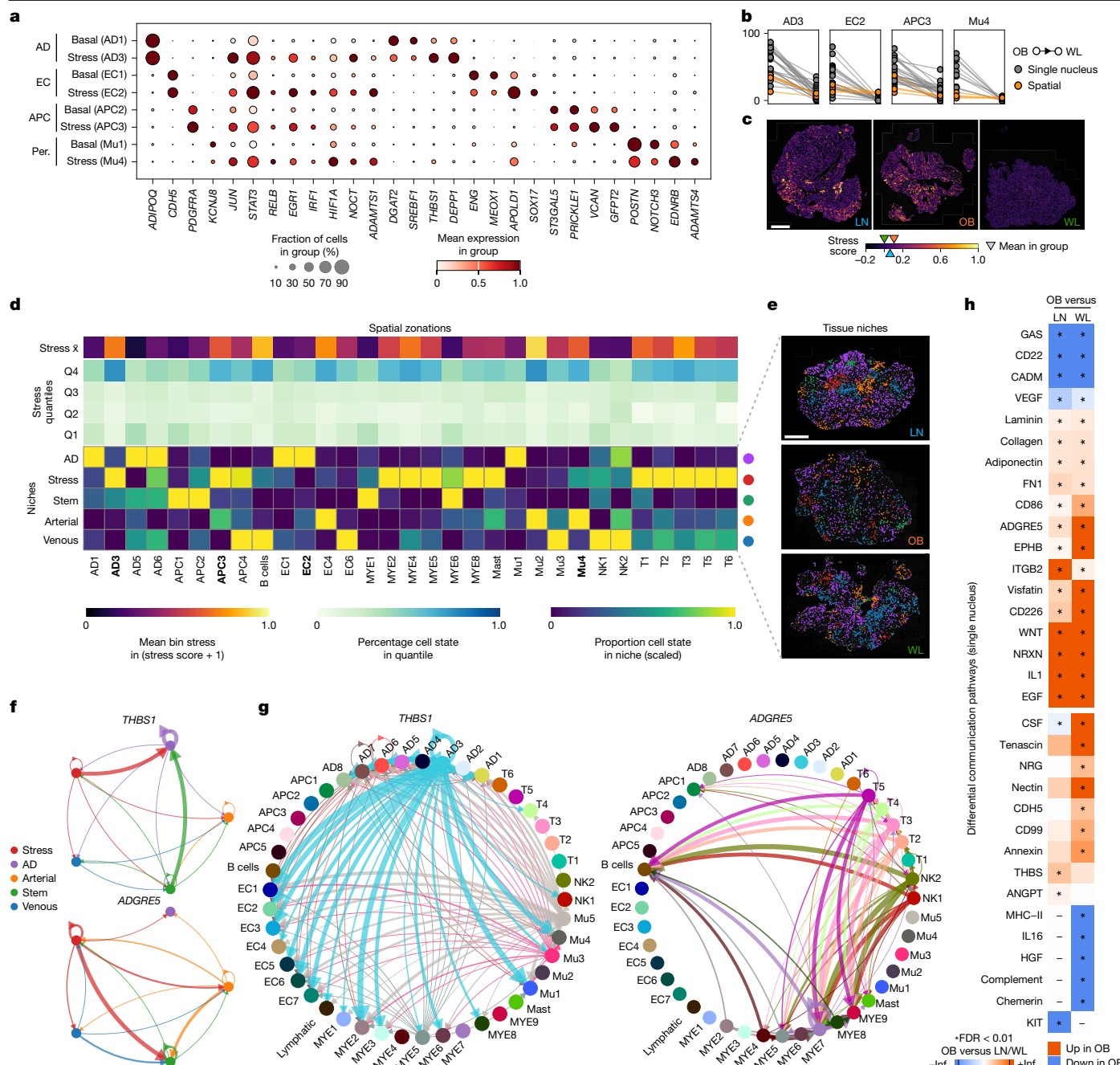

**Fig. 4 | Stressed cells form a spatial niche and enrich for stress-associated signalling pathways. a**, Marker-gene expression profiles in basal and stressed subpopulations of mature adipocytes (AD), precursors (APC), endothelial cells (EC) and mural pericytes (Per.). **b**, Pairwise changes in stressed cell proportions in OB and subsequent WL in single nucleus (grey) and spatial (orange) datasets. **c**, Tissue-wide stress scores (calculated from the 24 common upregulated stress genes present in the spatial dataset, logged score) in representative LN, OB and WL spatial tissue sections, and the mean stress score for each condition in all samples. **d**, Spatial zonations. Top, mean cell state stress score in 50-μm bins. Middle, percentage of cells in stress quantiles, across all conditions, per cell state (Q1 low, Q4 high stress). Bottom, cell state composition of tissue niches, represented as scaled percentage per cell state. Stressed states are shown in bold. **e**, Spatial niches in representative tissue sections. **f**, Imputed CellChat communication between spatial niches for *THBS1* (top) and *ADGRE5*

(bottom). Links represent the scaled mean probability (line thickness) and directions of connectivity. Line colour reflects signal source. All conditions were combined to identify the main niches underlying the pathway effects. **g**, CellChat communication between cell states for *THBS1* (left) and *ADGRE5* (right) in the single-nucleus dataset, across all conditions. Links represent the scaled mean probability (line thickness) and directions of connectivity. Line colour reflects signal source. Lower probability interactions for *ADGRE5* were removed for clarity. **h**, Ligand–receptor pathways with significant differential interactions in OB–LN and OB–WL comparisons (tissue-wide, single-nucleus dataset). Separated into reciprocal (significant in both comparisons, top) and other (significant in one comparison, bottom). Coloured by relative flow: red, OB high; blue, OB low; *FDR < 0.05. Infinity (Inf) represents pathways that were present in only one of the conditions. Dashes indicate null ligand–receptor interactions. Scale bars, 1 mm.

similarly enriched in obese macrophages and inflammatory LAMs (Fig. 2b,f), in keeping with its roles in inflammasome activation and immune recruitment. Tissue-level protein analyses confirmed that

NAMPT abundance increased in obesity and reduced markedly with WL (Extended Data Fig. 10b), together highlighting that NAMPT is a likely driver of AT SASP.

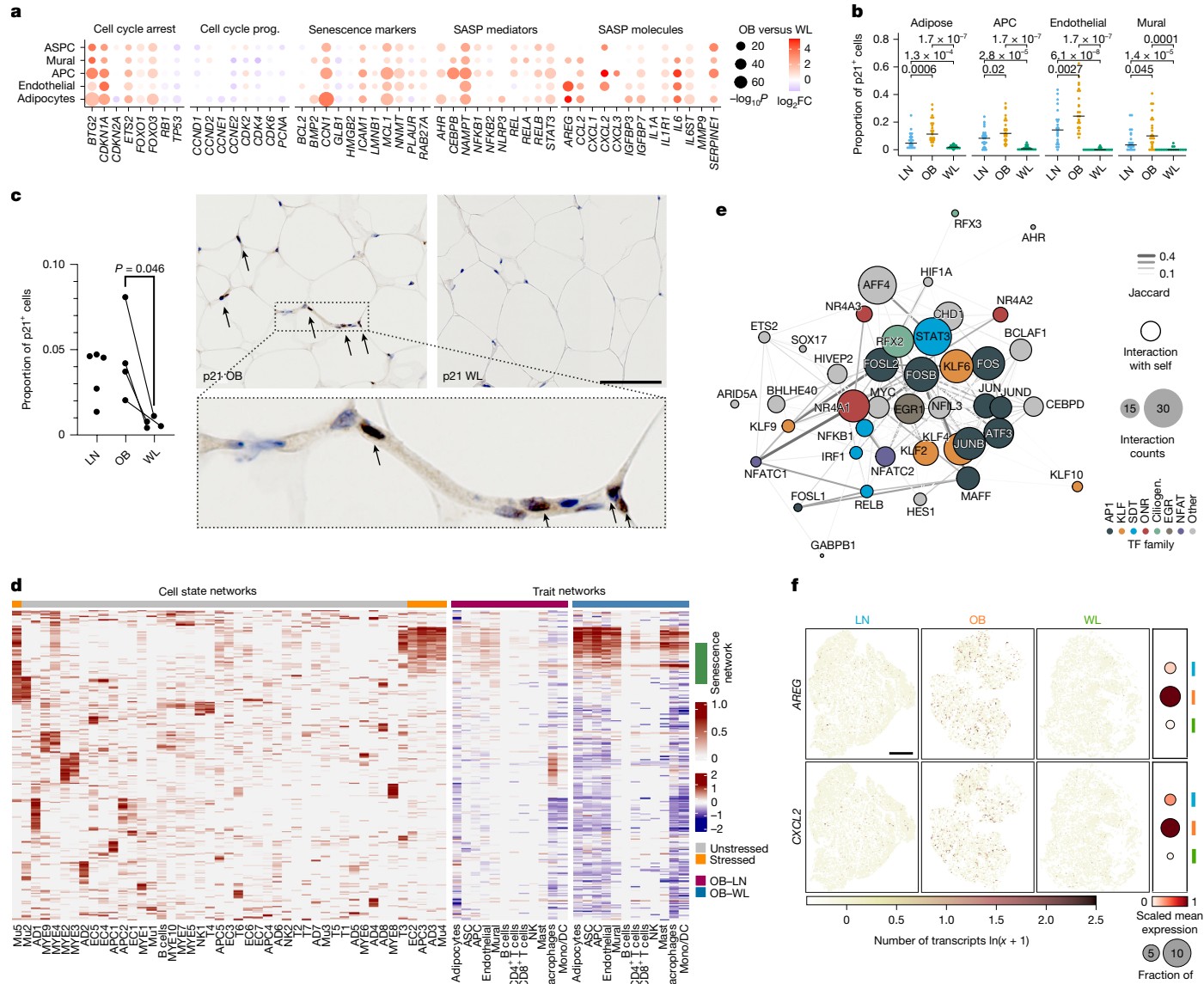

**Fig. 5 | WL potently reverses senescence and its mediators. a**, Differences in the expression of cell cycle and senescence marker genes in WL among vulnerable cell types. Prog., progression. **b**, Mean proportions of p21 (0–1)-positive cells in each sample across conditions in single-nucleus datasets. Separated into vulnerable cell types. Two-tailed Wilcoxon (unpaired LN–OB, LN–WL and paired OB–WL) test, FDR-adjusted $P$-values; $n = 24$ LN; $n = 25$ paired OB–WL donors). **c**, Immunohistochemistry showing the fraction of p21-positive cells (0–1) in tissue sections ($n = 5$ LN, $n = 4$ OB, $n = 4$ WL, paired Student's $t$-test, left). Representative images of a pair of OB and WL tissue sections stained for p21 (scale bar, 100 μm; arrows depict p21-positive nuclei). **d**, Differential gene regulatory networks (left) in each cell state (scaled log₂FC > 0.5 in one or more state versus all other states in that cell type; Wilcoxon two-tailed, FDR < 0.05) and in LN–OB (middle, dark red OB high) and OB–WL (right, red OB high)

comparisons in each cell type (log₂FC, Wilcoxon two-tailed, red OB high). Clustered on cell state networks. Non-significant networks at $P > 0.05$ Bonferroni corrected are coloured grey. **e**, A network of TFs conserved across stressed cell states (scaled log₂FC > 0.4 in three or more stressed cell states; Wilcoxon, FDR < 0.05), coloured by TF family, sized by number of forward interactions with other TFs, encircled if interaction with self (41 of 41 TFs) and linked by the shared number of target genes (width and colour, Jaccard index). AP1, activator protein 1-family TF; KLF, Krüppel-like TF; SDT, signal-dependent TF; ONR, orphan nuclear receptor; Ciliogen., ciliogenesis TF; EGR, early growth response TF; NFAT, nuclear factor of activated T cells TF. **f**, Tissue-wide (50-μm bins) expression of SASP components, *AREG* and *CXCL2*, in representative spatial tissue sections for each condition. Left, number of transcripts. Right, averaged across respective sections. Scale bar, 1 mm.

In summary, these analyses reveal diverse intracellular and extracellular mediators of the degenerative AT senescence cycle and support reversal of AT senescence as a key determinant of the metabolic health benefits of WL.

## Discussion

As the number of people living with obesity surpasses one billion, there has never been a greater need to understand the opposing effects of obesity and WL on metabolic health. Here, we report a high-resolution single-nucleus and spatial atlas of human AT in people with extreme obesity undergoing therapeutic WL and healthy lean counterparts. The simultaneous analysis of obesity and WL enables us to understand core tissue remodelling principles; capture more than 20 cell states that vary with body weight (including degenerative and adaptive populations that bridge cell ontologies); and define molecular pathways, regulatory factors and intercellular signals that may drive tissue injury and subsequent recovery.

Foremost, we reveal selective susceptibility to cellular stress and senescence in subpopulations of metabolic, precursor and vascular

cells, but not in their immune counterparts. This susceptibility is amplified in obesity but is evident in ageing and metabolically unhealthy leanness. We discover that WL has potent mitigating effects on senescence in vulnerable cell types. We predict from transcriptional patterns and previous mechanistic studies[43] that this leads to increased metabolic flexibility in mature adipocytes, improved differentiation capacity in precursors and recovery of vascular abnormalities. Deep molecular phenotyping across cell types, cohorts and modalities enables us to define a tightly conserved regulatory network that may elicit and reinforce human AT senescence, putative upstream triggers, key components of the degenerative AT SASP and vulnerabilities such as *MCL1* that might be exploited with therapy[44]. We conclude that reversal of AT senescence may be central to the multiorgan anti-inflammatory and metabolic benefits of human WL.

By modelling enzymatic gene expression, we show that WL induces global metabolic activation in mature adipocytes, presumably to release stored fuel to meet energy demands. Two prominent activated pathways are BCAA catabolism and lipid cycling (repetitive triglyceride hydrolysis and resynthesis). Contrary to expectation, both pathways are bioenergetic[22], challenging the assumption that WL reduces energy expenditure (at least) in AT[45], suggesting that there is compensation elsewhere and warranting further investigation. Irrespective of this, we propose that pervasive activation of substrate turnover in adipocytes has effects on insulin sensitivity and ectopic lipid that may be crucial to the multiorgan metabolic benefits of human WL.

Despite these effects, our lean cohort was the healthiest, indicating that other factors are involved. In the immune compartment, we confirm that human obesity leads to monocyte and macrophage infiltration and activation to a LAM phenotype. We extend this LAM phenotype to metabolic dysfunction independent of body weight and uncover gene regulatory mechanisms and metabolic pathways implicated in LAM specification and activation. Consistent with an adaptive–maladaptive spectrum, we find that LAMs exist on a continuum. At one extreme, we define a *TLR2–TREM1* inflammatory LAM signature that associates with adiposity and metabolic dysfunction, akin to a proatherosclerotic LAM subtype[46]. We show that after WL there are marked reductions in monocyte and macrophage (and lymphocyte) infiltration and inflammation pathways, which we predict, even in the absence of classical activation, to be anti-inflammatory. Despite overall reductions, we observe persistence of obesity-induced macrophage activation states that are probably epigenetically programmed[47,48]. In mice, AT immune and metabolic cell memory is implicated in weight regain and enhanced inflammation[48,49]. Thus, persistent macrophage activation in human AT may impede complete metabolic recovery, trigger weight regain (a major drawback of all WL interventions) and worsen long-term clinical outcomes[9,50].

We studied WL in its early phase to define potential driver mechanisms. Because of this, we cannot unravel the respective contributions of negative energy balance, weight change and absolute fat mass to the observed tissue and systemic effects. Other limitations include a focus on people without diabetes, the abdominal subcutaneous depot and surgical WL, variable biopsy methods and incomplete capture of rarer immune cells. Previous studies indicate that AT in different locations has important phenotypic differences that may contribute to variability in WL outcomes[8]. However, the degree of weight (fat mass) loss remains one of the strongest predictors of metabolic response, irrespective of intervention[3]. Thus, we anticipate conserved but also context-dependent adaptations across AT compartments and WL methods.

Collectively, our results reveal that WL has significant effects on cellular processes that are known to affect metabolic health and longevity. More broadly, our findings highlight the need for proactive obesity prevention and support the possibility that sustained lifestyle changes could have long-term health benefits mediated through dynamic remodelling of diverse AT cell types. This rich representation of human AT biology and pathophysiology offers a valuable resource for mechanistic and therapeutic exploration.

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

## Methods

### Study design

Single-nucleus RNA sequencing (snRNA-seq) was done in subcutaneous abdominal AT from 25 people with obesity before and after marked WL (paired samples) and in 26 healthy lean controls. Two lean samples were removed as described below. Obese case and control groups were well matched for age, sex and ethnicity. Spatial transcriptomics was done in equivalent groups ($n = 4$ per condition), as were histological cross-validation studies ($n = 4-5$ per condition). All molecular phenotyping was done after overnight fasting. The WL interval was a minimum of 5 months (median 7, range 5–18 months). Median percentage WL was 22% (range 13–33%). Primary snRNA-seq data were integrated with previously published snRNA-seq data in whole subcutaneous AT from nine obese and four lean people to increase obese and lean cell numbers and improve cell annotation[11]. Participant characteristics are provided in Extended Data Table 1.

### Sample collection

The AT samples were obtained intra-operatively from morbidly obese individuals (BMI > 35 kg m$^{-2}$) undergoing laparoscopic bariatric surgery (gastric bypass or gastric sleeve) and healthy controls (BMI < 26 kg m$^{-2}$) undergoing non-bariatric laparoscopic abdominal surgery[51]. Subcutaneous AT was collected from abdominal surgical incision sites. Follow-up subcutaneous AT samples were collected from the peri-umbilical region using needle biopsy more than 5 months after WL intervention. Whole AT samples were snap frozen at collection and stored at −80 °C for future use. Participants were unrelated, between 20 and 70 years of age, from a multiethnic background and free from systemic illnesses not related to obesity. People with treated type 2 diabetes were excluded because of the potential effects of hypoglycaemic medications on AT metabolism. Metabolic characteristics were collected at baseline and follow-up. All participants gave informed consent. The study complies with all relevant ethical regulations and was approved by the London – City Road and Hampstead Research Ethics Committee, United Kingdom (reference 13/LO/0477). Human tissue validation also used samples from the Imperial College Healthcare Tissue Bank, approved by Wales REC3 to release human material for research (reference 17/WA/0161).

### Nucleus isolation

The AT nuclei from individual participants were pooled for library preparation and sequencing to increase efficiency, cohort diversity and study power. Pooled samples were separated by condition to avoid cross-over (4–5 samples per pool; a total of 6 pools per group). Sample pools for each experimental group were processed through to sequencing in lean–obese–WL trios to minimize between-group batch effects. For each participant sample, nucleus extraction was done using a modified version of a previously described protocol[52]. In brief, frozen human AT (about 100 mg) was cut into pieces of less than 0.2 cm and homogenized with 1 ml ice-cold lysis buffer (Tris-HCl 10 mM (Invitrogen, 15567-027), NaCl 10 mM (Invitrogen, AM9760G), MgCl$_2$ 3 mM (Invitrogen, AM9530G), 0.1% NP40 (BioBasic, NDB0385), 0.2 U µl$^{-1}$ RNase inhibitor (Roche, 03335402001)) in a glass dounce homogenizer (Merck, T2690/P0485/P1110, 15 strokes, loose then tight pestles) on ice. After homogenization, samples were transferred through a 100 µM cell strainer (Greiner Bio-One, 542000) into a prechilled tube using ART wide-bore tips (Thermo Scientific, 2079 G). The filtered homogenate was then transferred to 1.5 ml low DNA-bind tubes (Sarstedt, 72.706.700) and centrifuged at 500$g$ and 4 °C for 5 min. After lipid/supernatant removal, the nuclei pellet was resuspended in 1 ml wash buffer (PBS with 0.5% BSA (Invitrogen, AM2616) and 0.2 U µl$^{-1}$ RNase inhibitor), transferred to new 1.5 ml low DNA-bind tubes and recentrifuged at 500$g$ and 4 °C for 5 min. After repeat lipid/supernatant removal, the nuclei pellet was resuspended in 300 µl wash buffer

containing DAPI (Thermo Scientific, 62248) at 0.1 µg ml$^{-1}$ to stain nuclei, and filtered through a 35 µM cell strainer into a fluorescence-activated cell sorting (FACS) tube (Falcon, 352235) on ice. At this point, the isolated nuclei from 4–5 samples from the same experimental group were pooled before sorting by flow cytometry.

FACS was used to clean up residual debris and lipid from isolated nuclei and to remove doublets. Pooled nuclei were sorted on a BD FACS Aria SORP. The sheath tank was bleach cleaned before each run and nuclease-free PBS (1×) (Invitrogen, AM9625) was used as sheath fluid. A 405 nm laser was used to excite DAPI, and emission was collected using a 450/50 nm bandpass filter. Single nuclei were selected by gating on the first DAPI-positive band on the DAPI versus forward scatter (FSC) plot and then subsequently gating on side scatter (SSC) versus FSC and FSC A versus FSC H to ensure better debris and doublet removal. All sorts were performed using an 85 µm nozzle. The sorted nuclei were collected into a BSA- and RNase inhibitor-rich collection buffer (70 µl of PBS with 1.375% BSA and 2.15 U µl$^{-1}$ RNase inhibitor) in low DNA-bind tubes kept at 4 °C. After sorting, nuclei were centrifuged at 500$g$ for 5 min at 4 °C to pellet. Supernatant was removed to leave about 40 µl, which was used to resuspend pellets with a wide-bore pipette tip.

### Single-nucleus library preparation and next-generation sequencing

Pooled single-nucleus suspensions were used to generate barcoded single-nucleus libraries for next-generation sequencing. For each pool, 5,000–10,000 nuclei were co-encapsulated with 10x barcoded gel beads to generate gel beads in emulsion (GEMs) using a 10x Chromium Controller and a 10x Genomics Single Cell 3′ v.3.1 kit, according to the manufacturer's instructions. After GEM-RT and clean-up, the quantity and fragment size distribution of amplified cDNAs derived from barcoded single-cell RNAs were assessed using an Agilent 2100 Bioanalyzer High Sensitivity DNA assay. From this cDNA, snRNA-seq libraries were constructed and sequenced (Illumina NextSeq2000) in three batches, containing equal numbers of obese, lean and control library pools, to minimize between-group batch effects. Each unique library was sequenced to a minimum depth of more than 20,000 paired-end reads per nucleus (read 1, 28 base pairs (bp) and read 2, 90 bp, with unique dual 10-bp indexes). Raw sequencing data were demultiplexed and analysed using CellRanger v.5.0.1 and bcl2fastq v.2.20.0. Libraries were demultiplexed using CellRanger mkfastq based on the sample indices (allowing one mismatch), and the CellRanger count pipeline was used to perform alignment against human genome GRCh38 (using STAR), filtering and counting unique molecular identifiers (UMIs) (including introns).

### Single-nucleus quality control

For each pooled library, raw count matrices from CellRanger were processed using CellBender[53] (--epochs 150-200, --learning-rate 0.0001-0.00005) to remove ambient RNA molecules and random barcode swapping, and filter inferred cells. The number of expected cells was based on CellRanger estimations. Filtered count matrices were processed separately using Seurat[54] and SeuratObject. Low-quality cells with low read or gene counts (less than 1,000 UMIs or less than 400 genes), low complexity ($\log_{10}$(genes per UMI) < 0.85) and high mitochondrial or ribosomal fractions (greater than 5%) were removed from each pooled dataset. Clean libraries were normalized and transformed (sctransform v.2 regularization[55]) to stabilize count variances. Potential doublet nuclei were detected using three approaches: expression-based DoubletFinder[56], using doublet estimates from genotyping to set the expectation; genotype-based, Vireo[57] (details below); and iterative clustering and detection of clusters with high expression or genotype-based doublet fractions. Assigned doublets, ambiguous cells and doublet clusters were then removed and singlet-only datasets were retransformed. Participant-level annotation

information from genotyping was then added to generate high-quality cell datasets.

## Participant annotation from genotype information

Genotype information present in the RNA sequencing reads was aligned to existing genome-wide genotyping to attribute specific cells to specific participants in each sample pool. Participant-level genotype data were generated from whole blood using Illumina Infinium OmniExpress-24 v.1.2 bead chips. Directly genotyped single-nucleotide polymorphisms (SNPs) with call rates of less than 90%, minor allele frequency of less than 0.01, Hardy–Weinberg equilibrium $P < 1 \times 10^{-6}$, SNPs on sex chromosomes and duplicated SNPs were removed. After quality control, 649,007 SNPs were taken forward for imputation. SHAPEIT[58] (v.2.r900) was used to infer haplotypes, and imputation was done in IMPUTE2 (v.2.3.2)[59] using a 1,000 genomes reference panel phase 3 (all ancestries). Each chromosome was divided into 5-megabase chunks for imputation and merged at the end. A random seed was supplied automatically. An effective population size ($N_e$) reflecting genetic diversity was 20,000, as recommended when using a multi-population reference panel. After imputation, genotype data were available for 81,656,368 SNPs.

Cell-level SNP data were generated for each pooled sample using cellsnp-lite[60] (using the combined imputed SNP list as the reference). Cell-level SNP data were then intersected with participant-specific genotype references in Vireo[57] to identify variants that segregated the samples, and we used these variants to demultiplex participant specific cells, ambiguous cells and doublets. A range of cellsnp-lite MAF settings were tested and MAF > 0.05 was selected to maximize singlet recovery. Participant-level cell annotations were then incorporated into pre-cleaned high-quality cell datasets.

## Integration

High-quality, doublet-removed cell libraries containing participant-level annotations were then integrated to a unifying atlas. Two samples, one with very high lymphocyte counts and one with very few cells, were removed at this stage, leaving 24 samples in the lean group. A further 13 whole subcutaneous AT samples from obese and lean people in a previously published dataset[11] were also incorporated in the integration phase to increase cohort diversity and improve cell annotation. Of note, only samples meeting the following criteria were selected: whole tissue; nucleus only; subcutaneous depot; and BMI < 26 or BMI > 30 kg m$^{-2}$. Previously published samples were individually reprocessed from raw counts using thresholds equivalent to our own datasets.

To integrate our dataset with the previously published dataset[11], we updated the gene IDs from the latter to match the same Ensembl release. Both datasets were then normalized to 10,000 counts per nuclei before proceeding with downstream analysis. To minimize any sample-driven effect for cell-type identification, we took a three-step approach. First, we regressed out the effects of number of original counts, as well as the percentage of mitochondrial and ribosomal genes. Then we calculated the PCA space on the highly variable genes, detected by Scanpy[61], followed by correction of the PCA space with Harmonypy[62] using samples as batches. Finally, we used BBKNN[63] with samples as a batch to identify neighbourhoods.

## Analysis overview

Cell type and state annotation was done in the combined (our own and that from ref. 11) integrated dataset. Primary exploratory analyses were performed in our own dataset, which was processed in experimental group trios (lean–obese–WL) to minimize batch effects and comprised paired obese–WL samples and age-, sex- and ethnicity-matched lean controls. Differential neighbourhood abundance and expression analyses between groups (in which biological, technical and batch covariates could be adjusted for) were repeated using the combined dataset to verify reproducibility.

## Cell annotation

We identified the main cell types with unbiased clustering, using a low-resolution (0.15) Leiden algorithm, and each cell type was annotated according to known markers. To identify cell states, we isolated the barcodes for each of the main cell type identities, except for mast and lymphatic endothelial cells, owing to low numbers. Each cell type was then reintegrated and reclustered twice, as described above. First, we used a high-resolution Leiden (1.2 or higher) to identify barcodes that contained a mixed signature, with markers of different lineages. These barcodes were flagged as 'unassigned' and were excluded from any downstream analysis. Then, we removed these barcodes and proceeded with the second round of reintegration and clustering. Resolution varied across cell types (0.65 or higher), with myeloid cells requiring the highest Leiden to identify rare, known cell types (2.25). Clusters that were similar to each other and had no unique identifiable features between them were merged. Cell states were annotated based on a mix between unbiased and known markers. To identify unbiased markers, we used Scanpy's rank_gene_groups function to perform a Wilcoxon test.

## Compositional analyses

To analyse changes in cellular composition, we used a neighbourhood graph-based approach in Milo[64]. We performed comparisons of lean–obese and WL–obese groups, adjusting for biological covariates in the lean–obese analyses. Neighbours were recalculated with BBKNN using samples as a batch, restricted to the comparison groups (lean–obese and WL–obese). To analyse global shifts, we used Milo on all cell types together and within each cell type to analyse shifts in cell state composition. Only neighbourhoods containing at least 90% of a single cell type or state were considered neighbourhoods, and those with a spatial FDR < 0.1 were considered significant. Fasting insulin adjusted for BMI abundance analyses were carried out in steady state lean and obese samples, using lean–obese neighbourhoods, adjusting for biological covariates.

## Metabolic analyses

The metabolic profiles of different cells were inferred using flux-based analysis modelling in COMPASS[65]. For this, we created an expression matrix for every main cell type, consisting of the mean expression of each gene per sample. These matrices were then used to run COMPASS. Statistical analysis to compare conditions was performed with a Wilcoxon test for every reaction, using their COMPASS score. COMPASS plots consisted of both positive and negative reactions grouped by their defined subsystem.

## Differential expression analyses

Differential expression analyses were carried out between obese cases and controls, and between obese–WL pairs, in Nebula[66] using negative binomial mixed-effect models to correct for subject- and cell-level correlation structure. In all comparisons, further thresholding was applied (mitochondrial fraction less than 1% and ribosomal fraction less than 1%) to minimize false discovery, and fractions of mitochondrial and ribosomal counts were incorporated as technical covariates; in obese–lean comparisons, age, sex and ethnicity were included as covariates; in obese–WL comparisons of paired samples, biological covariates were not included. Statistical significance was inferred at $P < 0.05$ Bonferroni corrected for obese–WL pairwise comparisons (where power was higher) and FDR < 0.01 for lean–obese comparisons. Cell type and state differences were examined using Scanpy's rank_gene_groups function to perform a Wilcoxon test, as were spatial differences in gene expression within cell types between conditions. Amphiregulin (AREG), which is known to be secreted[67], was added to the curated secretory protein list from the Human Protein Atlas[41] for comparisons in stressed and basal cells.

### Inference of regulatory networks

To infer regulon activity, we used the Python implementation of the SCENIC[68] pipeline (pySCENIC). The expression matrix used consisted of nuclei from all 3 conditions, downsampled to the same number of nuclei (20,000 each). Genes that were expressed in all nuclei, or in less than 5% of nuclei for any given cell state, as well as mitochondrial, ribosomal, haemoglobin, non-coding, antisense, contig and microRNA genes, were also removed from the analysis. For TF binding sites, we used the Encode 2019/06/21 ChIP-seq hg38 refseq-r80 10 kilobases up and down database. Only regulons with a minimum of five target genes were considered. Analyses in adipocytes were restricted to all TF genes and genes in dysregulated metabolic pathways from COMPASS. Differential networks between cell states and within cell types between conditions were identified by comparing cell-level network scores between groups (non-parametric Wilcoxon rank-sum test). Significance was inferred at $P < 0.05$ (Bonferroni corrected). Within a cell state, fold changes were scaled for visualization.

### Cell–cell communication

We used CellChat[69] to infer intercellular communication, based on known receptor–ligand interactions. For the purpose of this analysis, to compare the differences between each condition, cellular communication was inferred for each condition separately. Each condition was down-sampled to 20,000 barcodes to avoid any confounding effects arising from higher cell numbers in obese and lean groups, and cell types with very low numbers were removed because these cell types often have higher mean gene expression owing to low cluster background. To analyse the differential communication between two conditions, we used the rankNet function in CellChat to obtain overall signalling differences, as well as pairwise comparison with each cell type as a sender and as a receiver. To analyse communication at the cell state level, we performed a condition-agnostic analysis to maintain cell states with low numbers of nuclei. For intra- and inter-niche communication analyses, because of the lack of most ligand–receptor pairs in the Xenium gene panel, we imputed spatial data using ENVI[70]. This was done for each condition separately, training on the single-nucleus data for each condition. We did this step ten times and averaged the results in a final imputed expression matrix because of the stochastic nature of imputation. Imputed genes with low expression (below the mean across all genes, the gene-level quality control) and those with below the mean for that gene (cell-level quality control) were removed.

### Metabolic and senescence scores

Gene list scoring was done in Scanpy using the score_genes function, with the normalized ln expression and a control size of 50. Senescence signatures were obtained from MSigDB[71,72]. Housekeeping genes were obtained from the 20 most stable human transcripts in the Housekeeping Transcription Atlas[73], supplemented with commonly used housekeeping genes (*RRN18S*, *ACTB*, *GAPDH*, *PGK1*, *PPIA*, *RPL13A*, *RPLP0*, *ARBP*, *B2M*, *YWHAZ*, *SDHA*, *TFRC*, *GUSB*, *HMBS*, *HPRT1* and *TBP*). The BCAA score was performed using the genes associated with the respective pathways on COMPASS.

### Pathway analyses

Pathway analyses of differentially expressed genes were done in ClusterProfiler[74] using the Over Representation Analysis and MSigDB[71,72] datasets (H, C2 and C5) as inputs. All genes present in the comparison datasets were used as background. Significant pathway enrichment was inferred at FDR < 0.01.

### Tissue processing for spatial transcriptomics and histology

Frozen stored AT samples (stored at −80 °C) were directly thawed in a 4% paraformaldehyde solution and kept at 4 °C for 24 h. Samples were then transferred to a 70% ethanol solution and stored until paraffin embedding. Ethanol-dehydrated samples were cleared with xylene, infiltrated with molten wax using the Sakura Tissue Tek VIP6 vacuum infiltration processor and embedded in paraffin using the Sakura Tissue Tek TEC5 embedding system.

### Spatial transcriptomic preparation

**Slide preparation.** Formalin-fixed paraffin-embedded (FFPE) blocks were stored at 4 °C. Xenium slides stored at −20 °C were equilibrated to room temperature for 30 min before sectioning. The FFPE blocks were rehydrated in an ice bath with distilled water for 10–30 min and sectioned at 5 µm thickness. Sections were floated in a 42 °C water bath and slides containing tissue sections were incubated at 42 °C for 3 h and then dried overnight at room temperature in a desiccator. Slides were kept at 4 °C in a desiccator until use. All histology was done in RNase-free conditions using sterilized equipment.

**Technical pilot.** A technical pilot was done on a single frozen stored AT sample separated into three sections for fixation at 24 h, 48 h and 72 h to evaluate the effects on tissue integrity (H&E) and transcript recovery using the 10x Xenium Human Multi-Tissue and Cancer Panel (P/N 1000626), with two slides and one tissue section for each fixation time/slide (Institute of Developmental and Regenerative Medicine (IDRM), Oxford).

**Panel design.** A 10x Xenium Human Multi-Tissue and Cancer Panel (P/N 1000626) supplemented by 100 custom genes was selected to annotate prominent cell types, states and effector pathways identified in single-nucleus datasets.

**Xenium in situ transcriptomics.** The FFPE tissues were analysed on a 10x Xenium Analyser following 10x Genomics Xenium in situ gene expression protocols CG000580, CG000582 and CG000584. In brief, 5-µm FFPE tissue sections on Xenium slides were deparaffinized and permeabilized to make the mRNA accessible. Gene panel probes were hybridized for 20 h overnight followed by washing, ligation of probe ends to targeted RNAs, generating circular DNA probes with high specificity. Rolling circle amplification was used to generate hundreds of copies of gene-specific barcodes for each RNA-binding event, resulting in a strong signal-to-noise ratio. Background fluorescence was quenched chemically to mitigate tissue auto-fluorescence. Tissues sections were stained with DAPI nuclear stain and Xenium slides were loaded onto the Xenium instrument for imaging and then decoding of image data to transcripts. Secondary analysis to segment cells and assign transcripts was performed on-instrument (Xenium Analyser v.1.7.1.0). Xenium Explorer was used to evaluate the initial data quality and visualize morphology images, transcript localization at subcellular resolution, segmentation and data clustering.

**Post-Xenium processing.** After Xenium in situ transcriptomics, slides were kept in PBS and stored at 4 °C for up to 24 h. For immunofluorescence staining, slides were washed three times in PBS for 5 min and then incubated in CF 594 wheat germ agglutinin (1:200; Biotium, 29023-1) for 20 min. Slices were then rewashed three times with PBS, and tissue stained with DAPI (1:5,000; Thermo Scientific, 62248) for 10 min at room temperature. Finally, sections were rewashed as before and then mounted using antifade medium Vectashield (Vector Laboratories, H-1000). Full slide scans for the immunofluorescence channels were performed at 20× magnification using a ZEISS Axio Scan.Z1 slide scanner.

### Spatial data analysis

Xenium data were analysed by three different methods, depending on the purpose of the analysis. Regardless of the type of analysis, only transcripts with a quality value higher than 35 were considered.

To plot transcript and score densities, regardless of cell type we took a segmentation-free approach creating 50-µm bins using the transcript

coordinates provided by Xenium. Only bins that contained more than ten transcripts were kept for downstream analysis.

For cell-type identification, we took the nucleus segmentation from Xenium and assigned only transcripts within 2 μm of each nucleus (selected to maximize recovery of transcripts but minimize the capture of known cross-contaminating marker transcripts from adjacent cells, designated nucleus segmentation). The resulting matrices were then imported into Scanpy for analysis. Here, only nuclei with more than 40 transcripts were kept for downstream analysis. Clustering was performed similarly to the single-nucleus data, with Harmonypy[62] and BBKNN[63] used to correct batch effects in the PCA and neighbourhoods, respectively. However, here gene expression was scaled using Scanpy's[61] scale function to give more weight to low-expression genes. A low-resolution Leiden algorithm was then used to identify the main cell types, and cell states were identified by reintegrating and reclustering each of these cell types individually. Clusters were labelled to match the single-nucleus reference. Ambiguous clusters were labelled 'unassigned'. To delineate rarer LAM subtypes in the spatial dataset we used CellTypist for label transfer[75], creating a model trained on the single-nucleus LAM subtypes and applying a 'best match' prediction on the MYE2 LAM spatial cluster.

To correlate genes with adipocyte size, we performed a semi-manual segmentation using ImageJ, designated boundary segmentation. WGA staining, performed after the Xenium run, was aligned to the Xenium data using the DAPI channel as a guide and utilized for segmentation. To avoid any issues arising for multiple adipocytes being merged in the segmentation, we manually closed some gaps where the WGA staining was not strong enough to be detected by the binary threshold of ImageJ. We then used the Analyse particles function of ImageJ to detect each object and measure the area and centroid coordinates. Furthermore, we created a separate table with coordinates for each pixel contained in each object. To assign transcripts to the ImageJ objects, and to remove any noise derived from other cell types, we first removed any transcript that was assigned to non-adipocytes during the nuclei segmentation. We then created a distance tree between the remaining transcript coordinates and the pixel coordinates obtained for every ImageJ object. This was achieved using the KDTree function from Scipy's spatial module. Adipocyte transcripts that were found on the cell boundary were assigned to the closest adipocyte(s) (any adipocyte within 2 μm of the nearest segmented pixel). Only objects with an area greater than 1,000 μm² and less than 25,000 μm² were considered as adipocytes for this analysis. As larger objects were found to have higher probability of capturing more transcripts, gene expression was normalized to the total number of counts per cell. Clustering was done as described above, using a high resolution to identify and then remove fine clusters containing contaminating transcripts from other cell types. A Spearman correlation was done to investigate which genes correlated with adipocyte area.

Finally, to cluster cells in spatial niches, we made use of Scipy's KDTree function to create a distance tree between every cell in each sample. We then created a neighbourhood matrix by counting, for each cell, the number of proximate cells (within 300 μm) at a cell state level. Because adipocyte sizes increased in obesity, cells in lean samples had roughly twice the number of neighbouring cells that cells in obese samples did. To prevent this from biasing the niche clustering, the neighbourhood matrix was normalized such that each cell was represented by the percentage of neighbouring cells in each cell state. To cluster cells into niches, we created an anndata object of the neighbourhood matrix for use in Scanpy and corrected for batch effects with Harmony and BBKNN before Leiden clustering. Very similar clusters, driven by small fluctuations, were merged into the AD niche.

## Tissue immunohistochemistry

The FFPE blocks were sectioned at 5 μm thickness for immunohistochemistry and immunofluorescence. Sections were deparaffinized and hydrated, and then heat-mediated antigen retrieval was done in an EDTA-based pH 9.0 solution. Endogenous peroxidase was quenched with 3% hydrogen peroxide. Sections were incubated with rabbit monoclonal to p21 Waf1/Cip1 (1:50 dilution; Cell Signalling, 2947, clone 12D1), followed by anti-rabbit IgG conjugated with polymeric horseradish peroxidase linker (25 μg ml⁻¹; Leica Bond Polymer Refine Detection, DS9800). DAB was used as the chromogen and the sections were then counterstained with haematoxylin and mounted with DPX. Immunohistochemistry was performed on a Leica BOND RX. To evaluate p21-positive cells, full virtual slide scans were loaded into QuPath 0.5.1 (ref. 76) and the positive cell detection module was used to count the total haematoxylin and DAB-positive nuclei in two slices per sample. The fraction of p21-positive cells relative to the total cell number was then calculated for each slice, and the mean was used for between-group analyses.

## Tissue immunofluorescence

Tissue sections of 5 μm were deparaffinized by submerging three separate times in Histoclear (National Diagnostics, HS-200) for 5 min and then rehydrated by submerging in a series of graded alcohol solutions of decreasing concentrations for 5 min each. After rehydration, antigen retrieval was done by heating the samples in 10 mM sodium citrate buffer, pH 6 (Abcam, ab64236) for 5 min in a decloaking chamber (Biocare Medical, DC2012-220V). The sections were then permeabilized in 0.2% Triton X (Sigma-Aldrich, X100-500mL) in PBS for 10 min and subsequently blocked in 1× ACE blocking solution (Bio-Rad, BUF029) for 30 min. After blocking, sections were incubated in primary antibody solutions diluted in 0.5× block ACE at 4 °C overnight: anti-NAMPT (1:200, Affinity Biosciences, DF6059); anti-TREM2 (clone D8I4C, 1:400, Cell Signalling, 91068); or anti-TLR2 (clone TL2.1, 1:400, Invitrogen, 14-9922-82). After primary antibody removal, the tissue was washed in PBS and then incubated with secondary antibody, goat anti-rabbit Alexa Fluor 488 (1:200, Invitrogen, A11034), donkey anti-rabbit Alexa Fluor Plus 488 (1:250, Invitrogen, A32790) or goat anti-mouse Alexa Fluor Plus 647 (1:250, Invitrogen, A32728) in 0.5× block ACE for 45 min at room temperature. For NAMPT, sections were incubated with DyLight 594 Lycopersicon Esculentum Lectin (1:250, Invitrogen, L32471) for 20 min (room temperature), rewashed with PBS and then stained with a DAPI solution (1:5,000, Thermo Scientific, 62248) for 10 min at room temperature. For TREM2/TRL2 at CLS, only DAPI was used. Finally, sections were washed and mounted using antifade medium Vectashield (Vector Laboratories, H-1000). For each sample, representative images were taken at 40× magnification (NAMPT) or 20× (CLS) using a Leica SP8 DLS confocal microscope. Image analysis was done in QuPath 0.5.1 (ref. 76). To quantify the NAMPT:lectin ratio, the positive pixel area of the NAMPT and lectin channels was measured in two z-stack maximum projection images per sample using the pixel classifier module. Measurement precision was evaluated between two images per sample (to confirm low within-sample variability) and the mean sample intensity was used for between-group analysis.

## Macrophage isolation and HPG uptake

We used a modified SCENITH-based approach to evaluate human macrophage metabolic pathways ex vivo[77,78]. Fresh subcutaneous AT was cut into approximately 2-mm pieces with 30 ml HBSS (Gibco, 14175-053) in a 50 ml tube, washed and collected using a 100 μM cell strainer. Tissue was digested for 20 min at 37 °C with 3 mg ml⁻¹ collagenase II (Sigma C6885) in methionine-free RPMI (Sigma, R7513), 65 mg l⁻¹ L-cystine dihydrochloride (Sigma, C6727), 1× GlutaMAX (Gibco, 35050061), 10% dialysed fetal bovine serum (FBS, Gibco, A3382001). Digested tissue was filtered through a 100 μm strainer and digestion was terminated by adding methionine-free RPMI containing 10% FBS, followed by centrifugation (300g at 4 °C for 7 min). After resuspension in methionine-free RPMI (65 mg l⁻¹ cystine, 10% FBS, 1× glutamax), cells were plated (160 μl) into wells on a 96-well V-bottomed plate. Cells were methionine starved for a further 15 min (total starvation of 45 min including digestion and isolation) before treatment with inhibitors or control media (40 μl) for

15 min. The four treatments were medium, 2-deoxy-D-glucose (2-DG; 100 mM final concentration; Sigma, D8375), oligomycin (2 μM final concentration; Sigma, 495455) and 2-DG plus oligomycin (100 mM and 2 μM final concentration, respectively). Homopropargylglycine (HPG; Cayman Chemical, 11785) was then added to wells at a final concentration of 500 μM and incubated for 30 min to initiate cell HPG uptake. An additional well received cells and media but no HPG and no treatment (click chemistry negative control). After HPG uptake, cells were stained with zombie aqua live/dead stain (1:500 in PBS; BioLegend, 423101) for 20 min at room temperature in the dark, washed with PBS and then fixed with 2% PFA for 15 min.

## Click chemistry, staining and FACS analysis

Fixed cells were permeabilized (0.1% saponin and 1% BSA in PBS) for 15 min, washed with click buffer (100 mM Tris-HCl, pH 7.4; Invitrogen, 1556-027) and incubated with Fc receptor blocker (25 μg ml$^{-1}$ in PBS; Fc1, BD Biosciences, 564765) for 10 min. Cells were rewashed and incubated in 100 μl of click reaction mix in the dark at room temperature for 30 min. Click reaction mix was made sequentially, adding $CuSO_4$ (final concentration, 0.5 mM; Sigma, 209198), THPTA (final concentration, 2 mM; Antibodies.com, A270328), sodium ascorbate (final concentration, 10 mM; Sigma, A7631) and then AZDye 555 (final concentration, 25 μM; Vector Laboratories, CCT1479) to click buffer (final concentration, 100 mM Tris-HCl).

After click chemistry exposure, cells were washed using FACS buffer (PBS, 1% BSA, 5 mM EDTA, 25 mM HEPES) and stained with antibody mix (FACS buffer, anti-CD45 FITC (1:20; H130; BioLegend, 304006), anti-FOLR2 APC (1:20; 94b/FOLR2; BioLegend, 391705), anti-CD9 APC-fire (1:20; H19α; BioLegend, 312114), Fc block reagent (25 μg ml$^{-1}$)) at 4 °C in the dark for 30 min. After rewashing, cells were filtered (35 μM cap strainer) for FACS analysis.

Spectral flow cytometry was done on a Sony ID7000 in standardization mode. The ID7000 software was used to calculate distinct spectral signatures for each fluorochrome based on single stained controls. Fluorochrome signatures, together with autofluorescence signatures identified in unstained aliquots of each sample using the AF finder software feature, were used to unmix the signals in fully stained samples with the built-in WLSM algorithm. Unmixed signals were used for gating (Extended Data Fig. 2i and Supplementary Fig. 1) and analysis of median fluorescence intensity in FlowJo.

## In vitro stress studies

Immortalized human adipose-derived stromal cells (Bmi-1/hTERT, iHASC) were acquired from Applied Biological Materials (T0540). For differentiation experiment cells, iHASC were seeded in six-well plates. Differentiation was induced at confluence using growth medium (DMEM/F-12 (Gibco, D8437), 10% FBS (Gibco, F7524), 2 ng ml$^{-1}$ rhbFGF (Z101455), 1% gentamicin (G255)) supplemented with 10 μg ml$^{-1}$ insulin (Actrapic, Novo Nordisk), 500 μM 3-isobutyl-1-methylxanthine (Sigma, I5879), 1 μM dexamethasone (Sigma, D4902) and 2 μM rosiglitazone (Sigma, R2408) for 15 days. Etoposide (Sigma-Aldrich, E1383) was used to induce the DNA damage stress response[79]. From day 1 to day 5 of differentiation, cells were treated with DMSO (Fisher-Scientific, BP231100) (control) or etoposide 5 μM or 10 μM. Medium was refreshed every 3 days. For stress-marker experiments, undifferentiated cells were seeded in 96-well plates and treated with DMSO control or etoposide (5 μM and 10 μM) at 80% confluence.

O-Red-oil (ORO) staining was performed as previously described[51]. In brief, cells were fixed with formalin, washed with sterile water, treated with 60% isopropanol and stained with ORO solution (Sigma, O0625) and DAPI (1:5,000). After washing, stained cells were imaged on an Evos m7000 (Thermo Scientific) capturing a minimum of 100 fields at 20× magnification per well. Marker quantification was done in Qupath; nuclear segmentation was done using the cell-detection module in the DAPI channel. Mean ORO intensity was quantified in a 15 μm radius

to each nucleus. Positive cells were called empirically at a threshold greater than 32.2, 8-bit depth. The proportion of ORO-positive cells to the total number of nuclei was calculated.

For stress-marker quantification, after etoposide and media treatment, 96-well plates were fixed in 10% formalin for 10 min and then washed with PBS. The following primary antibodies were used for staining: anti-STAT3 (clone 124H6, 1:500; Cell Signalling, 9139S) and anti-JUN (clone 60A8, 1:500; Cell Signalling, 9165S). Otherwise, staining procedures used the same steps, reagents and concentrations as for tissue immunofluorescence. After staining, wells were kept in PBS and imaged using a high-throughput fluorescent microscope IN Cell Analyzer 2500HS (Cytiva, objectives 20× for JUN and 40× for STAT3). Positive cells were determined using IN Carta image analysis software (v.1.14), based on the nuclear fluorescence intensity for the target protein (empirical positive threshold for JUN, greater than 396.9, and STAT3, greater than 505.3, 16-bit depth). Data were expressed as the percentage of positive cells (JUN or STAT3) of the total number of nuclei.

## Statistics and reproducibility

Unless otherwise stated, significance was inferred at $P < 0.05$ for single-variable tests and FDR < 0.05 for multiple-hypothesis tests. For spatial datasets, where representative images are provided, all analyses were repeated in $n = 4$ samples per group. For histological verification, where representative images are shown, all analyses were repeated in $n = 4$–5 samples per group.

## Reporting summary

Further information on research design is available in the Nature Portfolio Reporting Summary linked to this article.

## Data availability

Raw single-cell and spatial transcriptomic datasets have been deposited on the Gene Expression Omnibus (accessions GSE295708 and GSE295862, respectively). Integrated single-nucleus and Xenium objects, together with auxiliary files, are available at the Single Cell Portal (accessions SCP3116 and SCP3117, respectively). The following publicly available datasets were used in this study: human AT single-nucleus transcriptomic data (Single Cell Portal, SCP1376; and GEO accession, GSE176171); human reference genome (https://cf.10xgenomics.com/refdata-gex-GRCh38-2020-A.tar.gz); Molecular Signatures Database (https://www.gsea-msigdb.org/gsea/msigdb/); secreted proteins in the Human Protein Atlas (https://www.proteinatlas.org/humanproteome/tissue/secretome); motifs for SCENIC (https://resources.aertslab.org/cistarget/databases/homo_sapiens/hg38/refseq_r80/tc_v1/gene_based/); and human GWAS (https://www.ebi.ac.uk/gwas/). Source data are provided with this paper.

## Code availability

Data analysis pipelines used in this work can be obtained from https://github.com/WRScottImperial/WAT_single_cell_analysis_Nature_2024.

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

**Acknowledgements** This work was funded by the Medical Research Council UK (MR/K002414/1, MC_UP_1605/7; to W.R.S.), the Wellcome Trust (219602/Z/19/Z; to W.R.S.), the National Institute for Health Research Imperial Biomedical Research Centre (BRC, CL-2018-21-501; to W.R.S.) and Diabetes UK (22/0006436; to W.R.S., T.T. and M.N.). The MRC Laboratory of Medical Sciences flow cytometry, genomics, bioinformatics and computing, and microscopy facilities provided support for the single-nucleus isolation and clean-up, single-nucleus and spatial library preparation, sequencing demultiplexing, alignment and feature counting, and high-resolution immune histochemistry and immune fluorescence microscopy. We thank J. Gil for insights into senescence pathobiology and detection.

**Author contributions** Participant recruitment, human sample collection and tissue processing: I.D., M.Ertugrul, J.K., A.P., M.A., A.J., M.Elkalaawy, B.M., M.H., P.O., A.R.A., R.L.B., J.C.C., J.S.K., T.T. and W.R.S. Nucleus isolation and library preparation: L.M., I.A., I.D. and B.P. Spatial transcriptomic library preparation and data generation: I.A., K.L.C., L.M., G.M., H.K., R.G., A.C., O.M., L.G., J.H. and F.I. Histology and tissue sectioning: G.M., H.K. and R.G. Tissue imaging and image analysis: G.M., S.B., S.N., Y.R. and A.M.A.M. Ex vivo and in vitro experimental studies and analysis: L.M., G.M., J.C., A.P., D.G.R. and A.M.A.M. Single nucleus and spatial data analysis: A.M.A.M., D.B., W.Z., M.J. and W.R.S. Study conception: M.N., T.T., D.B., J.C.C., J.S.K. and W.R.S. Data interpretation: A.M.A.M., D.B., M.N., T.T. and W.R.S. Project coordination: A.M.A.M. and W.R.S. Manuscript writing: A.M.A.M., L.M., G.M., D.B., M.N., T.T. and W.R.S.

**Competing interests** R.L.B. participated in committees or advisory boards for ViiV Healthcare, Gila Therapeutics, Novo Nordisk, Pfizer, Eil Lilly, the Royal College of Physicians, NHS England, the National Institute for Health and Care Excellence, the British Obesity and Metabolic Surgery Society, the National Bariatric Surgery Registry, the Association for the Study of Obesity, the Obesity Health Alliance, the International Federation for the Surgery for Obesity and Metabolic Diseases, Obesity Empowerment Network UK and the European Society for Endocrinology. R.L.B. has undertaken consultancy work for Novo Nordisk, ViiV Healthcare and Epitomee Medical, and is employed by Eli Lilly. The remaining authors declare no competing interests.

**Additional information**
**Correspondence and requests for materials** should be addressed to William R. Scott.

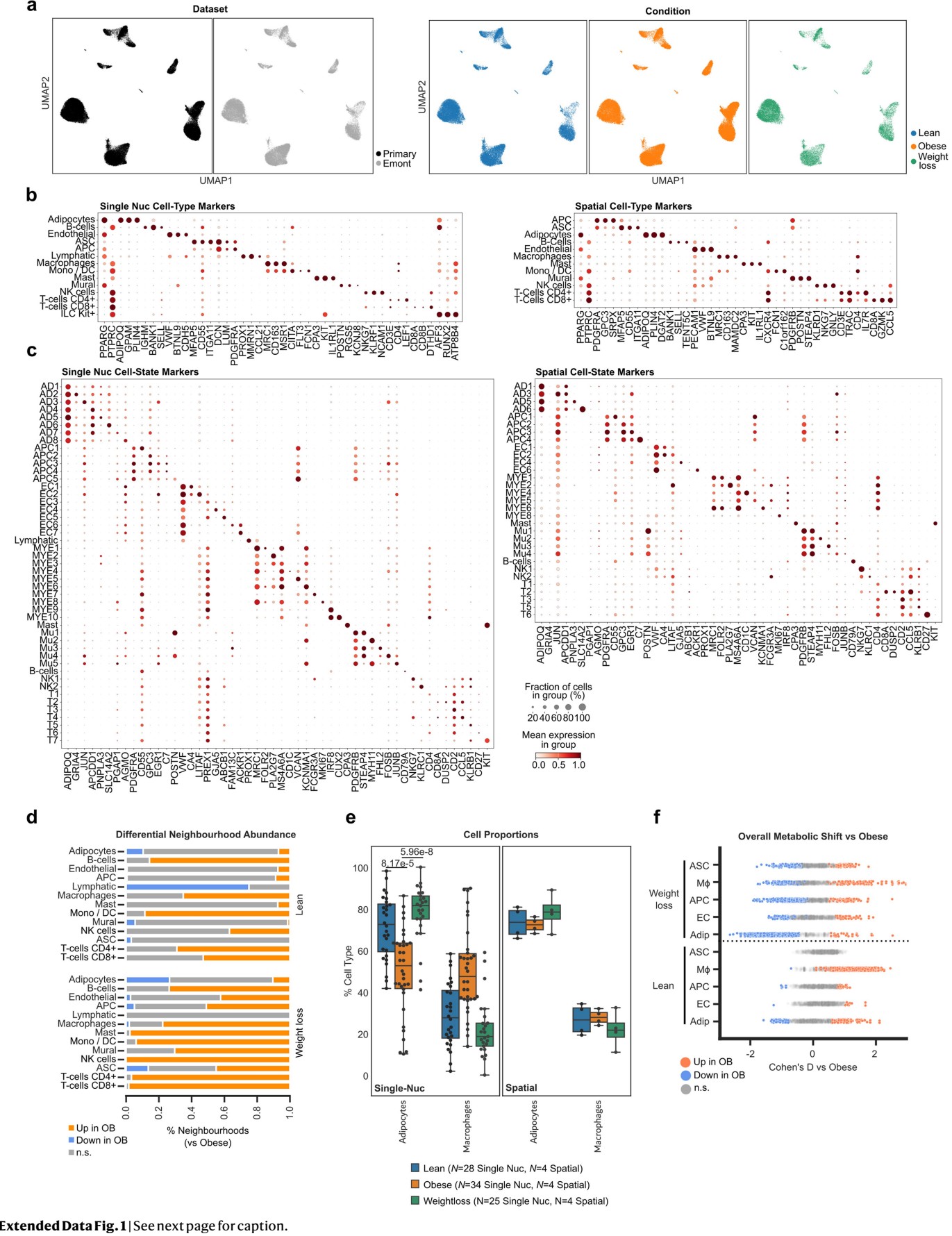

**Extended Data Fig. 1** | See next page for caption.

**Extended Data Fig. 1 | Single nucleus and spatially resolved variations in cell types and states in lean, obese and WL adipose tissues. a**, UMAP embedding of AT cell types across conditions and datasets demonstrating successful integration and cell type conservation. **b**, Cell type marker genes in the single nucleus (Nuc, left) and spatial datasets (right). ASC, adipose stem cells. APC, other adipose progenitor cells. Endothelial, vascular endothelial cells. ILC, innate lymphoid cells. Lymphatic, lymphatic endothelial cells. Mono/DC, monocytes and dendritic cells. **c**, Cell state marker genes in the Nuc (left) and spatial datasets (right). **b**,**c**, Scaled mean expression and proportion (%) of cells expressing marker. **d**, Proportion of cell neighbourhoods exhibiting significant differences in cell abundance between conditions (Spatial FDR < 0.1) for each cell type. Orange obese-high, blue obese-low, grey non-significant (NS). **e**, Proportional changes in adipocytes and macrophages between conditions in single Nuc and spatial datasets. Restricted to these cell types due to limited spatial cohort numbers ($N$ = 4/group) and intra-sample/group heterogeneity in vascular and precursor cell numbers. Boxplot, median IQR min/max. Wilcoxon Paired (OB-WL) and Unpaired (OB-LN), FDR adjusted P value. **f**, Alterations in pathway-wide metabolic flux between conditions in major AT cell types. Cohen's D, coloured at FDR < 0.05, red obese-high, blue obese-low.

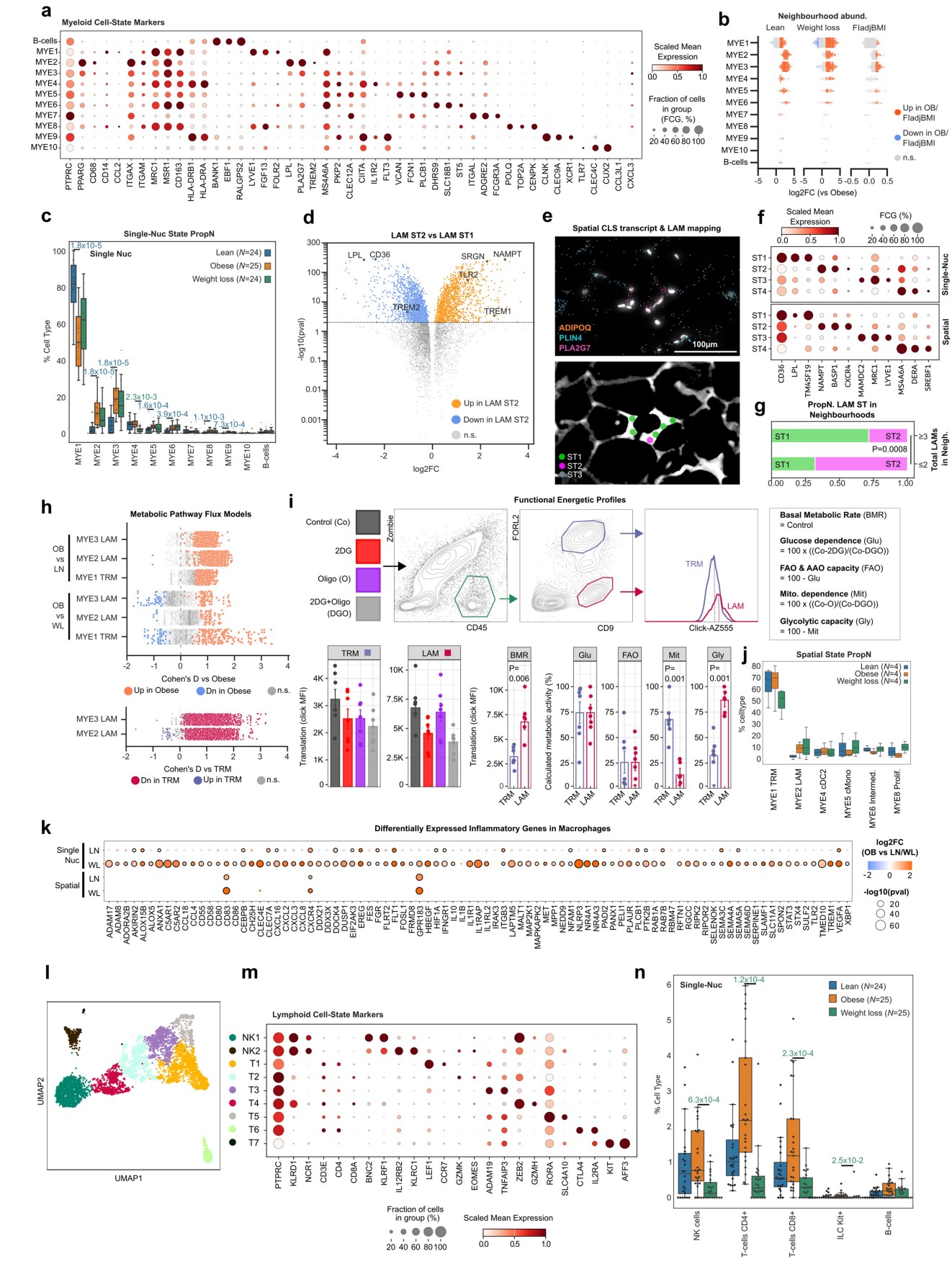

**Extended Data Fig. 2** | See next page for caption.

**Extended Data Fig. 2 | Adipose tissue immune system variations in human weight gain and WL. a**, Myeloid cell marker genes, scaled mean expression, proportion (%) of cells expressing marker. **b**, Beeswarm plots showing significant changes (Spatial FDR < 0.1) in neighbourhood abundance in myeloid cell classes. Lean-obese, obese-WL comparisons, Log2FC between conditions, red obese-high, blue obese-low. Fasting insulin adjusted for body mass index (FI adj BMI), Log2FC per unit change, red FI-high, blue FI-low. **c**, Proportional changes in myeloid cell abundance in single nucleus dataset. Boxplot, median IQR min/max. Lean-Obese unpaired, WL-Obese paired Two-tailed Wilcoxon test. FDR adjusted P values. **d**, Volcano plot of differentially expressed genes in LAM subtypes (ST) 1 (adaptive) and 2 (maladaptive/inflammatory). Two-tailed Wilcoxon unpaired test, FDR < 0.05. Red, LAM ST2-high, Blue LAM ST2-low. **e**, Representative spatial images of a CLS. Top, individual transcripts detected by Xenium for Adipocyte markers (*ADIPOQ* Orange, *PLIN4* Cyan), a LAM marker (*PLA2G7* Magenta), and a nuclei counterstain (DAPI Gray), showing LAMs surround a transcriptionally devoid/dead adipocyte. Bottom, CellTypist "best match" prediction of LAM ST at the CLS. **f**, Shared LAM subtype marker genes, scaled mean expression, proportion (%) of cells expressing marker, in the single nucleus (sNuc, top) and spatial (bottom) datasets. sNuc was used as the training dataset to predict a "best match" in the spatial dataset (CellTypist). **g**, Proportion of LAM ST1 and ST2 in CLS (defined as ≥3 LAMs) or isolated (defined as ≤2 LAMs in Neighbourhood). Two-tailed Chi² test. **h**, Alterations in pathway-wide metabolic flux. Top, between conditions in mature (MYE2) and immature (MYE3) LAM and TRM (MYE1). Red obese-high, blue obese-low. Bottom, between TRM and LAM. Wine-red LAM-high, Yale-blue LAM-low. Cohen's D, coloured at FDR < 0.05. **i**, SCENITH strategy (top) for LAM and TRM metabolic activity from Obese donors (*N* = 7). Cells were gated as single cells (FSC-A-SSC-A, FSC-A-FSC-H, not shown), Zombie-neg (Live/Dead dye) and CD45-pos (pan-immune marker), followed by FOLR2 (TRM marker) and CD9 (LAM marker). HPG-AZ555 Click chemistry was used to measure metabolic activity. Cells were treated with combinations of drugs (Control, 2DG, Oligo, 2DG+Oligo) to assess metabolic profiles, calculated using formulas (right panel). Bottom, Click intensity (MFI) for each drug treatment (left) and calculated metabolic profiles (right). Mean SEM. Paired Student's two-tailed t-test P value. **j**, Proportional changes in myeloid cell abundance in spatial dataset. **k**, Differentially expressed inflammatory cyto/chemokine genes between conditions in single nucleus (Nuc) and spatial datasets. Red obese-high, blue obese-low. Size adjusted -log10 P value, negative binomial mixed effects model. Circled dots represent comparisons with absolute log2FC > 0.3 and adjusted P value < 0.05. **l**, UMAP embedding of lymphoid cell classes, all conditions in single nucleus dataset. **m**, Lymphoid cell marker genes, scaled mean expression, proportion (%) of cells expressing marker. **n**, Global proportional changes (%) in cell abundance in broad lymphoid cell classes across conditions. Boxplot, median IQR min/max. Two-tailed Wilcoxon paired (OB-WL) and unpaired (OB-LN) test. FDR adjusted P values.

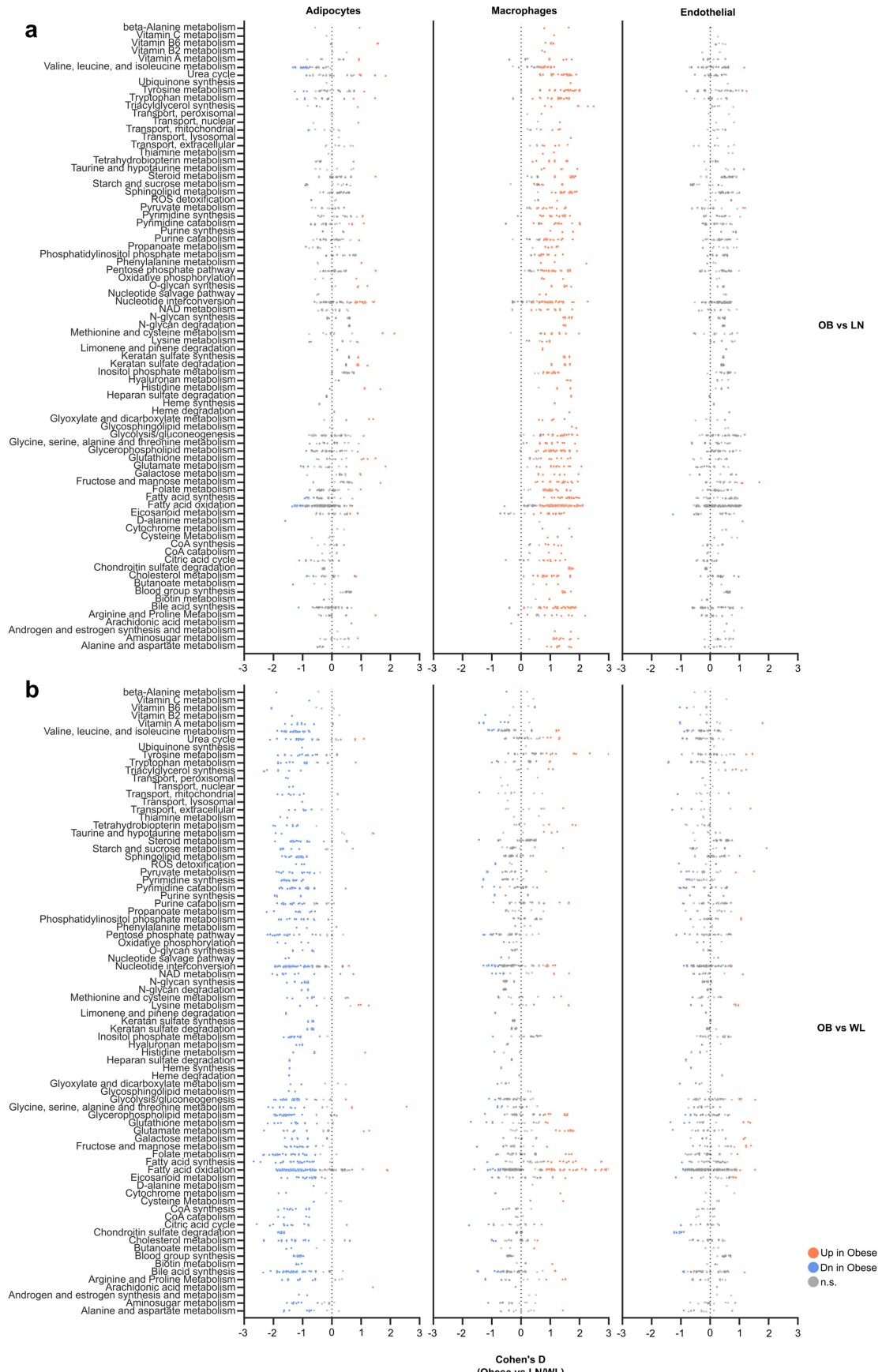

**Extended Data Fig. 3** | See next page for caption.

**Extended Data Fig. 3 | The full spectrum of metabolic pathway flux changes in mature adipocytes and macrophages (83 pathways, 1895 reactions).** All metabolic pathway changes in flux-based analyses in **a**, lean-obese and **b**, obese-WL comparisons. Presented for adipocytes and macrophages in which global metabolic shifts were observed and endothelial cells as a representative other cell type to demonstrate absence of global activation. Cohen's D, coloured at FDR < 0.05, red obese-high, blue obese-low.

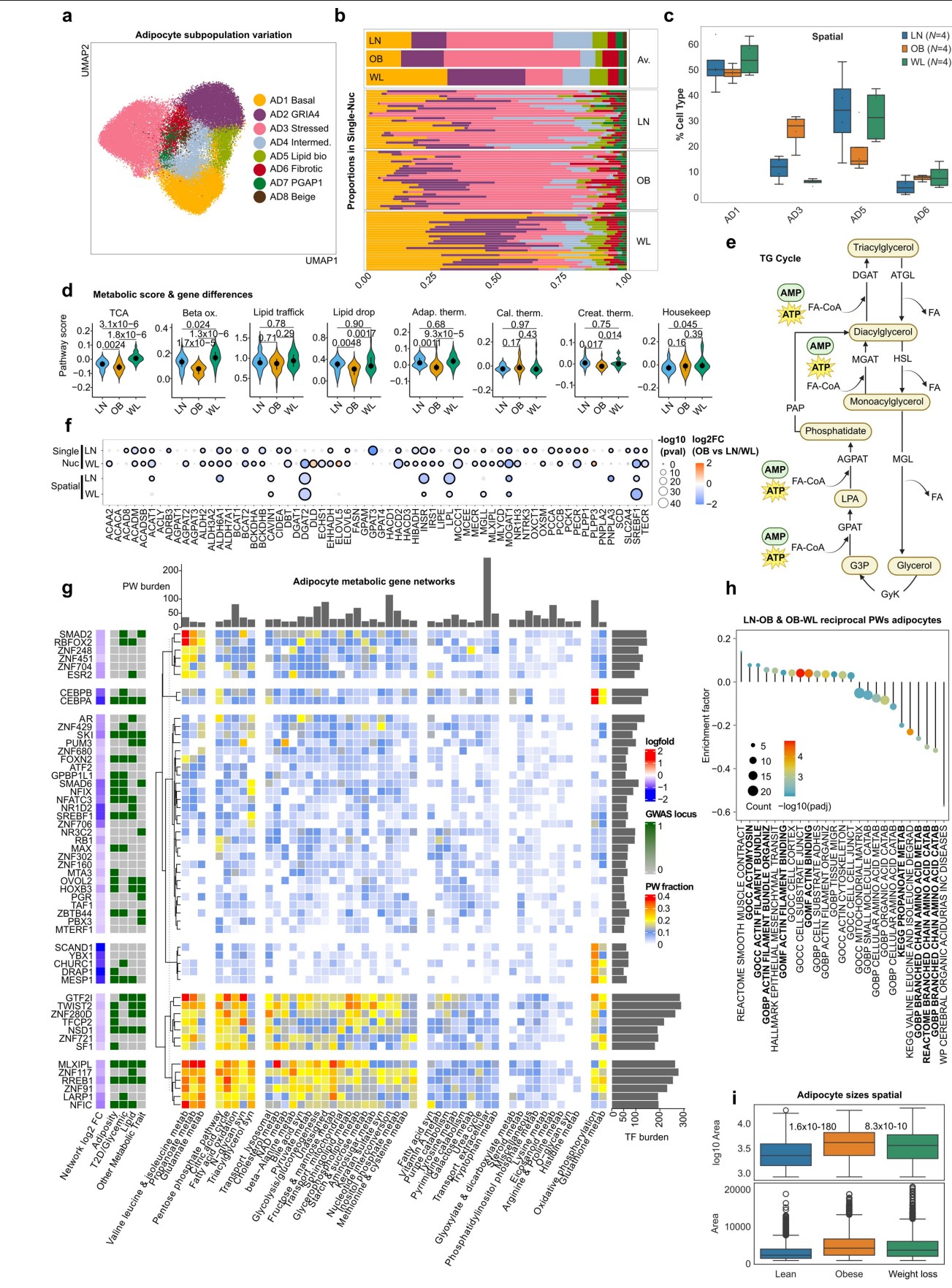

**Extended Data Fig. 4** | See next page for caption.

**Extended Data Fig. 4 | Mature adipocyte molecular heterogeneity and regulation in obesity and WL. a**, UMAP embedding of mature adipocytes, all conditions grouped. **b**, Adipocyte cell state proportions (0 to 1) in the combined cohort, mean (Av.) per group, and for each sample. **c**, Proportional changes in adipocyte cell abundance in spatial datasets. **d**, Scores measuring overall activity in major metabolic pathways in each adipocyte, averaged for each participant (density, median IQR) then compared between conditions. Two-tailed Wilcoxon unpaired (LN-OB, LN-WL) and paired (OB-WL) FDR adjusted P values ($N$ = 24 LN; 25 paired OB/WL donors). **e**, Schematic of the triglyceride (TG) to glycerol cycle, broken down into reaction steps, and annotated by reaction enzyme families. ATP consuming steps are highlighted. Adapted from Sharma et al.[23] **f**, Extended differentially expressed genes between conditions in single nucleus (Nuc) and spatial datasets in adipocytes. Encompassing enzymes in metabolic substrate pathways, including the TG cycle, and upstream regulators. Red obese-high, blue obese-low. Size adjusted -log10 P value, negative binomial mixed effects model. Circled dots represent comparisons with absolute log2FC > 0.3 and adjusted P value < 0.05. **g**, Differential gene regulatory networks between obesity and WL in mature adipocytes, restricted to metabolic pathway genes. TF networks with >50 metabolic genes/network and network P < 0.05 Bonferroni adjusted are shown. Coloured by proportion of all pathway genes in the network. Barplots show sum of genes in pathway (top) and network (right). Left, heatmaps show network (two-tailed Wilcoxon test) log2FC and human GWAS intersection. **h**, Pathways underlying reciprocally differentially expressed genes in lean-obese (LN-OB, log2FC > 0.5, FDR < 0.01) and obese-WL (OB-WL, log2FC > 0.5, P < 0.05, Bonferroni adjusted) comparisons. ORA, hypergeometric distribution, coloured by FDR adjusted -log10 P values, sized by count, enrichment factor is gene ratio/background ratio. **i**, Variations in mature adipocyte sizes (top, log10 Area; bottom, Area) between groups in spatial analyses, and two-tailed Wilcoxon test P value ($N$ = 4850 LN; 3315 OB; 3909 WL; number of segmented adipocytes across 4 donors in each group).

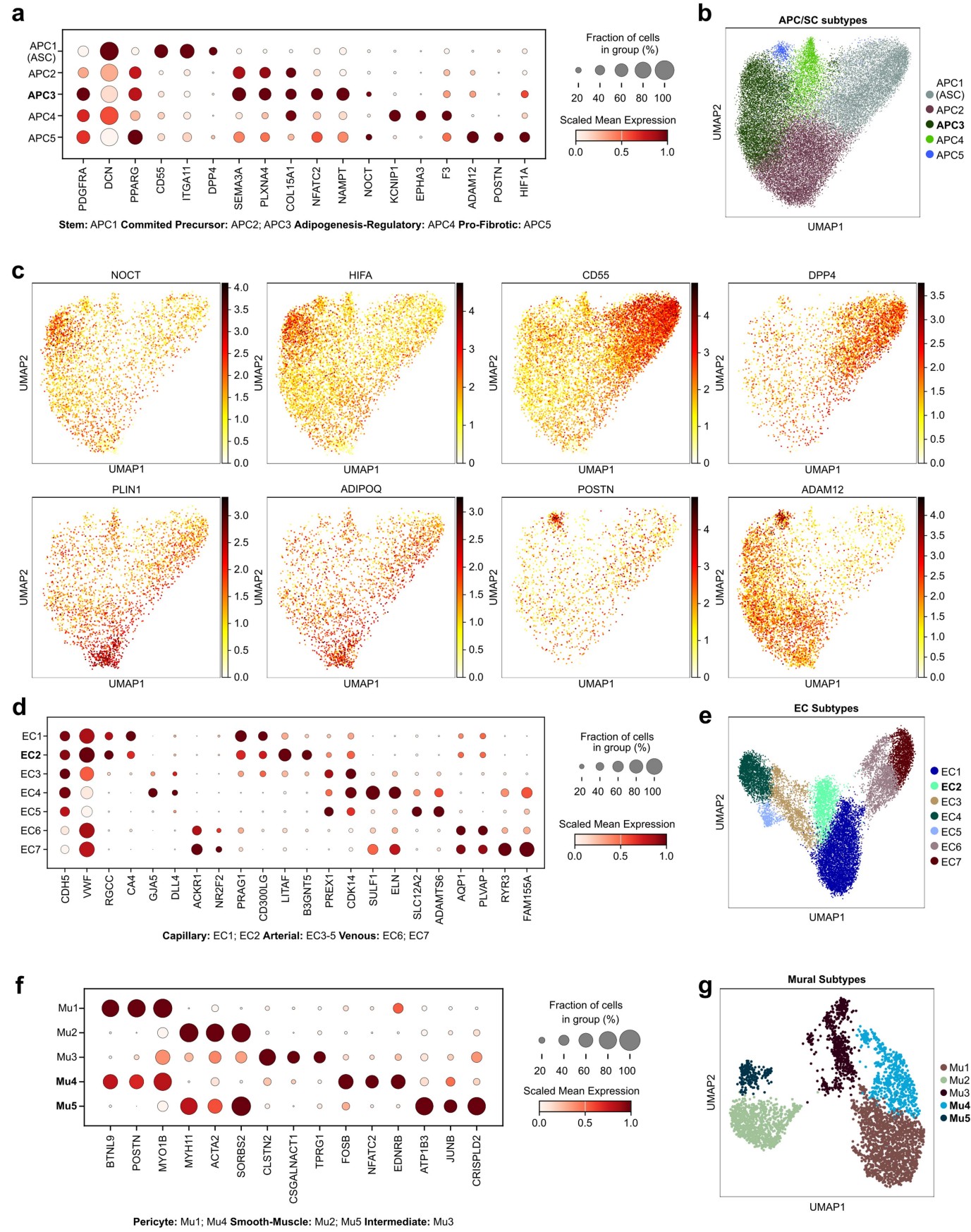

**Extended Data Fig. 5 |** See next page for caption.

**Extended Data Fig. 5 | Precursor and vascular cell phenotypes and adaptations in obesity and WL. a**, Adipocyte precursor (APC) subpopulation marker genes presented as scaled mean expression and proportion (%) of cells expressing marker. UMAP embedding of APCs, all conditions grouped, according to **b**, subtypes and **c**, subtype marker gene expression. **d**, Vascular endothelial cell (EC) subpopulation marker genes presented as scaled mean expression and proportion (%) of cells expressing marker. **e**, UMAP embedding of vascular EC, all conditions grouped. **f**, Mural cell subpopulation marker genes presented as scaled mean expression and proportion (%) of cells expressing marker. **g**, UMAP embedding, all conditions grouped. **a**,**b**,**d**–**g**, Cell states highlighted in bold represent stressed populations.

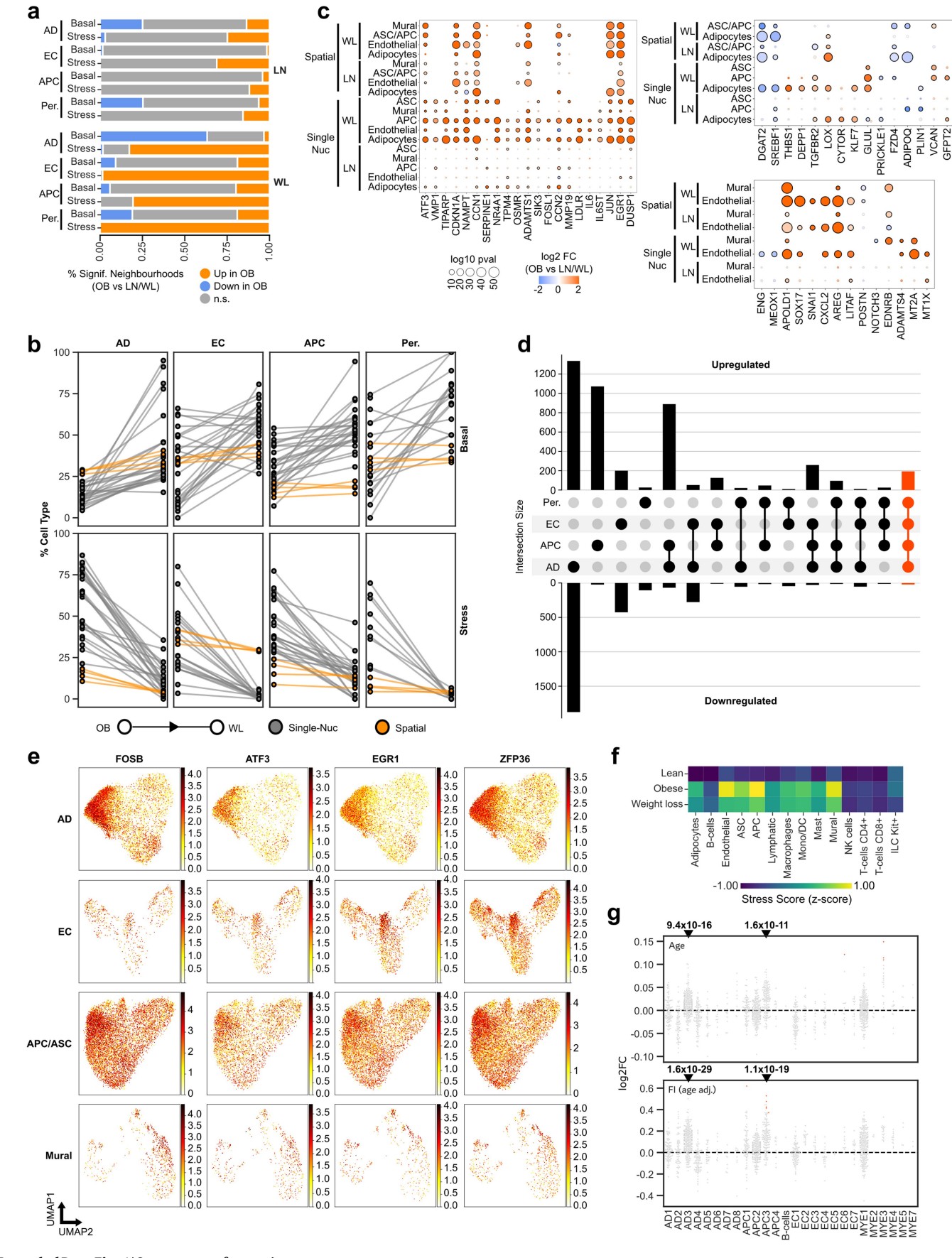

**Extended Data Fig. 6** | See next page for caption.

**Extended Data Fig. 6 | Stressed signatures are conserved across susceptible cell types. a**, Proportions (%) of differentially abundant neighbourhoods (Spatial FDR < 0.1) in lean-obese and obese-WL comparisons among basal and stressed cell states. Orange obese-high, blue obese-low, grey non-significant (n.s.). Mature adipocytes (AD), precursors (APC), endothelial cells (EC) and mural pericytes (Per.). **b**, Pairwise changes in basal and stressed cell proportions in obesity and subsequent WL for each donor in single nucleus (grey) and spatial (orange) datasets (*N* = 25 single nucleus; 4 spatial). **c**, Differential expression between conditions of common stress genes in all vulnerable cell types (left) and homeostatic and maladaptive genes in metabolic and precursor (right, top) and vascular (right, bottom) cell types. Red obese-high, blue obese-low. Size adjusted -log10 P value, negative binomial mixed effects model. Circled dots represent comparisons with absolute log2FC > 0.3 and adjusted P value < 0.05.

**d**, Overlap of differentially expressed genes in stressed states compared to the respective basal state, among vulnerable cell types (Wilcoxon test, FDR < 0.05). Red represents a common set of 188 differentially upregulated and 15 downregulated genes in all represented stressed cell states (Single Nuc. dataset). **e**, UMAP embedding of example stress genes across susceptible cell types. **f**, Stress score based on 188 conserved upregulated genes in stress cell states (AD3, EC2, APC3, Mu4), by cell type and condition, represented as a scaled z-score. **g**, Changes in neighbourhood abundance in lean tissues in association with age (top) and fasting insulin (bottom) adjusted for age (FI age adj., Log2FC per unit change in trait). For AD3 and APC3, two-tailed Binomial sign test P values comparing the observed directions of effect in each cell neighbourhood with the expected null of 0.5.

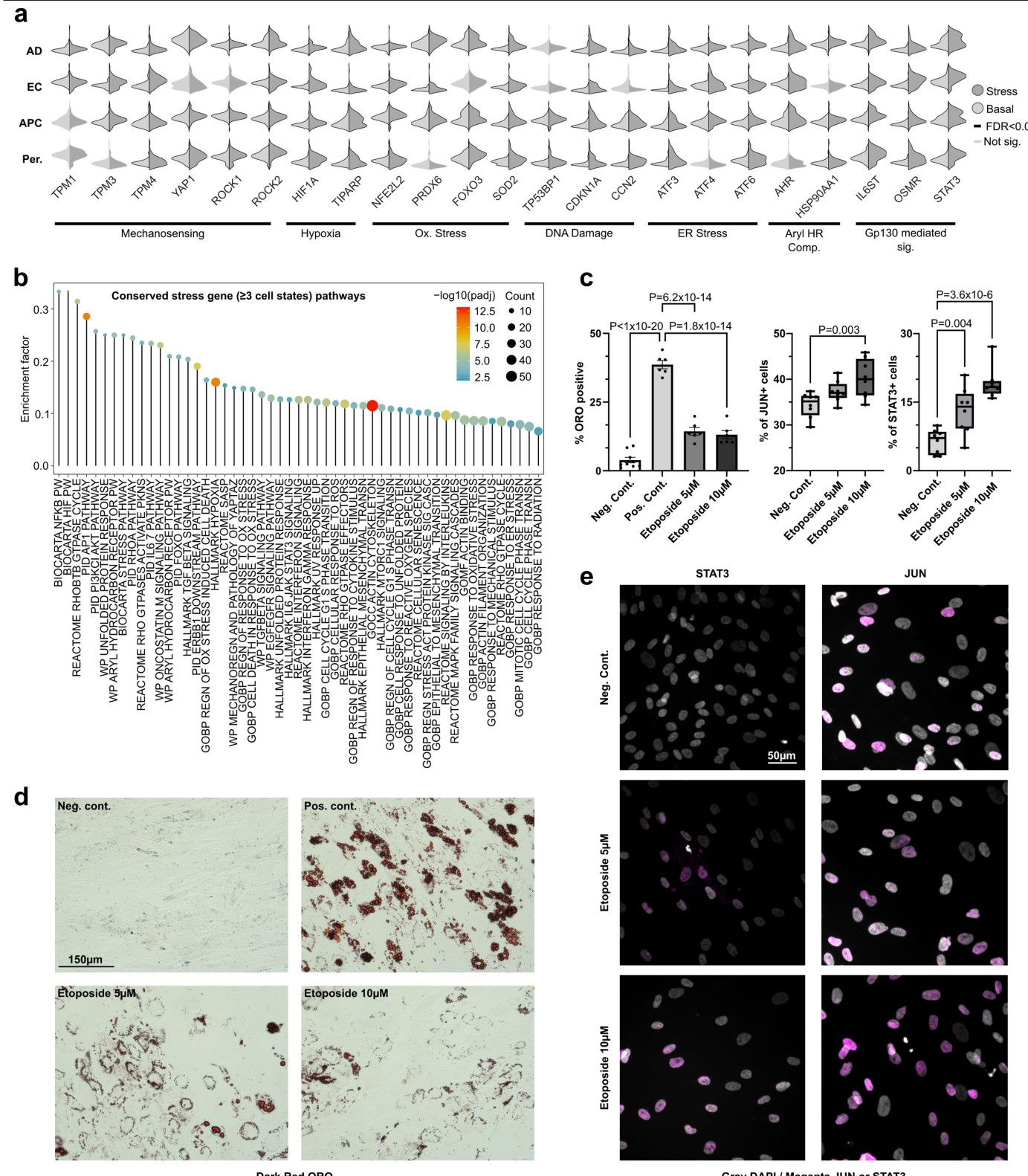

**Extended Data Fig. 7 | Regulation of cellular stress in adipose tissue.**
**a**, Violin plots of stress enriched-genes for example pathways, averaged per sample in stressed (dark grey) and basal (light grey) cell states. Violins outlined in black have Log2FC > 0.1 and FDR < 0.05 (Wilcoxon, Supplementary Table 12). **b**, Selected examples of enriched pathways underlying conserved stress genes (differentially expressed in ≥3 stressed-basal state comparisons). ORA, hypergeometric distribution, coloured by FDR adjusted -log10 P values, sized by count, enrichment factor is gene ratio/background ratio. **c**, In vitro effects of stress induction on: i. human adipocyte differentiation (left, % Oil Red-O

(ORO) positive mature adipocytes) in undifferentiated (Negative Control, *N* = 8), 14-day differentiated (Positive Control, *N* = 6), and 14-day differentiated 5-day Etoposide treated (5 μM and 10 μM, *N* = 6) cells; ii. expression of stress marker proteins (middle/right, % JUN and STAT3 positive nuclei, immunohistochemistry) in undifferentiated control and Etoposide treated cells (*N* = 8 per group). Bar plot, mean SEM. Boxplot, median IQR min/max. **d**, Representative images of ORO accumulation and **e**, JUN and STAT3 protein expression in each experimental group.

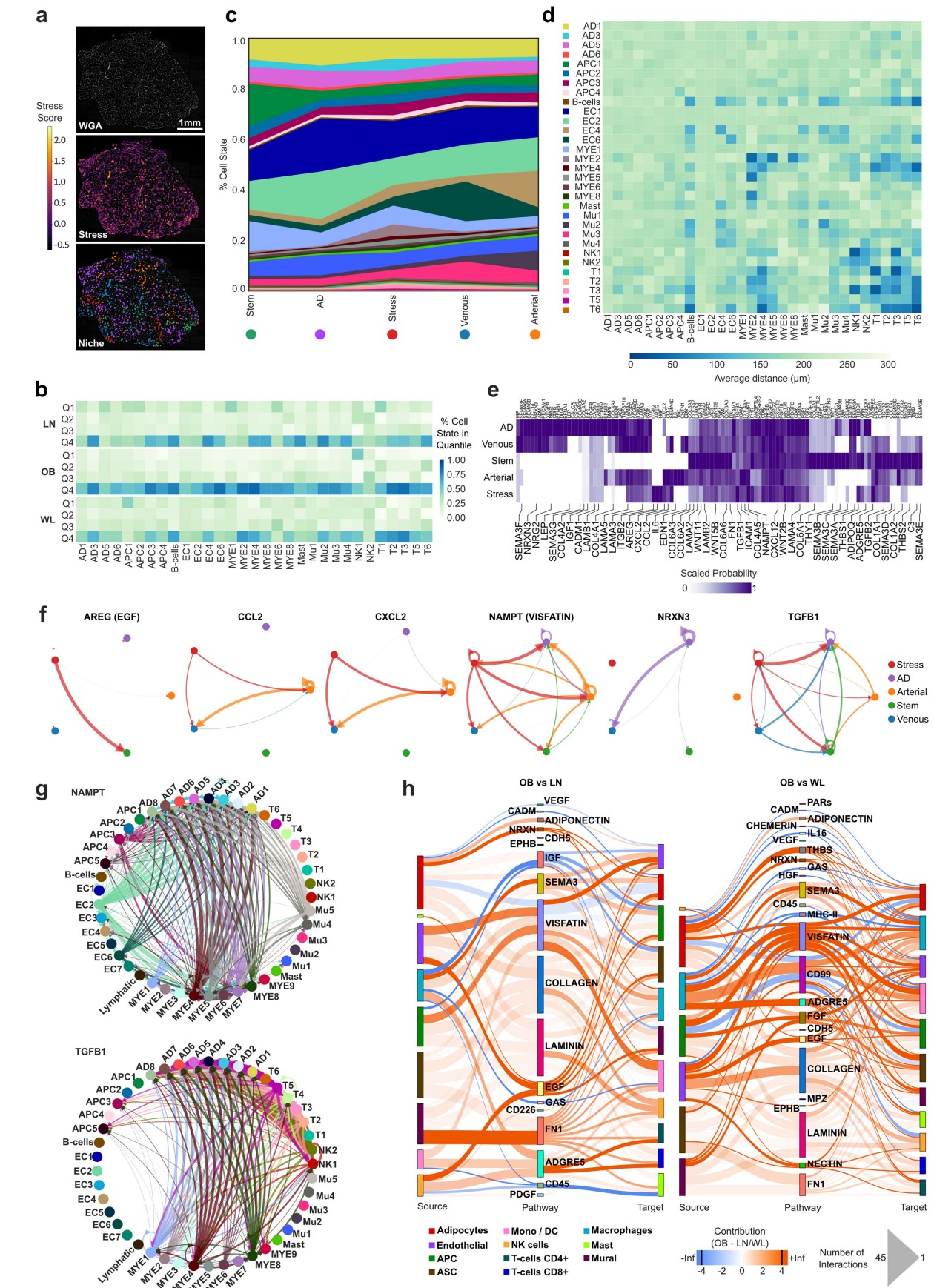

**Extended Data Fig. 8** | See next page for caption.

**Extended Data Fig. 8 | Tissue niche and tissue-wide communication patterns. a**, Representative images of the spatial datasets showing tissue architecture (top, WGA staining), stress scores in 50-μm bins (middle) and tissue niches (bottom). **b**, Proportion of cell states in stress quantiles for each condition (Q1 low stress; Q4 high stress). **c**, Proportions (0 to 1) of cell states in each tissue niche. **d**, average distance within 300 μm between spatial cell states. **e**, Clustermap of imputed scaled average ligand communication probabilities (CellChat) per tissue niche, limited to significant communications. **f**, Imputed CellChat communication between spatial niches for selected ligands. Links represent the scaled mean probability (line thickness) and directions of connectivity. Line colour reflects signal source. All conditions were combined to identify the main niches underlying pathway effects. **g**, CellChat communication between single nucleus cell states for *NAMPT* (Visfatin, top) and *TGFB1* (bottom). Links represent the scaled mean probability (line thickness) and directions of connectivity. Line colour reflects signal source. All conditions were combined to identify the main cell states underlying pathway effects. Lower probability interactions for *NAMPT* were removed to improve visualisation. **h**, Sankey plots showing differential signalling pathways between source and target cells in lean-obese (left) and obese-WL comparisons (right). Source and target cells and pathways sized by overall number of interactions. Connection size represents number of cell type interactions for each pathway and colour relative flow (red obese-high, blue obese-low).

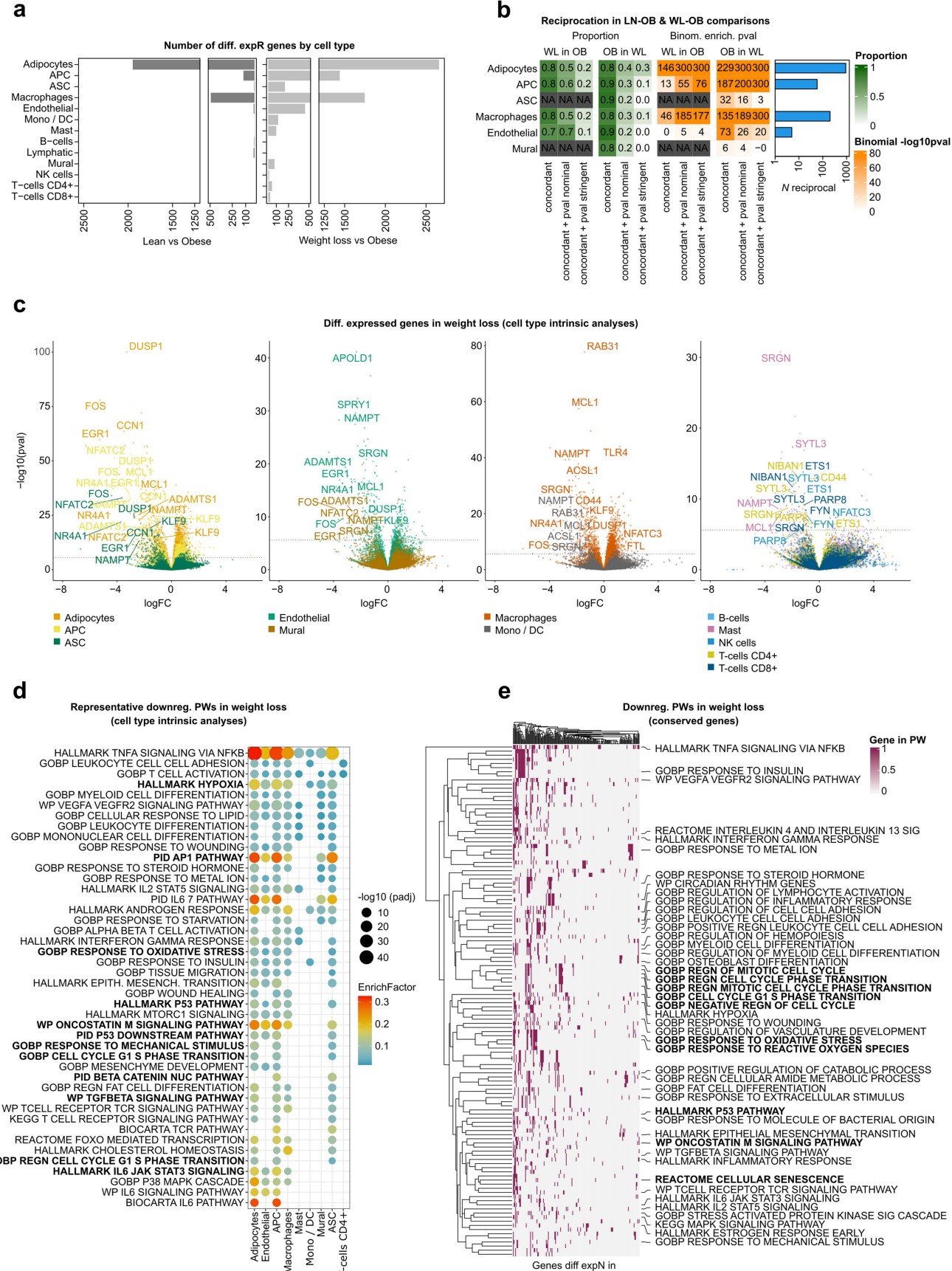

**Extended Data Fig. 9 | See next page for caption.**

**Extended Data Fig. 9 | Systematic differential gene expression and pathway analyses in human obesity and WL across the full spectrum of adipose tissue cell types. a**, Number of differentially expressed genes in major AT cell types in lean-obese (FDR < 0.01) and obese-WL (P < 0.05 Bonferroni adjusted) comparisons. **b**, Heatmaps showing proportion of significant genes (0–1, green) in the primary comparison that had i. concordant directions of effect (concordant), ii. concordant and significant at P < 0.05 (concordant + pval nominal) or iii. concordant and robustly significant (at FDR < 0.01 lean-obese or P < 0.05 Bonferroni adjusted obese-WL, concordant + pval stringent) in the alternative comparison, as well as the associated binomial test -log10 P value (orange). Barplots depict total number of robustly significant reciprocal genes. **c**, Volcano plots of differentially expressed genes associated with WL across

AT cell types. Log2FC positive obese-high and association -log10 P value. Horizontal line, Bonferroni adjusted significance threshold. Selected representative genes annotated. **d**, Pathway analysis of genes downregulated by WL (FC > 0.5, P < 0.05 Bonferroni adjusted) in cell type intrinsic analyses. Sized by FDR adjusted -log10 P values (ORA, hypergeometric distribution) and coloured by enrichment factor (gene ratio/background ratio). Shown 44 representative of 660 total pathways at FDR < 0.01. **e**, Pathway analysis of conserved genes, downregulated by weight-loss in ≥3 distinct cell types (FC > 0.5, P < 0.05 Bonferroni adjusted), clustered by gene (*N* = 213) and pathway (*N* = 304, ORA, hypergeometric distribution, FDR < 0.01). All differential expression analyses applied two-tailed neg. binom. mixed effect models.

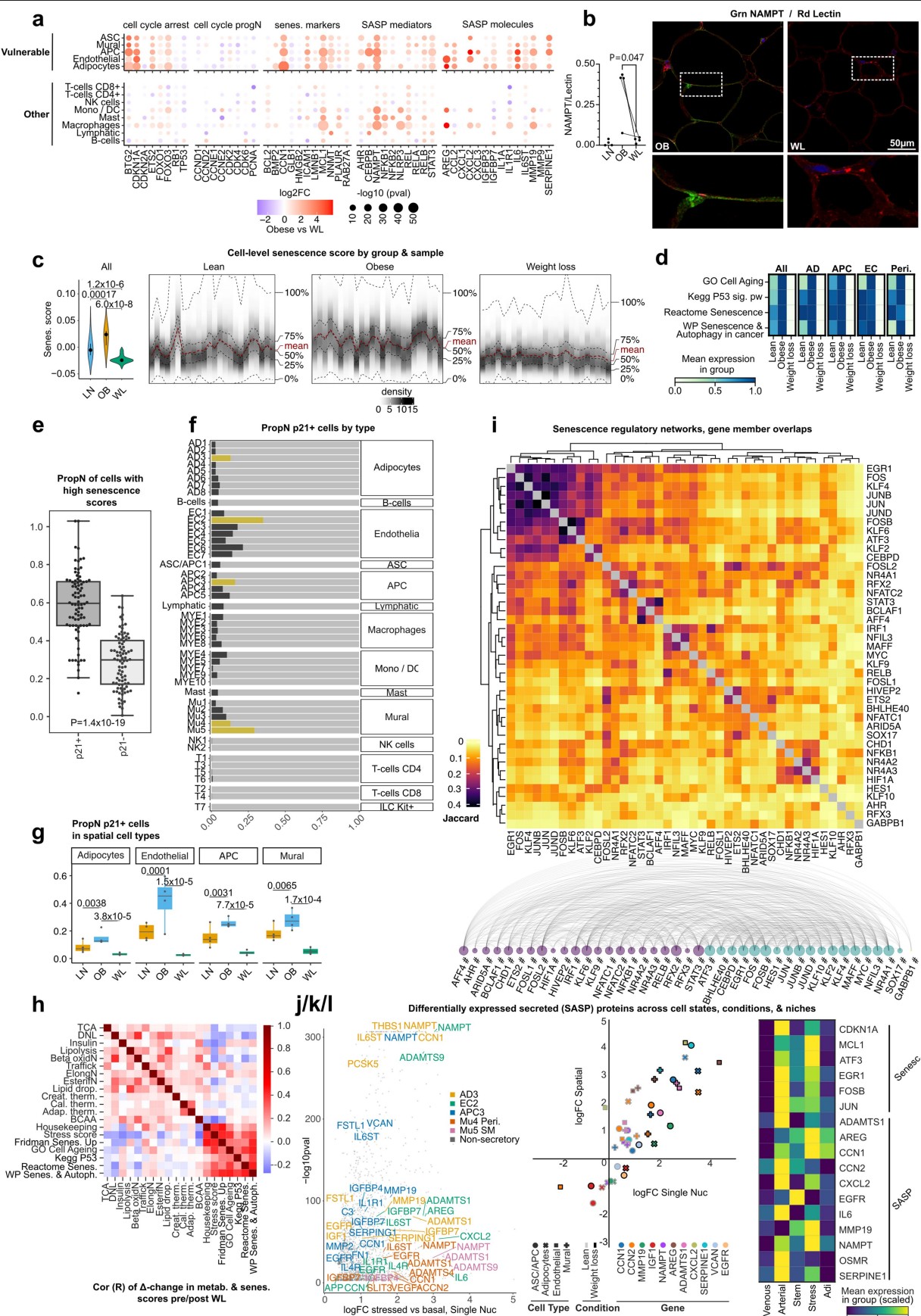

**Extended Data Fig. 10** | See next page for caption.

**Extended Data Fig. 10 | Senescence vulnerability and regulatory pathways in human adipose tissue cell types and the mitigating effects of WL.**
**a**, Differences in expression of cell cycle and senescence marker genes in WL, separated into vulnerable and other cell types. Coloured by log2FC, sized by -log10 P value, neg. binom. mixed effect models. **b**, Immuno-fluorescence of NAMPT protein expression ($N = 4$ samples/group), scaled to connective tissue marker Lectin, paired student's two-tailed t-test (left). Representative images of an obese and WL pair, scale bar 50 μm resolution, Grn NAMPT, Rd Lectin, blue DAPI nuclei. **c**, Left (All), tissue-wide senescence score (Oncogene induced), averaged across every cell for each participant (density, median IQR), then compared between conditions. Two-tailed Wilcoxon unpaired (LN-OB, LN-WL) and paired (OB-WL) P values. Right, density heatmaps of cell-level senescence scores (Oncogene induced) encompassing all cell types for each individual sample separated into Lean, Obese, WL groups, single nucleus datasets ($N = 24$ LN; 25 paired OB/WL donors). **d**, Other unbiased senescence score heatmaps across groups and vulnerable cell types. **e**, Proportion of p21 negative (−) and p21 positive (+) cells with high senescence scores (defined by score higher than median in ≥3 of 4 distinct senescence scores). Proportion presented for each sample ($N = 87$). Two-tailed Wilcoxon unpaired test. **f**, Proportion of p21 positive cells (range 0–1) in each cell state grouped by cell type. Stressed cell states coloured yellow, other cell states coloured dark grey. **g**, Mean proportions of p21 (range 0–1) positive cells in each sample across conditions in spatial datasets ($N = 4$/group). Boxplot, median IQR min/max. Two-tailed Wilcoxon unpaired (LN-OB) and paired (OB-WL) FDR adjusted P values. Separated into vulnerable cell types. **h**, Heatmap showing pairwise correlation (Pearson, R) between delta (Δ) changes in pathway scores before and after WL in paired samples. Pathway scores were calculated from the mean pathways score in mature adipocytes within each paired sample. **i**, Gene regulatory networks upregulated in stressed, senescent cells (scaled log2FC > 0.4 compared to all other cell states in cell type, and in ≥3 stressed cell states) and coloured by number of shared genes in the network (Jaccard index, top). Interactions between TFs within the network (bottom), sized by number of interactions with other TFs, connected by forward interactions, # annotates self-interaction, coloured by Walktrap community. **j**, Expression of secretory proteins from the Human Protein Atlas (HPA) in stressed compared to basal cell states among vulnerable cell types. Wilcoxon test, Log2FC (positive, stress-high) coloured by cell type, grey if non-secretory or non-significant ($P > 0.05$ Bonferroni adjusted). *AREG* which is not in the HPA was included as a well-established secreted protein[67]. **k**, Scatter plot of 11 predicted SASP proteins present in both single nucleus and spatial datasets according to dataset log2FC in lean-obese and obese-WL comparisons (obese-high). Border coloured by comparison, fill coloured by SASP gene, shape by cell type. **l**, Senescence and SASP gene expression (imputed) in tissue niches, represented as a scaled z-score.

# Extended Data Table 1 | Patient characteristics in the primary cohort and Emont dataset

| | Primary | | | Primary + Spatial | | | Lean-Obese pval | | Obese-WL pval | | Emont | |
|---|---|---|---|---|---|---|---|---|---|---|---|---|
| | Lean | Obese | Weight loss | Lean | Obese | Weight loss | Primary | All | Primary | All | Lean | Obese |
| *N* | 24 | 25 | 25 | 28 | 29 | 29 | NA | NA | NA | NA | 4 | 9 |
| Age | 47.2 (12.7) | 43.4 (10.9) | NA | 46 (13.2) | 46 (13.2) | NA | 2.6E-01 | 6.3E-01 | NA | NA | 58.2 (14.9) | 44.9 (11.3) |
| Sex (% Female) | 63% | 72% | NA | 68% | 76% | NA | 6.9E-01 | 7.1E-01 | NA | NA | 100% | 100% |
| Ethnicity (% EW) | 88% | 88% | NA | 86% | 79% | NA | 1.0E+00 | 7.7E-01 | NA | NA | 100% # | 78% # |
| Weight (kg) | 68.3 (11.5) | 127.9 (26.5) | 99.9 (22.2) | 67.4 (11.1) | 125.9 (25.8) | 98.6 (22.6) | 8.1E-12 | 1.3E-13 | 5.1E-16 | 1.3E-17 | NA | NA |
| BMI (kg/m2) | 23.7 (2) | 45.2 (6.7) | 35.2 (5.5) | 23.4 (2.1) | 44.8 (6.3) | 35.1 (5.5) | 2.8E-15 | 9.5E-11 | 2.8E-16 | 1.0E-17 | 23.9 (1.3) | 40.5 (6.3) |
| Fasting Insulin (mIU/L) | 4.9 (3.1) | 11.8 (3.5) | 7.6 (3.3) | 5 (2.9) | 11.8 (3.5) | 7.9 (3.5) | 2.3E-09 | 4.6E-10 | 1.0E-06 | 1.0E-06 | NA | NA |
| Fasting Glucose (mmol/L) | 4.9 (0.9) | 5.1 (0.5) | 4.5 (0.3) | 4.7 (1.2) | 5.1 (0.6) | 4.6 (0.5) | 7.5E-01 | 1.5E-01 | 1.9E-05 | 2.0E-05 | NA | NA |
| HbA1c (%) | 5.4 (0.5) | 5.5 (0.4) | 5.3 (0.3) | 5.2 (1.1) | 5.6 (0.4) | 5.3 (0.3) | 1.7E-01 | 1.0E-01 | 5.9E-05 | 1.3E-05 | NA | NA |
| HOMAIR | 1.1 (1.0) | 2.7 (0.9) | 1.5 (0.6) | 1.1 (0.9) | 2.7 (0.9) | 1.6 (0.8) | 5.9E-07 | 8.0E-08 | 4.5E-08 | 3.3E-08 | NA | NA |
| HOMAB | 90.7 (96.6) | 180.1 (109.5) | 193.2 (153.5) | 89.4 (90.7) | 173.2 (108.2) | 185 (147) | 4.3E-03 | 3.6E-03 | 7.0E-01 | 6.9E-01 | NA | NA |
| CRP (mg/dL) | 3.4 (7.1) | 7.4 (6.2) | 7.7 (14.1) | 3.7 (7) | 7.5 (6) | 7.3 (13.3) | 4.3E-02 | 2.9E-02 | 7.7E-01 | 9.7E-01 | NA | NA |
| Systolic BP (mmHg) | 123.1 (18.8) | 133.4 (14.2) | 120.1 (14) | 118.2 (29.8) | 134.7 (15.1) | 119 (14.9) | 4.7E-02 | 1.7E-02 | 4.3E-04 | 5.6E-05 | NA | NA |
| Diastolic BP (mmHg) | 74.8 (12.5) | 79.5 (10.1) | 78.6 (10.2) | 72.1 (18.8) | 79.2 (9.6) | 78 (10.1) | 1.8E-01 | 9.7E-02 | 6.3E-01 | 4.3E-01 | NA | NA |
| LDL cholesterol (mmol/L) | 2.9 (0.9) | 2.9 (0.7) | 3.0 (1.0) | 2.7 (1.0) | 2.9 (0.7) | 2.9 (1.0) | 9.8E-01 | 1.0E-01 | 7.0E-01 | 8.1E-01 | NA | NA |
| HDL cholesterol (mmol/L) | 1.6 (0.4) | 1.1 (0.2) | 1.4 (0.3) | 1.5 (0.5) | 1.1 (0.2) | 1.4 (0.3) | 3.8E-06 | 2.7E-04 | 4.1E-06 | 4.8E-07 | NA | NA |
| Triglycerides (mmol/L) | 1.0 (0.5) | 1.6 (0.8) | 1.4 (0.6) | 0.9 (0.5) | 1.5 (0.8) | 1.3 (0.6) | 4.4E-03 | 4.7E-03 | 3.6E-01 | 3.5E-01 | NA | NA |

Data presented as Mean (Standard Deviation) for continuous variables, and as percentage (%) for categorical variables. Continuous variables: Lean-Obese two-tailed unpaired Student's t-test, Obese-Weight loss (WL) two-tailed paired Student's t-test. Categorical variables: two-tailed Chi-Square test. EW: European White. # represents % Caucasian in the Emont dataset.

# Reporting Summary

## Statistics

For all statistical analyses, confirm that the following items are present in the figure legend, table legend, main text, or Methods section.

| n/a | Confirmed |
|---|---|
| ☐ | ☒ The exact sample size (*n*) for each experimental group/condition, given as a discrete number and unit of measurement |
| ☐ | ☒ A statement on whether measurements were taken from distinct samples or whether the same sample was measured repeatedly |
| ☐ | ☒ The statistical test(s) used AND whether they are one- or two-sided<br>*Only common tests should be described solely by name; describe more complex techniques in the Methods section.* |
| ☐ | ☒ A description of all covariates tested |
| ☐ | ☒ A description of any assumptions or corrections, such as tests of normality and adjustment for multiple comparisons |
| ☐ | ☒ A full description of the statistical parameters including central tendency (e.g. means) or other basic estimates (e.g. regression coefficient) AND variation (e.g. standard deviation) or associated estimates of uncertainty (e.g. confidence intervals) |
| ☐ | ☒ For null hypothesis testing, the test statistic (e.g. *F*, *t*, *r*) with confidence intervals, effect sizes, degrees of freedom and *P* value noted<br>*Give P values as exact values whenever suitable.* |
| ☒ | ☐ For Bayesian analysis, information on the choice of priors and Markov chain Monte Carlo settings |
| ☐ | ☒ For hierarchical and complex designs, identification of the appropriate level for tests and full reporting of outcomes |
| ☐ | ☒ Estimates of effect sizes (e.g. Cohen's *d*, Pearson's *r*), indicating how they were calculated |

*Our web collection on statistics for biologists contains articles on many of the points above.*

## Software and code

Policy information about availability of computer code

| | |
|---|---|
| Data collection | Raw single nucleus sequencing data: CellRanger (v5.0.1), bcl2fastq (v2.20.0)<br>Raw spatial transcriptomic data: Xenium Analyser (v1.7.1.0)<br>Human GWAS data: SHAPEIT (v2.r900), IMPUTE2 (v2.3.2) |
| Data analysis | Data preprocessing and quality control: CellBender (v0.2.0), Seurat (v4.3.0), SeuratObject (v4.1.3), Scanpy(v1.9.3)<br>Sample assignment from genotpye: Vireo (v0.5.6), cellsnp-lite (v1.2.2)<br>Integration: Harmonypy (v0.0.6), BBKNN (v1.6.0)<br>Spatial Segmentation: FiJi (v1.54f)<br>Exploratory data analysis: Nebula (v1.4.1), CellChat (v1.6.1), Compass (v0.9.10.2), pySenic (v0.12.1), MiloR (v1.10.0), ENVI (v0.3.6), ClusterProfiler (v4.2.2)<br>Data analysis pipelines used in this work can be obtained from: https://github.com/WRScottImperial/WAT_single_cell_analysis_Nature_2024 |

For manuscripts utilizing custom algorithms or software that are central to the research but not yet described in published literature, software must be made available to editors and reviewers. We strongly encourage code deposition in a community repository (e.g. GitHub). See the Nature Portfolio guidelines for submitting code & software for further information.

## Data

Policy information about <u>availability of data</u>

All manuscripts must include a <u>data availability statement</u>. This statement should provide the following information, where applicable:
- Accession codes, unique identifiers, or web links for publicly available datasets
- A description of any restrictions on data availability
- For clinical datasets or third party data, please ensure that the statement adheres to our <u>policy</u>

Raw single cell and spatial transcriptomic datasets are deposited on Gene Expression Omnibus (accessions GSE295708 and GSE295862 respectively). Integrated single-nucleus and Xenium objects, together with auxiliary files, can be found at the Single Cell Portal (accessions SCP3116 and SCP3117 respectively). The following publicly available datasets were used in this study: human AT single nucleus transcriptomic data (Single Cell Portal, SCP1376 and GEO accession GSE176171); human reference genome (cf.10xgenomics.com/refdata-gex-GRCh38-2020-A.tar.gz); Molecular Signatures Database (MsigDB, https://www.gsea-msigdb.org/gsea/msigdb/); secreted proteins in the Human Protein Atlas (https://www.proteinatlas.org/humanproteome/tissue/secretome); motifs for SCENIC (resources.aertslab.org/cistarget/databases/homo_sapiens/hg38/refseq_r80/tc_v1/gene_based/); human GWAS (https://www.ebi.ac.uk/gwas/).

## Research involving human participants, their data, or biological material

Policy information about studies with <u>human participants or human data</u>. See also policy information about <u>sex, gender (identity/presentation), and sexual orientation</u> and <u>race, ethnicity and racism</u>.

| | |
|---|---|
| Reporting on sex and gender | Adult males and females were included in the study design. Primary and spatial cohort: N=Female 41, N=Male 16. Numbers or proportions of each sex are provided in the design figures, participant characteristics tables and methods. Sex specific analyses were not carried out because of the limited sample size, and insufficient power to detect and therefore report sex specific effects. |
| Reporting on race, ethnicity, or other socially relevant groupings | Ethnicity is reported for all study participants within participant meta-data files. Ethnicity is based on self reported NHS ethnicity categories, which were then grouped into one of: European White; South Asian; and Black, Black British, Caribbean or African. Confounding was controlled for by selecting obese cases and lean controls that were well matched for age, sex and ethnicity. These biological and other technical factors were also included as covariates in regression based analyses. Paired analyses were used to control for participant level factors before and after weight loss. |
| Population characteristics | Detailed population characteristics are provided in extended data table 1. Obese participants had BMI>35kg/m2, lean pariticipants BMI<25kg/m2. Groups were well matched for age (within 5yrs), sex and ethnicity. People with systemic illnesses not related to obesity were excluded, as were people with treated type 2 diabetes due to the potential effects of medications on adipose tissues. |
| Recruitment | Prospective participants were recurited sequentially from bariatric and other general surgery preassessment clinics. Study participants were then selected from the larger cohort to enable groups to be well matched for age sex and ethnicity. People with diabetes taking medication that might impact adipose tissue function were excluded. This may skew the obese study group towards less severe pathobiology but it is unlikely to impact results from the between group comparisions. |
| Ethics oversight | All participants gave informed consent. The study was approved by the London – City Road and Hampstead Research Ethics Committee, United Kingdom (reference 13/LO/0477). Human tissue validation also used samples from the Imperial College Healthcare Tissue Bank (ICHTB) – approved by Wales REC3 to release human material for research (reference 17/WA/0161). |

Note that full information on the approval of the study protocol must also be provided in the manuscript.

# Field-specific reporting

Please select the one below that is the best fit for your research. If you are not sure, read the appropriate sections before making your selection.

☒ Life sciences          ☐ Behavioural & social sciences          ☐ Ecological, evolutionary & environmental sciences

For a reference copy of the document with all sections, see <u>nature.com/documents/nr-reporting-summary-flat.pdf</u>

# Life sciences study design

All studies must disclose on these points even when the disclosure is negative.

| | |
|---|---|
| Sample size | Prospective sample sizes calculations were performed using the hierarchicell package and dispersion estimates from a pilot cohort of N=6 samples. Extreme trait (N=24 lean, N= 25 obese) and paired longitudinal (N=25 marked weight loss) sampling were combined with cell level analyses (>100K cells) to provide sufficient discovery power at the cell type and common/infrequent cell state levels. |
| Data exclusions | Two lean samples were excluded at integration because of very low cell numbers (technical failure) and very high lymphocyte counts (suggesting lymph node content in adipose tissue biopsy) respectively. |
| Replication | Each study participant/timepoint was considered an experimental replicate (N=24 lean, N=25 obese, N=25 weight loss). Individual samples were processed in pools (4-5 samples/pool; total of 6 pools/group). Sample pools for each experimental group were processed through to |

sequencing in lean-obese-weight loss trios across 4 batches, to minimise between group batch effects. Single cell nucleus results were systematically replicated in independent samples (N=4/group) using a distinct spatial Xenium technology.

**Randomization**
Lean, obese and weight loss samples were processed in triplets, comprising the obese-weight loss pair and designated control, in random order to minimise batch effects.

**Blinding**
Unbiased genomic analyses were carried out unblinded. Blinding was not undertaken because single cell studies require iterative analysis, interpretaiton, and contextualization within the existing knowledge base. Tissue histological analyses were unblinded because manifest differences in human adipocyte sizes between conditions made blinding impossible. Cell culture validation experiments were quantified using unbiased imaging methods and did not thus require blinding.

# Reporting for specific materials, systems and methods

We require information from authors about some types of materials, experimental systems and methods used in many studies. Here, indicate whether each material, system or method listed is relevant to your study. If you are not sure if a list item applies to your research, read the appropriate section before selecting a response.

## Materials & experimental systems

| n/a | Involved in the study |
|-----|----------------------|
| ☐ | ☒ Antibodies |
| ☐ | ☒ Eukaryotic cell lines |
| ☒ | ☐ Palaeontology and archaeology |
| ☒ | ☐ Animals and other organisms |
| ☒ | ☐ Clinical data |
| ☒ | ☐ Dual use research of concern |
| ☒ | ☐ Plants |

## Methods

| n/a | Involved in the study |
|-----|----------------------|
| ☒ | ☐ ChIP-seq |
| ☐ | ☒ Flow cytometry |
| ☒ | ☐ MRI-based neuroimaging |

## Antibodies

**Antibodies used**

Immunohistochemistry:

anti-p21 Waf1/Cip1 (1:50, Cell Signalling, catalogue no. 2947, clone 12D1)
anti-rabbit IgG conjugated with polymeric horseradish peroxidase linker (25μg/ml, Leica Bond Polymer Refine Detection, DS9800).

Immunofluorescence:

anti-NAMPT (1:200, Affinity Biosciences #DF6059)
anti-TREM2 (clone D8I4C, 1:400, Cell Signalling #91068)
anti-TLR2 (clone TL2.1, 1:400, Invitrogen #14-9922-82)
anti-Stat3 (clone 124H6, 1:500, Cell Signalling # 9139S)
anti-c Jun (clone 60A8, 1:500, Cell Signalling # 9165S)
Goat anti-Rabbit Alexa Fluor 488 (1:200, Invitrogen #A-11034)
Donkey anti-Rabbit Alexa Fluor Plus 488 (1:250, Invitrogen # A32790)
Goat anti-Mouse Alexa Fluor Plus 647 (1:250, Invitrogen # A32728)

FACS
anti-human CD45 antibody conjugated to FITC (1:20, BioLegend # 304006, clone HI30)
anti-human CD9 antibody conjugated to APC-Fire (1:20, BioLegend # 312114, clone H19α)
anti-human FOLR2 antibody conjugated to APC (1:20, BioLegend # 391705, clone 94b/FOLR2 )

**Validation**

Histology antibodies:

The anti-p21 Waf1/Cip1 (clone 12D1) antibody has been validated by Western blot analysis of p21 Waf1/Cip1 knockout HeLa cells. Its specificity on human tissue has been previously demonstrated by Zhu et al. (2019) using immunohistochemistry (IHC). In our study, we utilised a human breast cancer sample as a positive control to assess the specificity and functionality of the antibody dilution.

The anti-NAMPT (DF6059) antibody is specific to human, rat, and mouse tissues. According to the manufacturer, this antibody has been validated by IHC on rat adipose tissue and human esophageal cancer. Further specificity validation was conducted by Tang et al. (2020) using Western blot analysis with viral-induced NAMPT overexpression. In our study, we included a staining protocol using only the secondary antibody to evaluate potential non-specific secondary antibody binding in adipose tissue.

The anti-TREM2 (D8I4C) antibody has been validated by the manufacturer by western blot analysis of extracts from 293T cells transfected with a construct expressing Myc-tagged full-length human TREM2 protein or a mock construct.

The anti-TLR2 (TL2.1) has been validated by da Rocha et al. (2021) where its specificity its specificity to dectec TLR2 postive monocytes was shown.

The anti-Stat3 (124H6) antibody has been validated on several human cells lines by the manufacturer and this further confirmed by siRNA knock-down by Peng et al. (2017).

The anti-c Jun (60A8) antibody has valdated by the manufacturer by western blot analysis of extracts from control HeLa cells or c-Jun knockout HeLa cells  and further confirmed by Yu et al. (2012) by knock-down using siRNA.

Zhu L, Ding R, Zhang J, Zhang J, Lin Z. Cyclin-dependent kinase 5 acts as a promising biomarker in clear cell Renal Cell Carcinoma. BMC Cancer. 2019 Jul 16;19(1):698. doi: 10.1186/s12885-019-5905-9. PMID: 31311512; PMCID: PMC6636025.

Tang JZ, Xu WQ, Wei FJ, Jiang YZ, Zheng XX. Role of Nampt overexpression in a rat model of Hashimoto's thyroiditis and its mechanism of action. Exp Ther Med. 2020 Apr;19(4):2895-2900. doi: 10.3892/etm.2020.8539. Epub 2020 Feb 21. PMID: 32256774; PMCID: PMC7086292.

da Rocha Sobrinho HM, Saar Gomes R, da Silva DJ, Quixabeira VBL, Joosten LAB, Ribeiro de Barros Cardoso C, Ribeiro-Dias F. Toll-like receptor 10 controls TLR2-induced cytokine production in monocytes from patients with Parkinson's disease. J Neurosci Res. 2021 Oct;99(10):2511-2524. doi: 10.1002/jnr.24916. Epub 2021 Jul 14. PMID: 34260774.

Peng C, Zhang S, Lei L, Zhang X, Jia X, Luo Z, Huang X, Kuang Y, Zeng W, Su J, Chen X. Epidermal CD147 expression plays a key role in IL-22-induced psoriatic dermatitis. Sci Rep. 2017 Mar 8;7:44172. doi: 10.1038/srep44172. PMID: 28272440; PMCID: PMC5341158.

Yu Z, Sato S, Trackman PC, Kirsch KH, Sonenshein GE. Blimp1 activation by AP-1 in human lung cancer cells promotes a migratory phenotype and is inhibited by the lysyl oxidase propeptide. PLoS One. 2012;7(3):e33287. doi: 10.1371/journal.pone.0033287. Epub 2012 Mar 15. PMID: 22438909; PMCID: PMC3305320.

Flow cytometry antibodies:

Antibodies for Flow cytometry were validated by the manufacturer (Biolegend) by staining of positive cells/tissue against matched Isotype control.

The anti-Human CD45-FITC  (clone H130) was validated for FACS in cell lines by manufacturer and authors using human and murine cells. These include the validation of anti-Human CD45-FITC  (clone H130) in FACS analysis of hepatic NK cells (Marquardt et al. 2015) and human hematopoietic progenitors (Chabi et al. 2019).

The anti-CD9-APC/Fire (clone H19a) antibody was validated for FACS using by Earley et al., (2021) in iPSC-derived neural cell mixture.

The anti-FOLR2 (clone 94b/FOLR2) has been validated by Western blot in analysis of folate receptor β (FRβ)–transfected B300-19 macrophage cells (Nagayoshi et al., 2005). Anti-human FRβ was validated in for FACS using human peripheral blood monocytes: https://www.biolegend.com/en-us/products/apc-anti-humanfolate-receptor-beta-fr-beta-antibody-15117

Chabi S, Uzan B, Naguibneva, I, Rucci, J, Fahy, L, Calvo, J, Arcangeli ML, Mazurier F, Pflumio F, Haddad R. Hypoxia Regulates Lymphoid Development of Human Hematopoietic Progenitors. Cell Rep. 2019 Nov 19;29(8):2307-2320.e6. doi: 10.1016/j.celrep.2019.10.050. PMID: 31747603

Marquardt N, Beziat V, Nystrom S, Hengst J, Ivarsson MA, Kekalainen E, Johansson H, Mjosberg, J, Westgren M, Lankisch TO, Wedemeyer, H, Ellis EC, Ljunggren HG, Michaelsson, J. Bjorkstrom NK. Cutting Edge: Identification and Characterization of Human Intrahepatic CD49a+ NK Cells. J Immunol (2015) 194 (6): 2467–2471. https://doi.org/10.4049/jimmunol.1402756

Earley AM, Burbulla LF, Krainc D, Awatramani R. Identification of ASCL1 as a determinant for human iPSC-derived dopaminergic neurons. Sci Rep. 2021 Nov 15;11(1):22257. doi: 10.1038/s41598-021-01366-4. PPMID: 34782629;  PMCID: PMC8593045

Nagayoshi, R, Nagai, T, Matsushita, K, Sato, K. Sunahara, N, Matsuda, T, Nakamura, T, Komiya, S, Onda, M, Matsuyama, T. Effectiveness of anti–folate receptor β antibody conjugated with truncated Pseudomonas exotoxin in the targeting of rheumatoid arthritis synovial macrophages. Arthritis Rheum. 2005 Sep;52(9):2666-75. doi: 10.1002/art.21228.  PMID: 16142741

# Eukaryotic cell lines

Policy information about cell lines and Sex and Gender in Research

| Cell line source(s) | Immortalized human adipose-derived stromal cells (Bmi-1/hTERT, iHASC) were acquired from Applied Biological Materials (T0540), derived from a female donor (30yrs-old). |
| --- | --- |
| Authentication | Cells were authentified by the manufactured by confirming the expression profile ( CD44,CD73,CD105). |
| Mycoplasma contamination | Cell tested negative for mycoplasma at source. |
| Commonly misidentified lines (See ICLAC register) | No commonly misidentified cell lines used in the study. |

# Plants

| | |
|---|---|
| Seed stocks | *Report on the source of all seed stocks or other plant material used. If applicable, state the seed stock centre and catalogue number. If plant specimens were collected from the field, describe the collection location, date and sampling procedures.* |
| Novel plant genotypes | *Describe the methods by which all novel plant genotypes were produced. This includes those generated by transgenic approaches, gene editing, chemical/radiation-based mutagenesis and hybridization. For transgenic lines, describe the transformation method, the number of independent lines analyzed and the generation upon which experiments were performed. For gene-edited lines, describe the editor used, the endogenous sequence targeted for editing, the targeting guide RNA sequence (if applicable) and how the editor was applied.* |
| Authentication | *Describe any authentication procedures for each seed stock used or novel genotype generated. Describe any experiments used to assess the effect of a mutation and, where applicable, how potential secondary effects (e.g. second site T-DNA insertions, mosiacism, off-target gene editing) were examined.* |

# Flow Cytometry

## Plots

Confirm that:

☒ The axis labels state the marker and fluorochrome used (e.g. CD4-FITC).

☒ The axis scales are clearly visible. Include numbers along axes only for bottom left plot of group (a 'group' is an analysis of identical markers).

☒ All plots are contour plots with outliers or pseudocolor plots.

☒ A numerical value for number of cells or percentage (with statistics) is provided.

## Methodology

| | |
|---|---|
| Sample preparation | We used a modified SCENITH-based approach to evaluate human macrophage metabolic pathways ex vivo29 . Fresh subcutaneous AT was cut into ~2mm pieces (30ml HBSS (Gibco 14175-053) in a 50ml tube), washed and collected using a 100-µM cell strainer. Tissue was digested for 20-mins at 37C (3mg/ml Collagenase II (Sigma C6885) in methionine-free RPMI (Sigma R7513), 65 mg/L L-cystine dihydrochloride (Sigma C6727), 1x GlutaMAX (Gibco 35050061), 10% dialysed foetal calf serum (Gibco A3382001)). Digested tissue was filtered through a 100-µm strainer and digestion was terminated by addition of Methionine-free RPMI containing 10% FCS, followed by centrifugation (300-g at 4C for 7-min). Following resuspension in methionine-free RPMI (65 mg/L cystine, 10% FBS, 1x Glutamax), cells were plated (160-µl) into wells on a 96-well V-bottom plate. Cells were methionine starved for a further 15mins (total starvation ~45mins including digestion and isolation) before treatment with inhibitors or control media (40µl) for 15mins. The four treatments were media, 2-Deoxy-D-glucose (2-DG) (100mM final conc.; Sigma D8375), Oligomycin (2 µM final conc.; Sigma 495455) and 2-DG+Oligomycin (100mM & 2µM final conc. respectively). Homopropargylglycine (HPG; Cayman Chemical, 11785) was then added to wells at a final concentration of 500-µM and incubated for 30-min to initiate cell HPG uptake. An additional well received cells and media but no HPG and no treatment (click-chemistry negative control). After HPG uptake, cells were stained with zombie aqua live/dead stain (1:500 in PBS; BioLegend 423101) for 20-mins at RT in the dark, washed with PBS and then fixed with 2% PFA for 15-min.<br><br>Fixed cells were permeabilised (0.1% saponin/1% BSA in PBS) for 15-mins, washed with Click buffer (100mM Tris-HCL, pH7.4; Invitrogen 1556-027), and incubated with Fc receptor blocker (25µg/ml in PBS; Fc1, BD Biosciences 564765) for 10-mins. Cells were rewashed and incubated in 100µl of Click reaction mix in the dark at RT for 30-mins. Click reaction mix was made sequentially, adding CuSO4 (Final conc. 0.5mM; Sigma 209198), THPTA (Final conc. 2mM; Antibodies.com A270328), Sodium Ascorbate (Final conc., 10mM; Sigma, A7631) and then AZDye 555 (Final conc. 25µM; Vector Laboratories, CCT1479) to Click buffer (final concentration, 100mM Tris-HCL).<br><br>After Click chemistry exposure, cells were washed using FACS buffer (PBS, 1% BSA, 5mM EDTA, 25mM HEPES), and stained with antibody mix (FACS buffer, anti-CD45 FITC [1:20; H130, Biolegend 304006], anti-FOLR2 APC [1:20; 94b/FOLR2, Biolegend 391705], anti-CD9 APC-fire [1:20; H19α, Biolegend 312114], Fc Block reagent [25µg/ml]) at 4C in the dark for 30mins. After re-washing, cells were filtered (35µM cap strainer) for FACS analysis |
| Instrument | Sony ID7000 |
| Software | AF Finder tool , Flowjo |
| Cell population abundance | CD45 positive immune cells, median 9.9% (IQR 6.5-12.3%) of live single cells. FORL2 positive TRMs, median 24.4% (IQR17.7-28.8%) of CD45 positive immune cells. CD9 positive LAMs, median 9.9% (IQR 7.7-11.9%) of CD45 positive immune cells. |

Gating strategy

LAM and TRM gating for SCENITH-based bioenergetic studies. Gate 1: All cells (FSC-A and SSC-A). Gate 2: Single cells (FSC-A and FSC-H). Gate 3: Live immune cells (CD45-hi, Zombie-Aqua-lo). Gate 4A: TRM cells (FORL2-hi, CD9-lo). Gate 4B: LAM cells (FORL2-lo, CD9-hi). AZ555: Click Chemistry histogram of AZ555 for respective Gates 4A and 4B. Fluorochrome and autofluorescence signatures were identified in unstained aliquots of each sample using the "AF Finder" software feature, were used to unmix the signals in fully stained samples with the built-in WLSM algorithm.

☒ Tick this box to confirm that a figure exemplifying the gating strategy is provided in the Supplementary Information.

