## [Peer Review file · Nature]

Selective remodelling of the adipose niche in obesity and weight loss.

Corresponding Author: Dr William Scott

Version 0:

Reviewer comments:

Referee #1

(Remarks to the Author)

In this manuscript by Miranda et al, the authors generate a single nuclear RNA-sequencing dataset of abdominal subcutaneous fat from individuals who are lean, obese, or who have lost weight following bariatric surgery. They have also generated a smaller spatial (Xenium) dataset from these subjects. They specifically interrogate macrophage and adipocyte population changes under these different conditions, and identify a signature of 'stressed' cells that is shared across a number of adipose cell types specifically in the obese condition. This is a large dataset that will undoubtedly be a resource to the adipose community, and this would be the first published adipose spatial dataset using Xenium (which provides greater resolution than the more commonly used Visium system). Of note, many of the conclusions in this paper are things that have previously been observed using other, older modalities. Specific comments are as follows:

Major concerns:

1. As mentioned above, I believe this would be the first published Xenium dataset of adipose tissue. From the analysis shown here, this appears to represent a significant improvement on previously published Visium datasets in terms of resolution. However, the dataset is underutilized. For example, in Fig. 3h the authors look at adipocyte size in relation to JUN expression. What about other genes? What about adipocyte subclusters- are there gene expression changes in larger vs smaller adipocytes independent of condition? Are there immune-adipocyte interactions that can be inferred from their spatial relationships?
2. The authors should better attempt to reconcile the subpopulations they identify in the spatial dataset with those found in the sNuc dataset. While some effort is made in Fig. 3g to look at marker genes, this is difficult to interpret. Some sort of matching here would greatly increase utility and reduce confusion. For example, in Fig. 5 the authors plot spatial proximity, but because this is on the spatial clusters and not the sNuc clusters it's difficult to interpret in relation to the rest of the work. What cells are the "beige" AD8 cells most like? What is the AD BAMBI population (a marker gene not mentioned outside of 2 plots of the spatial data)?
3. The authors call the AD8 subcluster "beige", but, besides ESRRG, none of the marker genes look particularly thermogenic. Is there an upregulation of UCP1 or other thermogenic genes in these cells?
4. The authors have combined their data with data from Emont, et. al. for most of their analysis. Those authors identified 7 adipocyte populations-- are all of those populations represented here? Are any of the AD1-8 populations specific either to this dataset or to the Emont dataset? Please directly compare these populations to the Emont populations, for example, using a label transfer approach. Please also provide an individual breakdown of the adipocyte subpopulations (as in Fig. 1d) to see how specific any of the populations are to an individual and/or dataset.
5. The authors perform a "metabolic flux analysis" using COMPASS and uncover some potentially interesting inferences regarding metabolite usage in macrophages under different physiological states. It's important to recognize that all of this was done using gene expression differences from the sNuc data. Ideally, the authors could sort out some LAMs from lean and obese individuals and test to see if their metabolic profiles have truly changed. At a minimum, the language describing these experiments should be made clearer to indicate that these are not actually flux measurements of metabolites.
6. In Fig. 4e, why are there regions with high stress scores in the lean samples? Are there common characteristics of those regions, like distance from vasculature? How do the authors explain the abundance of the stress signature in the lean compared to the weight loss samples?
7. The plots where the authors compare obese with lean or weight loss (i.e., Figs 2d, 3b, and 3c) are confusingly labeled, partially because there is no color key on the figure and partially because it is a little counterintuitive to have your upregulated points be the condition not labeled on the plot (i.e., the orange upregulated points on the "Lean" plot are in fact

upregulated in the obese condition).

Minor concerns

1. There is a typo in Figure 2d (“gluoc” instead of “gluco”).

Referee #2

(Remarks to the Author)

Miranda and colleagues aimed to characterize adipose tissue dynamics in obesity and therapeutic weight loss. For this, they generated a single-nucleus transcriptomic atlas complemented with a spatial transcriptomics map of human subcutaneous adipose tissue from individuals with extreme obesity undergoing therapeutic weight loss surgery, compared to healthy lean counterparts. They identified gene regulatory mechanisms and tissue signals that drive a cycle of senescence, tissue injury, and metabolic dysfunction. Weight loss was found to reduce adipocyte hypertrophy and biomechanical constraint pathways, enhancing metabolic flux and bioenergetic cycles, which correlated with systemic metabolic health improvements. Although weight loss decreased obesity-induced macrophage infiltration, it did not fully reverse their activation, leaving these cells primed for potential weight regain and metabolic dysfunction. Overall, the study provides new insights into the molecular basis of adipose tissue dysfunction in obesity and its reversal through weight loss.

These insights are derived from a comprehensive analysis of the generated single-nucleus (sn)RNA sequencing data using an extensive cohort. Integrated with data from Emont and colleagues, it serves as a valuable research resource, leveraging a powerful strategy in using samples from the same patients pre and post-surgery. However, and despite the undeniable value of the generated transcriptomic datasets, the manuscript is primarily of descriptive nature and lacks functional validation of its findings, while posing also conceptual issues, as detailed below. In addition, some novel and interesting pieces of information, such as the spatial transcriptomics datasets, are not thoroughly explored and could have contributed much more value to this work, as the transcriptomic characterization of cellular heterogeneity in subcutaneous adipose tissue has already been addressed in several studies (Miao et al., 2020; Whytock et al., 2022; Emont et al., 2022, Ferrero et al., 2024).

Our major comments can be found below:

1. Conceptually, it is challenging to understand why a lean condition is not necessarily the healthiest compared to obese patients who have lost weight but still have a high BMI (around 35). Indeed, despite the intervention, the post-surgery patients remain at a really high BMI range, yet they show less stress, at the gene expression level, than lean patients. This is intriguing, but also troubling, as it in a way suggests that it would be healthier longer-term to recover from extreme obesity compared to staying lean, which cannot be the message the authors want to communicate. How can these observations be explained from a molecular or physiological perspective? Could more validations beyond gene expression be provided? For instance, cell populations could be FACS-sorted to measure metabolic properties or stress response markers at the protein level. If blood samples are available, various parameters from blood samples could be used to analyze adipose-related secreted molecules testifying for stress or for a “healthier” profile.
2. It is unclear and unexplained what would be the drivers of the reported stress, as it seems systemic: “Moreover, stressed cell states did not form discrete niches, suggesting the micro-environmental drivers of these maladaptive changes may be extracellular or pervasive, and that selective vulnerability to stress may be cell intrinsic.” Identifying these drivers would significantly increase the study’s novelty, as these may constitute interesting targets for downstream therapeutic strategies. As a sidenote, in Figure 4, how was the threshold defined to determine whether a cell is classified as “stressed” or not? The authors should also better interpret and explain the lean/obese results presented in this figure.
3. Although the bioinformatic analyses are extensive and of seemingly high quality, the study often lacks clear hypotheses or justifications for these analyses or any functional validation of the main claims. For example, the authors claim “we propose that sustained saturated FA exposure in obese AT may activate TLR2-TREM1-mediated LAM polarization from a lipid-buffering to a pro-inflammatory phenotype.” but this stays at the hypothetical level and no functional data is provided to support that. In a similar vein, statements like “We infer this leads to increased metabolic flexibility in mature adipocytes, improved differentiation capacity in precursors and recovery of vascular abnormalities” cannot be made in the absence of a true experimental paradigm. Could the authors at least try to support some of these claims experimentally (metabolic flexibility, improved differentiation capacity, etc.)?
4. It is regrettable that the spatial transcriptomics data are not utilized more extensively. We expected more from this dataset than a few superficial validations, as it feels underused and could also have enhanced the novelty of the paper. For example, by examining if the different cellular states of each population (adipocytes, ASCs, etc.) are spatially arranged. This, in turn, could help support some of the intercellular communication cues proposed later in the manuscript.
5. Following the above point, there are some notable disconnections between the snRNA-seq data and the spatial transcriptomic datasets. For example, the authors claim stress based on c-Jun expression. Yet, c-Jun is expressed at very high levels in lean patients, while low in the AD4 cluster on the snRNA-seq dataset. In contrast, c-Jun is expressed much lower in the lean patients than in the AD4 cluster in the spatial transcriptomic dataset. Other markers also suffer from this sort of contradictory outcome. How can these results be reconciled? Which one should be trusted in order to build a conclusion for example linked to cellular stress?

6. Some interpretations are overly speculative. For example, while predicting metabolic fluxes from transcriptomic data is interesting, the limitations of this approach and need for functional validation should be clearly acknowledged. The term "phenotype" is frequently used throughout the manuscript, despite not referring to an actual biological phenotype. Similarly, the terms "functional" and "adipocyte activation" are ambiguous and their meanings are unclear in the context of adipose biology, using nomenclature of the field could be more useful. As such, we feel that the large levels of speculation lead to statements that feel too bold and actually weaken the manuscript, as also already indicated above.

7. It is unclear whether the dataset that is presented and analyzed throughout the manuscript is solely the one generated by the authors or if it is systematically combined with the dataset generated by Emont and colleagues. The authors should clarify this distinction.

8. To address the ambiguity of the above point (and in fact judge the quality of the presented analyses in general), it would be very useful to make the code accessible. This would help clarify some of the analysis parameters, such as integration. It appears that the integration between the two datasets was either remarkably seamless or possibly overintegrated, which could mask meaningful biological differences. For instance, could the authors provide the UMAP from their own dataset without integrating it with the dataset from Emont and colleagues?

9. The manuscript frequently mentions the aspect of "therapeutic development", but the authors should be more cautious suggesting any therapeutic applications based on their results. The findings are highly descriptive, and thus the potential for actual clinical application may be limited.

Minor comments:

1. In Figure 1, it would have been interesting and expected to see some correlations on the patient characteristics with cell type abundance. Additionally, abundance plots could have been included for other cell types as well.

2. In the text, the authors claim: "and relative increases in other prevalent adipocyte subpopulations including lipogenic AD5 cells." There are less calories / lipids, so why make more lipids? How can this be explained?

3. The clustering is based on only a few genes, this might not be enough. Moreover, some are stressed (c-Jun) and some are fibrotic (SLC14A2) markers, would we not expect high c-Jun expression in the fibrotic cell population as well?

4. In general, figure legends could be improved for clarity within the figure panels. For example, in Figure 3, the color labeling is difficult to understand, requiring frequent reference to the figure legend text. This would greatly improve the reading flow.

5. The main message of Figure 5 is not clear and requires further clarification.

6. The contrast in Figure 6G could be increased for better visibility.

7. The term "likelihood" used in Figure 8D has a very specific statistical meaning and appears inaccurate in this context.

8. In Figure S2D, the y-axis label seems wrong as the axis is already scaled.

9. In Figure S4A, where is c-Jun expression? How stable is this grouping?

Referee #3

(Remarks to the Author)

Summary

While this paper is well written and the data is of great interest as a resource, it is not the first single-cell atlas of adipose tissue. However, inclusion of paired weight-loss samples improves the novelty and usability of the data. Additionally, it may be the first to include spatial analysis of human adipose in lean and obese individuals. Ultimately, despite utility as a reference and very intriguing findings, the results are largely correlative and have not been extensively validated at the protein level. It remains unclear which components (SASP, global metabolic changes, etc) are drivers of metabolic disease in obesity or recovery in weight loss and if those factors are manipulatable as treatment options. Regardless, the authors highlight significant areas that have not yet been explored in human adipose tissue biology and that should be the basis for continued future work by them and others. One strength is that the Obese/weight loss patients are matched samples (weight loss are post-surgical samples). However, the average BMI of weight loss group is still in the morbidly obese classification (extended table 1). This raises some concern about their comparator for the lean group. The quality of the figures is excellent and the descriptions are also very high quality. I appreciated that many "hits" are displayed in figures, but the focus is centered on a subset of those hits with explanations for their relevance in the disease setting.

Major

• The authors identified LAM subsets in single cell data (adaptive and inflammatory), but did not evaluate these subpopulations similarly in the spatial datasets. Colocalization of adipocytes/LAMs would be extremely valuable knowledge

that does not currently exist and would benefit an atlas. Do the adaptive LAMs appear near early CLS, but the inflammatory near late-stage CLS? Colocalization (figure 5a) appears to be low for all cell types, but it is known that adipocytes are densely packed and frequently surrounded by immune cells. Is this an artifact of cell segmentation of the Xenium data? Perhaps the bin size should be increased to better show colocalization of large cells such as adipocytes?

• Senescence/SASP was identified as a key change (reduction) in the obese to weight loss transition. However, SASP, including p21, was similar between lean and obese individuals (i.e. does not appear to be an obesity-driven phenomenon, but rather a WL impaired phenomenon). The Discussion should further evaluate this finding and offer potential explanations (or limitations) as to why this may be. For example, the authors describe enhanced bioenergetics (e.g. lipid cycling) in weight loss that may not be also enhanced in lean individuals compared to obese; is this directly associated with reductions in SASP? Finally, NAMPT decreases with aging and has been shown to increase with exercise; age range of the study participants is 20-70 years old. Thus, difference in the image in 6h could be due to age rather than obesity vs weight loss.

Minor:

- The authors suggest inflammation is reduced in weight loss, but see little evidence of inflammatory macrophages in all groups. It would be beneficial to include speculation in the Discussion.
- The y-axis on extended data figure 2d may be incorrect (labeled as -log, but also presented on a log scale).
- It would be helpful if the text referred to LAM1 and LAM2 (adaptive and inflammatory) as they are presented in the figures (LAM ST1 and LAM ST2)
- Bar graphs (e.g figure 2C) would benefit from statistics. If these are not significant, the results may be overstated (trend vs. statistically significant finding)
- The section labeled “Weight loss reduces innate immune infiltration” appears to be mistitled, as it refers to specifically adaptive populations of cells.
- Adipocyte area presented as a log-scale makes it hard to recognize how large of difference in adipocyte size there is in Lean -> obese -> weight loss
- Need clarification on whether Figure 4e is scaled expression (across all samples or per group?) vs. normalized expression.
 - o Unclear if expression is overall lower (normalized) or if there is less variance in expression (scaled).
 - o Would also be valuable to indicate the 24 genes that were used for scoring of spatial data in the extended data file (perhaps by bold or a second sheet; Supp File 12).
- The gene AREG was noted in the methods to be added as part of the secreted SASP markers. A brief mention as to why this was added in the Results would be an improvement over mention in the Discussion.
- The Discussion would be improved by the inclusion of a limitations section, which could elaborate on many of the following:
 - o Before weight loss, samples were collected by surgical incision, but post-operative samples were collected by needle biopsy.
 - o Discussion of non-surgical weight loss should be included. This is particularly important given the comparable weight loss induced by the new drugs on the market.
 - o All patients are not classified as T2D despite elevated insulin and HOMA-IR. This should be mentioned in the discussion/limitations.
 - o Discussion of potential differences between subcutaneous and visceral adiposity should be mentioned.
 - o Low overall numbers of some cell types limit interpretation
- It is excellent that authors are providing processed transcriptomics data in two formats as a resource; however, the Github page is not currently accessible.

Referee #4

(Remarks to the Author)

Please see comments of co-reviewer.

Referee #5

(Remarks to the Author)

Please see comments of co-reviewer.

Version 1:

Reviewer comments:

Referee #1

(Remarks to the Author)

The authors have made significant efforts to respond to my concerns. The work is technically sound, and they have done more to leverage the spatial dataset. They have also performed more experiments to validate some of the predictions raised by the single cell data. I have no further substantive concerns.

Referee #2

(Remarks to the Author)

The revised version of Miranda et al. shows some significant improvements and the authors must be commended for their

efforts. They make better use of their spatial transcriptomic data, and new hypotheses have been included as to why adipose tissue from lean people showed higher stress than that of obese individuals undergoing body weight loss interventions. Nevertheless, we think that some main aspects remain vague and would need to be further addressed:

- Although we acknowledge the attempt of the authors to better use their spatial transcriptomic results, this exciting dataset could still have been used more extensively. The different conditions (lean vs. weight loss) could have been exploited more.

- One of the main take-home messages of the paper, namely that overcoming extreme obesity is seemingly more beneficial for health than simply remaining lean, is still very difficult to understand. The authors push forward a hypothesis relying on the senescent state of the tissue. This is an unexpected finding and, given that it is the cornerstone to explain a major counterintuitive phenotype, more experimental validations (e.g.: ex vivo) would be required to solidify it. Indeed, with the current level of provided support, we remain uncomfortable underwriting such a conclusion, as it is in our opinion highly provocative and may reflect latent issues with the lean cohort that is biasing the findings.

- Some claims that are made in this manuscript might be misleading in terms of the scope of the current work. We previously mentioned a sentence claiming that "we propose that sustained saturated FA exposure in obese AT may activate TLR2-TREM1-mediated LAM polarization from a lipid-buffering to a pro-inflammatory phenotype." In their rebuttal, the authors confirm that these causal mechanistic links are inferred from the literature, which should be clarified in the main text. In a second claim, the authors stated that "We infer this leads to increased metabolic flexibility in mature adipocytes, improved differentiation capacity in precursors and recovery of vascular abnormalities" which now the authors tried to address experimentally. Yet, the presented experiment only helps to show that adipocyte precursors treated with etoposide, as a stressor causing DNA damage, fail to differentiate. This poorly relates to the claim made by the authors. Nobody would probably question that substantial DNA damage can have a significant impact on adipocyte differentiation. Could the authors at least show ex vivo that the differentiation capacity of adipocyte precursors from lean and obese patients is reduced compared to those of people after weight loss (and, therefore, less stressed)? If so, and given the main hypothesis that the authors put forward at the moment, they could examine if senolytic agents could improve the differentiation capacity of the stressed groups compared to cells from non-stressed tissues.

- We thank the authors for providing the separated UMAPs from their own dataset and the Emont et al. one. It would have been interesting and valuable to provide the stratification UMAP plots as well, to visualize how the cell clustering is affected by the different conditions. Additionally, can the authors discuss the similarities between the Emont et al. plots with their weight-loss condition? Finally, responses Fig. 10 should be placed in the main figures.

Referee #3

(Remarks to the Author)

The authors have extensively revised the manuscript. All of my concerns have been addressed.

Referee #4

(Remarks to the Author)

The revised manuscript by Miranda et. al. presents a novel atlas of subcutaneous adipose tissue during the settings of obesity and weight loss using single nuclei transcriptomics and spatial gene expression analysis. Using nuclear and cell-boundary segmentation approaches, the authors identify regions of cellular stress occupied by large adipocytes and lipid-associated macrophages. Furthermore, the authors highlight how weight loss modifies signatures of cell stress in paired-specimens. The dataset generated for this study is a novel and valuable resource and the authors have made their code available to improve accessibility of this dataset for future work. The inclusion of paired specimens of obese and weight-loss tissue provides additional value and novelty. Revisions to improve nomenclature/labels and validate findings with external approaches (SCENITH, microscopy) greatly enhance the readability and support the authors conclusions. Finally, inclusion of a new figure evaluating the spatial cellular niches that present in lean, obese, and weight loss subcutaneous adipose tissue is a significant improvement. The authors have also improved their discussion to reduce speculative statements and include limitations.

Minor Comments:

1. In Fig. 2A, three populations are listed as MY1, MY2, and MY3 and should be renamed to MYE1, MYE2, and MYE3 in line with the rest of the figures.

2. While the limitations section is newly included, it is very brief and could be improved by providing some description as to how these limitations might affect the interpretation of the study. For example, the authors state, "Other limitations include...the abdominal subcutaneous depot and surgical weight loss...". Differences between subcutaneous and visceral are well reported and therefore the authors have a great opportunity to speculate on how visceral may differ in regard to their major findings (cell stress, signaling, spatial niches, et). Furthermore, medical weight loss is a rapidly expanding field and input from the authors on how medical weight loss may result in differences (if any) from surgical weight loss in the context of their findings would be very interesting.

Referee #5

(Remarks to the Author)

Version 2:

Reviewer comments:

Referee #1

(Remarks to the Author)

The authors have made clear that they are not claiming that losing weight is more healthy than never having been obese in the first place. I believe further experiments are beyond the scope of the current study.

Referee #2

(Remarks to the Author)

We thank the authors for thoughtfully replying to our comments and for better contextualizing the biomedical implications of their findings. In our opinion, further evidence would still be needed to fully demonstrate that adipocyte precursors and vascular cells from lean healthy patients display higher senescence and stress markers than those of class III obese patients after body weight loss – being still remarkably obese. Since the causes and functional implications of this rather controversial finding, at present, remain unknown, we cannot underwrite it. However, we also acknowledge that the study provides a valuable dataset and that the current version of the manuscript constitutes already a rich and dense piece of work. Therefore, we understand that a comprehensive exploration of the above topics could be beyond the scope of a single paper.

NATURE 2024-05-10393, RESPONSE TO REVIEWERS.

We thank the reviewers for highlighting the overall quality of our work, and value of our datasets. We are grateful for their considered and constructive comments. We have made extensive revisions to our manuscript to address the points raised by them. In particular, we have significantly strengthened our spatial analyses, answered conceptual questions about the differences between leanness and weight loss, and provided greater ex-vivo and experimental validation. The details are provided in two separate sections:

1. Responses to common themes raised by more than one reviewer.

2. Point by point responses to individual referees.

1. Responses to common themes raised by more than one reviewer:

1.1 Comprehensive reanalysis to make more use of the spatial dataset:

We agree with the reviewers our Xenium dataset represents a significant advance on previously published (mainly Visium) datasets in adipose, and acknowledge we should make more use of it.

In response, we have made two major changes. First, we have tested multiple methods to re-define and re-annotate spatial cell types/states. The resulting spatial populations better reflect the larger 'reference' single nucleus dataset. Second, we have developed new approaches to delineate and interrogate tissue niches. This has enabled us to address important biological questions raised by the reviewers about the features of stressed tissue microenvironments, colocalisation of metabolic and immune cells, and intercellular cues that may elicit tissue remodelling in obesity and weight loss.

Spatial reannotation:

We have compared several spatial annotation methods to improve concordance with the larger single nucleus reference. Most approaches work equally well at cell type level. Direct clustering and annotation using marker genes selected for their specificity performs best at recovering the majority of cell states. Label transfer (without or with ENVI imputation, Haviv et al.) is the only method that delineates rarer cell types. We therefore use direct clustering for spatial population annotation, and label transfer to specifically investigate rare cell states such as LAM subtypes (which we cross-verify with marker genes). To further improve dataset concordance, we remove clusters that separate on single non-marker genes (for example *BAMBI*, which are flagged as unassigned). The result is significantly better overlap between the single nucleus and spatial datasets (**Responses Fig. 1a/b, Extended Data Fig. 1b/c**).

Spatial niches:

We have comprehensively reanalysed the spatial arrangement of cells within tissue niches using several unbiased methods. Distance-based neighbourhood clustering allows us to define tissue niches enriched for: i. distinct vascular compartments thus validating our methods; ii. adipose stem cells and tissue resident macrophages; iii. stressed adipocytes, stressed precursors, LAMs, and other immune cells (**Responses Fig. 1c/d**). Regional bin-based scores reveal additional characteristics of high-stress zonation: i. concordance of stressed regions across clinical traits; ii. unexpected association with arteries; iii. greater T-cell accumulation in obesity (**Responses Fig. 1c/d**). Cell colocalisation analyses demonstrate that 'adaptive' LAMs aggregate together and map to crown-like structures (CLS). Whereas 'inflammatory' LAMs frequently occur as singletons or in pairs, (possibly early- or late-stage CLS, **Response to Reviewer 3 Point 1**). These new analyses orient cells to previously unrecognised zonation that may coordinate tissue homeostasis and dysfunction.

Spatially resolved communication signals:

Identification of tissue niches means we can infer diverse interniche communication signals within our spatial dataset (by imputing ligand and receptor gene expression), and then intersect the results with intercellular interactions in the larger (more powerful) single nucleus reference. This enables us to predict pathways that alter in obesity and weight loss, and map them to tissue niches and cells. Several of the identified ligands (for example *AREG*, *THBS1*, *NAMPT*) are presumptive SASP components (**Responses Fig. 1e and Extended Data Fig. 8/10**), suggesting these signals may drive senescence pathology. Other identified cues may have inflammatory and fibrotic roles.

In summary, this in-depth reanalysis significantly improves the interpretability and utility of our single nucleus and spatial datasets. It adds valuable new insights into tissue niches, and communication pathways within these niches that may propagate tissue remodelling. We have dedicated an entire results section, **Main Fig. 5**, and **Extended Data Fig. 6** to these important results.

Spatial uses:

- Comprehensive validation of all major study findings in an independent patient cohort using a distinct technology; this is equivalent to using RNA scope at scale.

- Systematic cross-verification of adipose tissue cell states in situ.
- Confirmation of the main cell abundance changes in situ.
- Analysis of adipocyte size-gene expression-trait relationships; discovery of biomechanical signalling, cell stress and fibrosis pathways associated with adipocyte size/hypertrophy.
- Identification of previously undescribed tissue niches, and their cellular components (new).
- Characterisation of low and high stress tissue regions within/between conditions (new).
- Understanding of inter-niche communication cues that alter in obesity and weight loss (new).
- Colocalisation of senescence markers to arterial and stressed tissue niches (new).
- LAM aggregation and orientation to CLS around transcriptionally devoid adipocytes (new).
- Localisation of LAM subtypes to distinct tissue features and locations (new).

Responses Fig. 1. A/B. Overlap of marker genes in the respective single nucleus and spatial datasets. Here limited to adipocyte, precursor and endothelial subtypes for space reasons, full annotation in **Extended Data Fig. 1b/c**. **C.** Representative tissue section showing bin-based (regional) stress scores (left); tissue architecture (WGA stain) with LAMs superimposed (middle); and tissue-niches defined by cell colocalisation and unbiased neighbourhood clustering (right). **D.** Heatmaps illustrating tissue mean

stress score (left) and quantiles for spatial cell states (middle), and cell state composition of individual tissue niches (right). **E.** Circos plots showing examples of interniche signals including AREG and NAMPT that alter reciprocally in obesity and weight loss.

Haviv et al. *The covariance environment defines cellular niches for spatial inference.* *Nat Biotechnol* (2024).
<https://doi.org/10.1038/s41587-024-02193-4>

1.2 Explaining levels of stress in leanness, obesity and weight loss:

The reviewers raise several questions about the differences in stress in lean, obese and weight loss tissues, and the wider physiological impact. We have condensed these down to key areas which we address here.

Are the findings real?

In our original submission we showed that stressed cell states are characterised by high senescence gene signatures, suggesting this stressed state reflects senescence vulnerability (**Extended Data Fig. 10e**). We also showed that stress, senescence and known senescence drivers were present in lean tissues, were significantly higher in obesity, and were potentially mitigated by weight loss. We did this using distinct single cell and spatial technologies (**Main Fig. 4b/e, Main Fig. 6b/c, Extended Data Fig. 10f**), validating our senescence findings at protein level (p21, NAMPT, **Main Fig. 6d/h**). These findings are consistent with previous studies showing:

- i. senescent cells propagate further senescence in a self-reinforcing feedforward deleterious cycle (Wiley et al.);
- ii. adipose is particularly vulnerable to senescence which arises early in this tissue compared to other organs, irrespective of obesity (Smith et al.);
- iii. senescence is a feature of lean adipose where it is observed at lower levels than obese adipose (Li et al.).

We have now further verified these patterns. In our single nucleus dataset, we confirm and better illustrate stress variations across conditions and cell types using cell-level stress scores (**Responses Fig. 2a**). In our reanalysed spatial dataset, we demonstrate that senescent markers and mediators enrich in high-stress tissue niches (i.e. stress and arterial, **Responses Fig. 2b**), reinforcing that stress and senescence are interlinked and self-propagating. In new ex vivo analyses, we quantify the stress proteins STAT3 and Jun in situ. Although our findings are non-significant due to the limited sample size (N=5/condition), STAT3 and Jun mirror trait-associated stress patterns (**Responses Fig. 2c**) and our previous senescence/SASP protein verifications. We conclude our findings are robust and reproducible across modalities, independent cohorts and samples.

Why is there stress in lean tissues?

Despite not being powered to examine this, we find that stressed cells systematically increase with age (**Responses Fig. 2d**), a major driver of adipose tissue senescence. We also identify an increase in stressed cells in lean tissues in association with worsening metabolic function, independent of adiposity and age (**Responses Fig. 2e**). We surmise that other biological factors such as age contribute to stress and senescence vulnerability in lean adipose tissue, and that senescence may be an unrecognised driver of metabolically unhealthy leanness.

How can we rationalise the tissue and systemic effects?

Using multiple approaches, we show that weight loss potentially mitigates tissue stress, senescence, and known drivers of senescence pathobiology (**Main Fig. 6** and **Extended Data Fig. 10**). This strongly suggests that weight loss (either through chronic caloric restriction or active fat mass loss) breaks the self-reinforcing degenerative senescence cycle. As this is one of the most widespread and sizeable changes in weight loss, we infer it is crucial for the observed improvements in metabolic function (**Main Fig. 1b**).

From a physiological perspective, our lean group is healthier than the comparators (**Main Fig. 1b**). At face value this appears contradictory. However, it is important to recognise that other cells impact adipose tissue and other tissues regulate overall metabolic health. There is a clear example of this in the immune compartment where weight loss does not reverse the macrophage activation states induced by obesity. Thus, the compound effects of other cells and tissues are likely to explain the differences in metabolic health between lean and weight loss groups.

Overall, this data implicates multicellular stress/senescence in metabolic dysfunction in the contexts of human obesity, age, and unhealthy leanness. It establishes that human weight loss has previously

undescribed, potent multicellular stress lowering and senolytic effects. We include these important new results in our revised manuscript together with expanded discussion.

Responses Fig. 2: **A.** Density heatmap showing cell-level stress scores (calculated from the 188 conserved stress genes) in each condition (scaled aggregate cell-level scores). **B.** Stress and senescence marker gene expression (imputed from large single nucleus reference) across diverse tissue niches (scaled mean expression). **C.** In situ quantification of the stress marker proteins STAT3 and JUN in lean, obese and weight loss tissues (N=5 per condition). **D/E.** Differential neighbourhood abundance analyses in association with age (left), and fasting insulin adjusted for age (right) in lean tissues. Enrichment P values in stressed adipocytes (AD3) and precursors (APC3) using the binomial sign test to compare the observed directions of effect with the expected null of 0.5.

Wiley et al. *The metabolic roots of senescence: mechanisms and opportunities for intervention.* *Nat Metab* 3, 1290–1301 (2021). <https://doi.org/10.1038/s42255-021-00483-8>

Smith et al. *Cellular senescence and its role in white adipose tissue.* *Int J Obes* 45, 934–943 (2021). <https://doi.org/10.1038/s41366-021-00757-x>

Li et al. *Obesity and hyperinsulinemia drive adipocytes to activate a cell cycle program and senesce.* *Nat Med* 27, 1941–1953 (2021). <https://doi.org/10.1038/s41591-021-01501-8>

1.3 Increased validation.

Previously we used the Xenium to systematically replicate cell level transcriptomic findings in an independent patient cohort with a separate technology. We considered this a substantial improvement over RNA scope, the method of choice for 'comprehensive' verification in many previous single cell studies. We then validated several of our findings at protein level.

We have now extended this in two main ways, by increasing protein-level verification and adding new experimental validation:

- i. We have evaluated and confirmed the existence of 'inflammatory' TLR2 +ve LAM subtypes in obese adipose tissues, because of the novelty of these populations, and their potential impacts on tissue homeostasis and dysfunction (**Response Fig. 3a, Fig. 2d**).
- ii. We have quantified the stress marker proteins Jun and STAT3 in situ across patient groups (N=5/condition) to provide additional evidence to support the transcriptomic variations seen in leanness, obesity and weight loss (**Response Fig. 2c**).
- iii. We have verified macrophage metabolic activation states, inferred from transcriptomic information (COMPASS), experimentally in obese adipose tissues. We used a flow cytometry-based method (SCENITH, Argüello et al.) to determine the metabolic profiles of tissue-resident (TRM) and lipid associated (LAM) macrophages in obese tissues (N=7). The main benefits of this approach are the allowance for low cell numbers (essential for small ex vivo samples) and avoiding metabolic biases introduced by culture. SCENITH analyses show that LAMs have higher basal metabolism and glycolytic capacity than TRMs (**Response Fig. 3b, Response to Reviewer 1 Point 5**), substantiating these and other metabolic activation states predicted in transcriptional flux analyses.
- iv. We have also tested and shown that in vitro induction of cell stress (using Etoposide to mimic DNA damage, a predicted stress trigger, see **Response to Reviewer 2 Point 2**) alters the basal functions of stromal cells. In human adipocyte precursors (APC), DNA damage (Singh et al.) induces the stress markers JUN and STAT3, and impairs in vitro differentiation competence (**Response Fig. 3c**). This experimental data reinforces the conclusion that tissue stress impairs normal stromal cell functions, and that reversal of stress by weight loss mitigates such effects.

Our new ex vivo and experimental data verify central manuscript findings, significantly strengthening the revised submission.

Responses Fig. 3: **A.** Spatial (top) and histological (middle and bottom) CLS. Immunofluorescence analyses confirmed the existence of TLR2 positive (LAM ST2, minority) and negative (majority) LAMs at CLS. **B.** Modified SCENITH-based analysis of ex vivo isolated, FACS-sorted macrophages from obese tissues (N=7 samples) confirmed the COMPASS-predicted increases in basal respiration and glycolytic capacity in LAMs compared to TRMs. **C.** Induction of cellular stress (using Etoposide to mimic the DNA damage response pathway, Smart et al.) upregulates the stress marker proteins Jun and STAT3 and impairs differentiation capacity as measured by Oil Red-O lipid accumulation in human adipocytes in vitro.

Argüello et al. SCENITH: A Flow Cytometry-Based Method to Functionally Profile Energy Metabolism with Single-Cell Resolution, *Cell Metabolism*, Volume 32, Issue 6, 2020, Pages 1063-1075.e7, ISSN 1550-4131, <https://doi.org/10.1016/j.cmet.2020.11.007>

Singh et al. Quantification of single-strand DNA lesions caused by the topoisomerase II poison etoposide using single DNA molecule imaging, *Biochemical and Biophysical Research Communications*, Volume 594, 2022, Pages 57-62. <https://doi.org/10.1016/j.bbrc.2022.01.041>

1.4 Recent study by Hinte et al.

We highlight upfront a study by Hinte et al. that was published just before our resubmission investigating the epigenetic memory of obesity after weight loss in adipose tissues.

We view this study as complementary and strongly synergistic to ours. It does not directly overlap or detract from the novelty of our findings or datasets. Hinte et al. examine the epigenetic legacy of obesity after weight loss. They do this specifically in the late weight maintenance phase rather than in early active weight loss. They focus on persistent epigenetic reprogramming in adipocytes rather than other cellular compartments. The work is framed as human and mouse. However, while the analysis of mouse data is extensive and of high quality, there is limited analysis of the human datasets meaning the main findings pertain to mouse alone. In addition, the human datasets are from different, individually small population cohorts which are analysed separately, and these are only available on request, limiting dataset utility. The pathways that drive the beneficial effects of weight loss are largely unstudied.

In contrast, we report a single nucleus and spatial atlas of human adipose tissues comprising >170K cells. To our knowledge, this is still the largest tissue-wide exploration of adipose in human obesity and the largest study of weight loss in any human tissue. It would also be the first reported cell resolution spatial annotation of human adipose using Xenium technology. Importantly, we investigate weight loss in its early phase to specifically define potential driver mechanisms. Together, this enables us to capture a rich representation of the cell types, regulatory mechanisms, and signalling niches that may mediate and mitigate adipose tissue dysfunction in human obesity.

It is not possible for us to integrate the human data from Hinte et al. at this stage. We would have to reanalyse the entirety of our dataset which would take many months. Instead, we discuss this important but substantially different study.

In summary, our human adipose tissue single nucleus and spatial datasets continue to represent a benchmark for human obesity and weight loss, and a crucial resource for mechanistic and therapeutic exploration.

Hinte et al. Adipose tissue retains an epigenetic memory of obesity after weight loss. Nature (2024).
<https://doi.org/10.1038/s41586-024-08165-7>

2. Point by point responses to individual referees.

Referee #1 (Remarks to the Author):

In this manuscript by Miranda et al, the authors generate a single nuclear RNA-sequencing dataset of abdominal subcutaneous fat from individuals who are lean, obese, or who have lost weight following bariatric surgery. They have also generated a smaller spatial (Xenium) dataset from these subjects. They specifically interrogate macrophage and adipocyte population changes under these different conditions, and identify a signature of 'stressed' cells that is shared across a number of adipose cell types specifically in the obese condition. This is a large dataset that will undoubtedly be a resource to the adipose community, and this would be the first published adipose spatial dataset using Xenium (which provides greater resolution than the more commonly used Visium system). Of note, many of the conclusions in this paper are things that have previously been observed using other, older modalities. Specific comments are as follows:

We thank the reviewer for highlighting the value of our results and datasets. We acknowledge some of our conclusions are not new. However, many are and these represent a significant advance given the profound global health impacts of obesity and weight loss.

Major concerns:

1. As mentioned above, I believe this would be the first published Xenium dataset of adipose tissue. From the analysis shown here, this appears to represent a significant improvement on previously published Visium datasets in terms of resolution. However, the dataset is underutilized.

We agree our Xenium dataset provides a sizeable improvement in resolution compared to existing spatial datasets in adipose. We have therefore comprehensively reanalysed it to improve its utility. This is detailed above in **Section 1.1**.

For example, in Fig. 3h the authors look at adipocyte size in relation to JUN expression. What about other genes? What about adipocyte subclusters- are there gene expression changes in larger vs smaller adipocytes independent of condition?

Previously we examined the correlation between adipocyte size and every gene expressed in adipocytes on the Xenium (**Main Fig. 3g**). We then presented *JUN* as an example (**Main Fig. 3h**). These analyses included gene expression-size relationships independent of condition (**Main Fig. 3g**, correlations panel). We regret this was not clear.

In our revised manuscript, we have reanalysed adipocyte populations in the spatial dataset to improve concordance with the nucleus reference. For size-based correlation analyses, we have used a new segmentation approach that assigns ambiguous transcripts on the boundary between ≥ 2 adipocytes to both/all cells. Previously we assigned the transcript to one neighbouring cell. This does not materially change the outcomes but is a conceptual improvement. For non-size-based (e.g. marker gene) analyses, we have used the reannotated nucleus segmentation instead. This 'cleaner' nucleus dataset reduces ambiguities arising from the contaminating effects of mis-assigned transcripts, and thus better reflects the nucleus reference (**Responses Fig. 1a**). We designate these 'boundary' and 'nucleus' segmentation in our revised manuscript.

Overall, this enables us to identify genes that vary with adipocyte size, independent of condition, and better predict the underlying cell states in the complementary single nucleus and spatial datasets (**Responses Fig. 4**).

Responses Fig. 4: Mean expression of mechanosensitive, stress, fibrotic, and homeostatic genes across conditions and adipocyte subpopulations, in respective single nucleus (left) and spatial (middle) datasets. Correlation (right) of genes with adipocyte areas in each condition and across all conditions combined (spatial dataset). Based on revised spatial analyses.

Are there immune-adipocyte interactions that can be inferred from their spatial relationships?

Revised spatial analyses mean we can now map cells to tissue niches (spatial), investigate inter-niche communication signals (spatial), predict the source and target cells (larger single nucleus reference), and link several findings to obesity and weight loss (single nucleus). Many of the identified communications involve immune-adipocyte interactions. However, few are exclusive to these cells as multiple target cells share equivalent receptor expression patterns.

Two biologically relevant examples are *TGFB1* and *THBS1* which enrich in the stress niche. *TGFB1* mainly arises in myeloid and lymphoid cells (which are also enriched in the stress compartment) and signals to adipocytes. While *THBS1* signals between stressed AD3 and LAMs, and alters reciprocally in obesity and weight loss. We provide these and other example interactions in a new results section, new **Main Fig. 5** and new **Extended Data Fig. 8**.

2. The authors should better attempt to reconcile the subpopulations they identify in the spatial dataset with those found in the sNuc dataset. While some effort is made in Fig. 3g to look at marker genes, this is difficult to interpret. Some sort of matching here would greatly increase utility and reduce confusion. For example, in Fig. 5 the authors plot spatial proximity, but because this is on the spatial clusters and not the sNuc clusters it's difficult to interpret in relation to the rest of the work. What cells are the "beige" AD8 cells most like? What is the AD BAMBI population (a marker gene not mentioned outside of 2 plots of the spatial data)?

We have completely reanalysed our spatial dataset (detailed in **Section 1.1**) to reconcile the single nucleus and spatial populations. We have also relabelled spatial populations to reflect the single nucleus reference. The improvement is shown in **Responses Fig. 1a and 4** as well as new **Main Fig. 3g and Extended Data Fig 1c**.

One disadvantage of direct clustering in the spatial dataset is separation based on single non-marker genes. *BAMBI* is an example of this. In the single nucleus dataset, *BAMBI* is expressed at low levels in multiple adipocyte subtypes (**Responses Fig. 5**) but at high levels in its own cluster in the spatial dataset. To mitigate this and similar effects, we have removed ambiguous clusters arising from single non-marker genes like *BAMBI* (flagged as unassigned).

Another limitation of direct clustering is inability to separate out rare cell states such as AD8. AD8 can be recovered using label transfer but is too rare to confidently report (it equates to <1 cell per sample).

Despite this, we faithfully annotate 31 of 47 of the reference single nucleus populations in our spatial dataset (**Extended Data Fig. 1c**), substantially improving its utility.

Responses Fig. 5: BAMBI expression in the larger single nucleus reference, across adipocyte subtypes and conditions.

3. The authors call the AD8 subcluster “beige”, but, besides ESRRG, none of the marker genes look particularly thermogenic. Is there an upregulation of UCP1 or other thermogenic genes in these cells?

There is very little expression of *UCP1* across the entire adipocyte population. This is not unexpected (Ding et al.) because the samples were collected in the fasted state to mitigate the potential confounding effects of different feeding status. AD8 does however overexpress *PRDM16*, the beige adipocyte master regulatory transcription factor, albeit at low levels, and *GATM* which also fits a ‘beige’ thermogenic classification (Kazak et al.). We have included *PRDM16* and *GATM* in the adipocyte marker plots (**Main Fig. 3a**).

Ding et al. Fasting induces a subcutaneous-to-visceral fat switch mediated by microRNA-149-3p and suppression of *PRDM16*. *Nat Commun* 7, 11533 (2016). <https://doi.org/10.1038/ncomms11533>

Kazak et al. Genetic Depletion of Adipocyte Creatine Metabolism Inhibits Diet-Induced Thermogenesis and Drives Obesity, *Cell Metabolism*, Volume 26, Issue 4, 2017, Pages 660-671.e3. <https://doi.org/10.1016/j.cmet.2017.08.009>.

4. The authors have combined their data with data from Emont, et. al. for most of their analysis. Those authors identified 7 adipocyte populations-- are all of those populations represented here? Are any of the AD1-8 populations specific either to this dataset or to the Emont dataset? Please directly compare these populations to the Emont populations, for example, using a label transfer approach. Please also provide an individual breakdown of the adipocyte subpopulations (as in Fig. 1d) to see how specific any of the populations are to an individual and/or dataset.

It is important to recognise that the Emont dataset integrates subcutaneous (N=14,396) and visceral (N=11,475) adipocyte nuclei, and reports strong depot-specific associations (Emont et al.). Thus, our combined dataset (N=41,727) which only includes subcutaneous nuclei is a more faithful representation of human subcutaneous adipocyte heterogeneity.

Nevertheless, we have analysed marker overlap between datasets (**Responses Fig. 6a**). This shows that markers reported as subcutaneous specific in Emont (hAd3-*PNPLA3*, hAd4-*GRIA4*, hAd7-*AGMO*) are largely conserved in our dataset while visceral specific markers (hAd2-*TNFSF10*, hAd6-*EBF2*) are not. Equally the markers we report are generally higher in the Emont subcutaneous clusters. One notable exception is *JUN* which forms a unique cluster in our dataset, presumably because of the large obese contingent.

Participant-level breakdown plots confirm the adipocyte subtypes we report are conserved across individuals and datasets, and reinforce the differences in abundance between traits (**Responses Fig.**

6b). The Emont samples are more variable. However, we cannot interpret this without more detailed phenotypic information.

Responses Fig. 6: **a.** Adipocyte marker genes reported by Emont and Miranda in the respective Emont (subcutaneous only) and Miranda (combined reported) datasets. **b.** Adipocyte subtype proportions in the Miranda dataset split by condition (LN lean, OB obese, WL weight loss) ranked by BMI, and in the Emont dataset also ranked by BMI.

Emont et al. A single-cell atlas of human and mouse white adipose tissue. *Nature* 603, 926–933 (2022). <https://doi.org/10.1038/s41586-022-04518-2>

5. The authors perform a “metabolic flux analysis” using COMPASS and uncover some potentially interesting inferences regarding metabolite usage in macrophages under different physiological states. It’s important to recognize that all of this was done using gene expression differences from the sNuc data. Ideally, the authors could sort out some LAMs from lean and obese individuals and test to see if their metabolic profiles have truly changed. At a minimum, the language describing these experiments should be made clearer to indicate that these are not actually flux measurements of metabolites.

We have now made it clear these analyses represent transcriptional modelling of metabolic flux rather than actual metabolic flux. In addition, we have carried out experimental studies to verify some of the metabolic activation states inferred in our datasets. This is also shown above in **Section 1.3**.

Specifically, we used a modified FACS-based metabolic profiling strategy (SCENITH, Argüello et al.) to compare the metabolic states of LAMs and tissue resident macrophages (TRMs) in obese adipose tissue (**Responses Fig. 7**). We did this for practical reasons: sorting LAMs from lean tissues would require very large tissue samples to retrieve very low cell numbers. Consistent with our transcriptomic results, we show that LAMs have significantly higher basal metabolic activity and capacity to utilise glycolysis than TRMs. This confirms our LAM-TRM predictions, substantiates the metabolic activation changes observed in obesity (compared to leanness) where LAMs are the predominant altered subtype, and upholds other transcriptome-based metabolic flux predictions.

SCENITH-based functional Energetic Profiles

Responses Fig. 7: Sorting of LAM and TRM macrophages from obese adipose tissues after treatment with media (Control) and the metabolic inhibitors 2DG, Oligo, and 2DG + Oligo. The effects of treatment on bioenergetic pathways were estimated using Homopropargylglycine HPG uptake (measured by Click-AZ555) as a proxy for cellular energy consumption.

Argüello et al. SCENITH: A Flow Cytometry-Based Method to Functionally Profile Energy Metabolism with Single-Cell Resolution, *Cell Metabolism*, Volume 32, Issue 6, 2020, Pages 1063-1075.e7, ISSN 1550-4131, <https://doi.org/10.1016/j.cmet.2020.11.007>

6. In Fig. 4e, why are there regions with high stress scores in the lean samples? Are there common characteristics of those regions, like distance from vasculature? How do the authors explain the abundance of the stress signature in the lean compared to the weight loss samples?

We have carried out new analyses to investigate this.

In the single nucleus dataset, we find that stressed cells systematically increase in lean tissue with age (**Responses Fig. 2c**). This is not unexpected. Previously we showed that the stressed cell states encompass senescent cells (**Extended Data Fig. 10e**), and age is a major driver of adipose tissue senescence. More interestingly, we find that stressed cells accumulate in lean adipose in association with metabolic dysfunction, independent of BMI and age (**Responses Fig. 2e**). This has not been shown before in humans, to our knowledge. It raises the possibility that senescence is an important driver of metabolically unhealthy leanness.

In the spatial dataset, we identify several characteristics of stressed regions. Cell neighbourhood analyses reveal that stressed adipocytes and precursors colocalise (and may therefore crosstalk) with LAMs and other immune cells in 'stressed' niches (**Responses Fig. 1d**). By contrast, stressed capillary EC2 enrich in the adipocyte niche (**Responses Fig. 1d**) but make up a large fraction of all tissue niches (**Extended Data Fig. 8c**). Thus, despite the conserved hypoxia-*HIF1A* signature in stressed cells, distance from micro-vasculature is unlikely to be a major stress driver. Interestingly, regional bin-based analyses demonstrate unexpected association between arterial ECs and high-stress zonations. A finding that is conserved between lean, obese and weight loss samples (**Extended Data Fig. 8b**). Correspondingly, the arterial niche has communication cues that overlap those of the stressed niche, and is enriched for senescence and SASP marker genes (**Responses Fig. 1e/2b**). Viewed together, this suggests localised crosstalk between stressed, senescent and inflammatory cells within these niches may have a central role in maladaptive tissue remodelling, something we discuss in our revised manuscript, **Main Fig. 5** and **Extended Data Fig. 8**.

We infer the abundance of stress signatures in lean tissues reflects senescence vulnerability associated with age and metabolic dysfunction. While the potent reduction in stress signatures is likely to represent

a weight loss- (or chronic caloric restriction-) specific effect on the self-reinforcing senescence degenerative cycle.

7. The plots where the authors compare obese with lean or weight loss (i.e., Figs 2d, 3b, and 3c) are confusingly labeled, partially because there is no color key on the figure and partially because it is a little counterintuitive to have your upregulated points be the condition not labeled on the plot (i.e., the orange upregulated points on the “Lean” plot are in fact upregulated in the obese condition).

We have changed the plot text and included a colour key to improve clarity.

Minor concerns:

8. There is a typo in Figure 2d (“gluoc” instead of “gluco”).

We have changed this.

Referee #2 (Remarks to the Author):

Miranda and colleagues aimed to characterize adipose tissue dynamics in obesity and therapeutic weight loss. For this, they generated a single-nucleus transcriptomic atlas complemented with a spatial transcriptomics map of human subcutaneous adipose tissue from individuals with extreme obesity undergoing therapeutic weight loss surgery, compared to healthy lean counterparts. They identified gene regulatory mechanisms and tissue signals that drive a cycle of senescence, tissue injury, and metabolic dysfunction. Weight loss was found to reduce adipocyte hypertrophy and biomechanical constraint pathways, enhancing metabolic flux and bioenergetic cycles, which correlated with systemic metabolic health improvements. Although weight loss decreased obesity-induced macrophage infiltration, it did not fully reverse their activation, leaving these cells primed for potential weight regain and metabolic dysfunction. Overall, the study provides new insights into the molecular basis of adipose tissue dysfunction in obesity and its reversal through weight loss.

These insights are derived from a comprehensive analysis of the generated single-nucleus (sn)RNA sequencing data using an extensive cohort. Integrated with data from Emont and colleagues, it serves as a valuable research resource, leveraging a powerful strategy in using samples from the same patients pre and post-surgery. However, and despite the undeniable value of the generated transcriptomic datasets, the manuscript is primarily of descriptive nature and lacks functional validation of its findings, while posing also conceptual issues, as detailed below. In addition, some novel and interesting pieces of information, such as the spatial transcriptomics datasets, are not thoroughly explored and could have contributed much more value to this work, as the transcriptomic characterization of cellular heterogeneity in subcutaneous adipose tissue has already been addressed in several studies (Miao et al., 2020; Whytock et al., 2022; Emont et al., 2022, Ferrero et al., 2024).

We thank the reviewer for highlighting the novelty and value of our work. We have addressed the conceptual questions, spatial transcriptomic usage, and need for further validation below and in our revised manuscript. We believe our in-depth characterisation of adipose tissue cellular heterogeneity in the contexts of human obesity and weight loss substantially advances the existing literature.

Our major comments can be found below:

1. Conceptually, it is challenging to understand why a lean condition is not necessarily the healthiest compared to obese patients who have lost weight but still have a high BMI (around 35). Indeed, despite the intervention, the post-surgery patients remain at a really high BMI range, yet they show less stress, at the gene expression level, than lean patients. This is intriguing, but also troubling, as it in a way suggests that it would be healthier longer-term to recover from extreme obesity compared to staying lean, which cannot be the message the authors want to communicate. How can these observations be explained from a molecular or physiological perspective? Could more validations beyond gene expression be provided? For instance, cell populations could be FACS-sorted to measure metabolic properties or stress response markers at the protein level. If blood samples are available, various parameters from blood samples could be used to analyze adipose-related secreted molecules testifying for stress or for a “healthier” profile.

This is a central question which we have addressed with new analyses and better explanation (also detailed in **Section 1.2**).

Previously we showed that:

- i. the observed stressed cell phenotypes reflect senescence vulnerability;
- ii. tissue/cell stress and senescence levels are highest in obesity but evident in leanness, something others have shown;
- iii. this stress/senescence phenotype is significantly mitigated by weight loss.

We then validated our findings across obese, lean and weight loss groups at protein level, using p21 to reflect senescence and NAMPT the SASP (**Main Fig. 6d/6h**). We have now further confirmed this pattern by measuring the stress marker proteins STAT3 and JUN across conditions (**Responses Fig.**

2c). We used a histological approach because: i. FACS would have required new sample collection after equivalent (>6 months) weight loss which was not feasible within the manuscript timelines; and ii. the majority of stress proteins (>99% in the human cell atlas) are not specific to adipose tissues making their levels in blood uninterpretable. Together these reproducible transcriptomic and protein findings indicate that stress/senescence is potentially mitigated by weight loss. We predict this is important for the metabolic benefits of weight loss because of the size and extent of the effect.

In new analyses, we have explored and identified potential causes of stress/senescence vulnerability in lean tissue (detailed in **Section 1.2**). From a physiological perspective, the differences in metabolic health between leanness and weight loss are likely to result from the composite effects of: i. other cell types/states in adipose tissue; and ii. other metabolic tissues. Our results provide direct evidence of this in the immune compartment where weight loss does not reverse obesity induced immune cell phenotypes. As do prior studies in mice (Cottam et al., Caslin et al. Zou et al.).

In summary, our results establish that weight loss has potent effects on adipose tissue stress/senescence vulnerability, and may thus improve metabolic health in this group. They uncover reasons for stress/senescence in lean tissue, and highlight the opportunity to modify this pathway not just in obesity but also in ageing and unhealthy leanness. We have incorporated these clinically-relevant new findings in our revised manuscript. While making it clear that being/staying lean is healthiest in the long-term.

Cottam, M. A. et al. *Multimomics reveals persistence of obesity-associated immune cell phenotypes in adipose tissue during weight loss and weight regain in mice.* *Nat. Commun.* 13, 2950 (2022). <https://doi.org/10.1038/s41467-022-30646-4>

Caslin, H. L. et al. *Weight cycling induces innate immune memory in adipose tissue macrophages.* *Front. Immunol.* 13, 984859 (2023). <https://doi.org/10.3389/fimmu.2022.984859>

Zou, J. et al. *CD4+ T cells memorize obesity and promote weight regain.* *Cell. Mol. Immunol.* 15, 630–639 (2018). <https://doi.org/10.1038/cmi.2017.36>

2. It is unclear and unexplained what would be the drivers of the reported stress, as it seems systemic: “Moreover, stressed cell states did not form discrete niches, suggesting the micro-environmental drivers of these maladaptive changes may be extracellular or pervasive, and that selective vulnerability to stress may be cell intrinsic.” Identifying these drivers would significantly increase the study’s novelty, as these may constitute interesting targets for downstream therapeutic strategies.

This is also a crucial question which we evaluated in part in our first submission, showing that:

- i. stress markers in mature adipocytes associate with cell size as well as the expression of mechano-sensitive genes;
- ii. stressed stromal cell states (adipocytes, precursors, endothelial, mural cells) are enriched for senescent cells, suggesting this state reflects senescence vulnerability;
- iii. stressed/senescent cells share highly-conserved gene regulatory networks that are likely to be self-reinforcing;
- iv. stressed/senescent cells have strongly-conserved SASP profiles that are also likely to be self-reinforcing;
- v. various presumptive SASP factors overlap the tissue-wide intercellular communication cues that alter reciprocally in obesity and weight loss.

Viewed together this suggests that tissue stress and senescence are interlinked and self-propagating, and identifies adipocyte size and biomechanical stress as a potential driver. In our revised manuscript, we have made the link between these results clearer.

In addition, we have carried out more in-depth analysis into (a) the genes and pathways and (b) the micro-environmental cues that may initiate and/or exacerbate tissue stress.

(a) putative driver genes and pathways:

Re-evaluation of differentially expressed marker genes reveals that stressed cell states constitutively overexpress hallmark members of major stress induction and response pathways (**Responses Fig. 8a** and revised **Main Fig. 4f**):

- i. mechanosensing and -signalling (*TPM1/3/4*, *YAP1*, *ROCK1/2*);
- ii. tissue hypoxia (*HIF1A*, *TIPARP*);
- iii. ROS production and oxidative stress (*NRF2-NFE2L2*, *PRDX6*, *FOXO3*, *SOD2*);
- iv. the DNA damage response and cell cycle arrest (*TP53BP1*, *CDKN1A*, *CCN2*);
- v. endoplasmic reticulum stress and the unfolded protein response (*ATF3/4/6*);
- vi. the Aryl hydrocarbon receptor complex (*AHR*, *HSP90AA1*)
- vii. Gp130 cytokine family-mediated signalling (*IL6ST*, *OSMR*, *STAT3*).

Unbiased pathway analyses of conserved stress marker genes (in ≥ 3 stressed cell states) verify the results, and identify other putative stress pathways (**Responses Fig. 8b** and revised **Extended Data Fig. 7e**). Taking one representative pathway as an example, our new experimental studies show that DNA damage induces stress and impairs differentiation competence in adipocyte precursors in vitro (described in **Section 1.3** and below in **Response to Reviewer 2 Point 3**). Thus, stressed cell states exhibit several conserved pathophysiological changes that may impair their basal functions.

(b) inferred microenvironmental niches and cues:

Improved spatial analyses enable us to define high stress zonations and cellular niches (as detailed in **Section 1.1**). This means we can predict signals derived from these regions, infer source and target cells, and then refine these to signals that alter reciprocally in obesity and weight loss. Using this approach, we show that high stress tissue niches enrich for proinflammatory chemo-cytokines and presumptive SASP cues, several of which have significant trait associations (**Main Fig. 5**, **Extended Data Fig. 8**). Prominent examples of stress signals with significant trait associations are *AREG*, *THBS1*, and *NAMPT*.

Viewed together, this data supports multiple interconnected and mutually reinforcing inter- and intra-cellular processes underlying multicellular stress, senescence and tissue injury. Establishing which are the 'causal' drivers will necessitate high-throughput screening in an in vivo model, something that is beyond the scope of this work. Nonetheless, we highlight significant areas for future exploration by us and others.

Responses Fig. 8: **a.** Volcano plot of differentially expressed genes in basal compared to stressed stromal cells (Adipocytes, APC, Endothelia, Pericytes). Labeled with established stress induction and response genes. **b.** Pathway analyses of genes overexpressed in ≥ 3 stressed relative to basal cell states, restricted to putative stress mediators.

As a sidenote, in Figure 4, how was the threshold defined to determine whether a cell is classified as “stressed” or not? The authors should also better interpret and explain the lean/obese results presented in this figure.

The stressed and basal cell states presented in **Fig. 4** are unbiased clusters within each stromal cell type. For example, in mature adipocytes this is AD1 (basal) and AD3 (stress). We have added the numeric identifiers to **Fig. 4** and biological identifiers to **Extended Data Fig. 5** to make this clearer.

3. Although the bioinformatic analyses are extensive and of seemingly high quality, the study often lacks clear hypotheses or justifications for these analyses or any functional validation of the main claims. For example, the authors claim “we propose that sustained saturated FA exposure in obese AT may activate TLR2-TREM1-mediated LAM polarization from a lipid-buffering to a pro-inflammatory phenotype.” but this stays at the hypothetical level and no functional data is provided to support that. In a similar vein, statements like “We infer this leads to increased metabolic flexibility in mature adipocytes, improved differentiation capacity in precursors and recovery of vascular abnormalities” cannot be made in the absence of a true experimental paradigm. Could the authors at least try to support some of these claims experimentally (metabolic flexibility, improved differentiation capacity, etc.)?

In addition to new protein validation (**Section 1.3**), we now provide experimental evidence to support two main study findings.

i. “We infer this leads to increased metabolic flexibility in mature adipocytes, improved differentiation capacity in precursors and recovery of vascular abnormalities” cannot be made in the absence of a true experimental paradigm.

We have tested and shown experimentally that induction of stress in stromal cells impacts their phenotypes and functions. Taking DNA damage as an example trigger (identified above, **Reviewer 2 Point 2**) we use Etoposide to induce cellular stress in human adipocyte precursors in vitro. We then establish that stress induction, confirmed using the stress marker proteins Jun and STAT3, impairs the differentiation of precursors into mature adipocytes (Oil RedO to quantify lipid accumulation, **Responses Fig. 3c, Extended Data Fig. 7f/g/h**). This experimental data provides support for our previous transcriptomic inferences.

ii. “We propose that sustained saturated FA exposure in obese AT may activate TLR2-TREM1-mediated LAM polarization from a lipid-buffering to a pro-inflammatory phenotype.”

This statement was based on the extensive work already done on saturated fatty acids (SFA) and TLR2-mediated inflammation in macrophages in vitro and in vivo (Jialal et al.). We see little value in repeating these elegant experiments. As an alternative, we tried to isolate macrophages from lean human adipose tissue for SFA stimulation studies. Regrettably, we found this was unfeasible due to small lean tissue biopsies, significant cell requirements, and limited number of myeloid cells in lean tissue. We have therefore removed this statement.

Instead, we used this opportunity to verify our transcriptome-based metabolic flux predictions in adipose tissue macrophages (**Section 1.3** and **Responses Fig. 7**).

Jialal et al. Toll-like Receptor Status in Obesity and Metabolic Syndrome: A Translational Perspective, The Journal of Clinical Endocrinology & Metabolism, Volume 99, Issue 1, 1 January 2014, Pages 39–48, <https://doi.org/10.1210/jc.2013-3092>

4. It is regrettable that the spatial transcriptomics data are not utilized more extensively. We expected more from this dataset than a few superficial validations, as it feels underused and could also have enhanced the novelty of the paper. For example, by examining if the different cellular states of each population (adipocytes, ASCs, etc.) are spatially arranged. This, in turn, could help support some of the intercellular communication cues proposed later in the manuscript.

We have comprehensively reanalysed our spatial datasets to enhance their utility and the novelty of our revised manuscript. This is described in detail above in **Section 1.1**. It enables us to: i. define previously unrecognised spatial arrangements among adipose tissue cell types; and ii. deliver valuable insights into inter-niche and inter-cellular communication mechanisms that change in obesity and subsequent weight loss. We have devoted an entire new results section, **Main Fig. 5** and **Extended Data Fig. 8** to reflect this substantial improvement.

5. Following the above point, there are some notable disconnections between the snRNA-seq data and the spatial transcriptomic datasets. For example, the authors claim stress based on c-Jun expression. Yet, c-Jun is expressed at very high levels in lean patients, while low in the AD4 cluster on the snRNA-seq dataset. In contrast, c-Jun is expressed much lower in the lean patients than in the AD4 cluster in the spatial transcriptomic dataset. Other markers also suffer from this sort of contradictory outcome. How can these results be reconciled? Which one should be trusted in order to build a conclusion for example linked to cellular stress?

We regret some of the previous disconnects were caused by ambiguous labelling of our spatial cell populations. This was because we numbered adipocyte populations according to their size in the respective single nucleus and spatial datasets. Thus, AD3 was 'stressed' and AD4 was 'intermediate' in the single nucleus dataset, and AD4 was 'stressed' in the spatial dataset.

In our revised manuscript, we have changed the identifiers in the spatial dataset to reflect the single nucleus dataset. We have also made two changes to our spatial analyses that improve the concordance between datasets in adipocytes:

i. For size-based analyses, we have used a new segmentation approach that assigns ambiguous transcripts on the boundary between ≥ 2 adipocytes to both cells. Where previously we assigned the transcript to one neighbouring cell. This does not markedly change the outcomes of size-based analyses but is, at least, conceptually an improvement.

ii. For non-size-based analyses, including marker gene expression in adipocyte subtypes, we have now used the 'cleaner' nucleus segmentation (which we have reannotated as detailed in **Section 1.1**) rather than the boundary segmentation to minimise ambiguities arising from the contaminating effects of mis-assigned transcripts.

We have termed these 'boundary' and 'nucleus' segmentation in the manuscript for greater clarity. The changes significantly improve the concordance of the expression results presented in **Main Fig. 3g**, also shown in **Responses Fig. 4**, strengthening the existing manuscript conclusions.

6. Some interpretations are overly speculative. For example, while predicting metabolic fluxes from transcriptomic data is interesting, the limitations of this approach and need for functional validation should be clearly acknowledged. The term "phenotype" is frequently used throughout the manuscript, despite not referring to an actual biological phenotype. Similarly, the terms "functional" and "adipocyte activation" are ambiguous and their meanings are unclear in the context of adipose biology, using nomenclature of the field could be more useful. As such, we feel that the large levels of speculation lead to statements that feel too bold and actually weaken the manuscript, as also already indicated above.

We have carried out significant additional ex vivo and experimental validation (described in **Section 1.3**), reduced speculation where validation is not provided, and addressed the issues of terminology. As examples: we have changed 'phenotype' to 'transcriptomic phenotype' or 'profile', 'activation' to 'metabolic gene activation' or 'flux' and removed functional where it is not appropriate; we have also made it clear that all metabolic flux analyses leverage indirect transcriptomic rather than direct metabolite information.

7. It is unclear whether the dataset that is presented and analyzed throughout the manuscript is solely the one generated by the authors or if it is systematically combined with the dataset generated by Emont and colleagues. The authors should clarify this distinction.

We integrated our single nucleus data with the Emont dataset to: i. provide a more comprehensive reference atlas of human whole subcutaneous adipose tissue cell types and their compositional frequency; and ii. specifically improve annotation of rarer cell types/states.

For exploratory analyses between conditions, we used our own dataset for the following reasons. First, our cohort comprised people at the population extremes (morbid obesity undergoing weight loss and lean counterparts) rather than people from across the BMI spectrum. Second, our obese and weight loss samples were paired, while our lean controls were selected to match for age, sex and ethnicity. Third, we excluded individuals with comorbidities or medications that might impact adipose biology.

Fourth, we processed our samples in experimental trios (lean-obese-weight loss) to minimise technical and sequencing batch effects (which are common in adipose).

For differential cell abundance and gene expression analyses (that permit adjustment for technical and biological covariates), we illustrate results from our dataset alone (reported) and the combined dataset to evaluate reproducibility in **Responses Fig. 9**. This shows high levels of concordance between results derived from the respective analyses. We have clarified this important distinction in the main text and methods of the revised manuscript.

Responses Fig. 9: **a.** Differential expression effects (log₂ fold change, logFC) between lean and obese groups in major cell types. X axis: reported (our dataset alone). Y axis: reported and Emont datasets combined. **b.** Neighbourhood abundance effects (logFC) in reported (Miranda-only) and combined (Miranda reported and Emont) datasets for all cell types (top) and myeloid cluster cell states (bottom).

8. To address the ambiguity of the above point (and in fact judge the quality of the presented analyses in general), it would be very useful to make the code accessible. This would help clarify some of the analysis parameters, such as integration. It appears that the integration between the two datasets was either remarkably seamless or possibly overintegrated, which could mask meaningful biological differences. For instance, could the authors provide the UMAP from their own dataset without integrating it with the dataset from Emont and colleagues?

We have made our code accessible as requested at:

https://github.com/WRScottImperial/WAT_single_cell_analysis_Nature_2024

For integration, we evaluated multiple approaches (Seurat CCA/RPCA, MNN, scVI, Scanorama, Harmony, BBKNN, BBKNN+Harmony). We then selected the method leading to the closest integration of the datasets and phenotype groups. This was BBKNN+Harmony. We did this to minimise 'false positive' signal driven by unwanted variation (sampling, biological, or technical) between groups. In our experience, adipose tissues are particularly prone to technical artefacts. BBKNN and Harmony are widely used for integration, and perform well in benchmarking studies (Luecken et al.). We provide the requested UMAPs which shows the stability of the identified cell populations in our dataset alone and the Emont dataset alone (**Responses Fig. 10**).

Even with close integration, we discover extensive differences in cell abundance, regulatory networks, genes, and signalling pathways between groups; findings we systematically replicate in a limited

independent spatial dataset. We appreciate this could mask meaningful differences beyond those identified. However, less tight integration would increase the amount of false discovery, something we prioritised avoiding.

We are making our datasets available in several formats to maximise their usability. This will enable researchers to re-integrate them, alone or with existing/new datasets, with the aim of uncovering additional changes linked to obesity and weight loss, or other clinically relevant traits.

Responses Fig. 10: UMAP plots of major adipose tissue cell types in: i. our data without integrating it with the Emont data (left); and ii. the Emont data without integrating it with our data.

Luecken et al. *Benchmarking atlas-level data integration in single-cell genomics. Nat Methods* 19, 41–50 (2022). <https://doi.org/10.1038/s41592-021-01336-8>

9. The manuscript frequently mentions the aspect of "therapeutic development", but the authors should be more cautious suggesting any therapeutic applications based on their results. The findings are highly descriptive, and thus the potential for actual clinical application may be limited.

We have moderated and reduced references to therapeutic development in our revised manuscript.

Minor comments:

1. In Figure 1, it would have been interesting and expected to see some correlations on the patient characteristics with cell type abundance. Additionally, abundance plots could have been included for other cell types as well.

We have included correlations with patient characteristics in **Main Fig. 1**, and adipocytes cell subtype proportions separated by individual in **Extended Data Fig. 5b**.

2. In the text, the authors claim: "and relative increases in other prevalent adipocyte subpopulations including lipogenic AD5 cells." There are less calories / lipids, so why make more lipids? How can this be explained?

We have renamed these cells to 'lipid biosynthetic' to better reflect the likely presence of lipid cycling in these cells.

3. The clustering is based on only a few genes, this might not be enough. Moreover, some are stressed (c-Jun) and some are fibrotic (SLC14A2) markers, would we not expect high c-Jun expression in the fibrotic cell population as well?

The single nucleus clustering is based on many genes. For space purposes we illustrate a subset. The others are made available in the Supplementary files.

JUN has crucial and diverse roles in adipocytes. As such it is expressed in all populations but highest in the stress cluster. This is in keeping with its known effects on driving the senescence transcriptional program. JUN is also marginally higher in fibrotic than basal, GRIA4-hi and intermediate adipocytes in the single nucleus and spatial datasets, in keeping with expectation.

We named the SLC14A2 cluster fibrotic because it enriched for fibrotic and collagen ECM genes (e.g. NOX4, LOX, LOXL, VGLL3) and the TGFB signaling pathway.

4. In general, figure legends could be improved for clarity within the figure panels. For example, in Figure 3, the color labeling is difficult to understand, requiring frequent reference to the figure legend text. This would greatly improve the reading flow.

We have systematically amended our figure panels to improve the clarity and help reading flow.

5. The main message of Figure 5 is not clear and requires further clarification.

We have completely remade **Main Fig. 5** and **Extended Data Fig. 8** to present our extensive new spatial niche analyses.

6. The contrast in Figure 6G could be increased for better visibility.

We have changed the colours to improve the visibility in this figure.

7. The term “likelihood” used in Figure 8D has a very specific statistical meaning and appears inaccurate in this context.

We agree and have changed this term.

8. In Figure S2D, the y-axis label seems wrong as the axis is already scaled.

We have changed this to numerical rather than log labels.

9. In Figure S4A, where is c-Jun expression? How stable is this grouping?

We show UMAPs with expression of c-Jun and other stress as well as generic marker genes (**Responses Fig. 11, Extended Data Fig. 7b**). This expression signature is strong making the cluster very stable across different Leiden settings.

Responses Fig. 11: UMAPs of population clusters, generic (ADIPOQ, PLIN4) and stress marker gene (JUN, FOSB, ATF3, EGR1, ZFP36) expression levels (log normalised) in mature adipocytes.

Referee #3 (Remarks to the Author):

Summary:

While this paper is well written and the data is of great interest as a resource, it is not the first single-cell atlas of adipose tissue. However, inclusion of paired weight-loss samples improves the novelty and usability of the data. Additionally, it may be the first to include spatial analysis of human adipose in lean and obese individuals. Ultimately, despite utility as a reference and very intriguing findings, the results are largely correlative and have not been extensively validated at the protein level. It remains unclear which components (SASP, global metabolic changes, etc) are drivers of metabolic disease in obesity or recovery in weight loss and if those factors are manipulatable as treatment options. Regardless, the authors highlight significant areas that have not yet been explored in human adipose tissue biology and that should be the basis for continued future work by them and others. One strength is that the Obese/weight loss patients are matched samples (weight loss are post-surgical samples. However, the average BMI of weight loss group is still in the morbidly obese classification (extended table 1). This raises some concern about their comparator for the lean group. The quality of the figures is excellent and the descriptions are also very high quality. I appreciated that many “hits” are displayed in figures, but the focus is centered on a subset of those hits with explanations for their relevance in the disease setting.

We thank the reviewer for highlighting the value and expected utility of our work. We agree its main impact will be as a resource (and guide) for ongoing mechanistic and therapeutic exploration. We have therefore undertaken additional ex vivo and experimental validation to cross-verify key findings.

We think our lean comparator is useful (if not perfect) because it allows us to define obesity-induced effects that are not mitigated by weight loss (e.g. persistent LAM activation), and weight loss effects that are not driven by BMI per se (e.g. the potent effects of weight loss on senescence and adipocyte bioenergetics).

Major:

1. The authors identified LAM subsets in single cell data (adaptive and inflammatory), but did not evaluate these subpopulations similarly in the spatial datasets. Colocalization of adipocytes/LAMs would be extremely valuable knowledge that does not currently exist and would benefit an atlas. Do the adaptive LAMs appear near early CLS, but the inflammatory near late-stage CLS? Colocalization (figure 5a) appears to be low for all cell types, but it is known that adipocytes are densely packed and frequently surrounded by immune cells. Is this an artifact of cell segmentation of the Xenium data? Perhaps the bin size should be increased to better show colocalization of large cells such as adipocytes?

We thank the reviewer for the helpful suggestions.

We have developed several new methods to interrogate cell colocalisation within tissue niches (detailed in **Section 1.1**). This means we can begin to address the crucial questions raised about LAMs. However, the apparent complexity (Geng et al.) and low frequency of CLS in tissue (~3.9% in obese mice (also Geng et al.) where CLS are much more prevalent than in humans) mean a more complete understanding is not possible without: i. significantly more Xenium sections and biological replicates; ii. clearer appreciation of CLS characteristics. This is not feasible within the manuscript timelines.

What we can (and do) now show is relevant and valuable.

First, we find that LAMs colocalise in niches with stressed adipocytes, stressed precursors, and other immune cells, highlighting the potential for crosstalk between these cell types in maladaptive tissue remodelling (**Response Fig. 1d**). This enables us to predict communication patterns arising from cells of the stressed niche, including inflammatory chemo-cytokines and stress/SASP cues, and link these to obesity and weight loss (detailed above in **Section 1.1, Main Fig. 5, Extended Data Fig. 8**).

Second, we find that LAMs aggregate with other LAMs at CLS, replicating known findings and verifying our colocalisation methods (**Response Fig. 12a/b**). Xenium resolution elegantly demonstrates that

LAMs surround transcriptionally devoid/dying adipocytes, which means we (and others) cannot use the transcriptome to infer direct communications (**Response Fig. 12b**).

Third, we use unbiased label transfer to recover LAM subtypes in our spatial dataset, and cross-verify the results using marker genes (**Response Fig. 12c**). This enables us to show in a limited independent patient cohort that ‘inflammatory’ LAMs are generally higher in obesity and significantly reduce with weight loss, relative to ‘adaptive’ LAMs (**Response Fig. 12d**). Equally, we discover that adaptive LAMs aggregate in groups (N \geq 3) at CLS (**Responses Fig. 12b/d**). Whereas inflammatory LAMs form a greater proportion of LAM singletons or LAM pairs (N $<$ 3, **Responses Fig. 12d**). We confirm these patterns in situ using TREM2 and TLR2 protein markers at histologically verified CLS (**Responses Fig. 3a**). In the absence of a time-course, it is unclear whether the latter represent early-, late- or non-CLS, at which inflammatory LAMs might have different impacts, for example immune recruitment vs. harmful remnant.

In summary, we provide important new insights into LAM biology, and the impetus for us and others to undertake much needed follow-up studies.

Responses Fig. 12: **A.** Cell-cell colocalisation by distance in the spatial datasets showing LAM-LAM (and LAM-other immune cell) enrichment. **B.** LAM aggregation at two representative CLS (top); coloured by LAM subtypes. LAM- (PLA2G7) and adipocyte- (ADIPOQ, PLIN4) specific transcripts mapped to the same CLS (bottom); adipocytes surrounded by LAMs were transcriptionally devoid. **C.** LAM subtype markers in the spatial dataset (identified by label transfer and cross-verified with marker genes). **D.** The proportion of LAM ST1 and ST2: i. in groups (≥ 3) or singletons/pairs (≤ 2 , top); ii. in each condition; Chi-Square test of independence; LAM ST3/4 frequencies were too low to evaluate.

Geng et al. 3D microscopy and deep learning reveal the heterogeneity of crown-like structure microenvironments in intact adipose tissue. *Sci. Adv.* 7 (2021). <https://doi.org/10.1126/sciadv.abe2480>

2. Senescence/SASP was identified as a key change (reduction) in the obese to weight loss transition. However, SASP, including p21, was similar between lean and obese individuals (i.e. does not appear to be an obesity-driven phenomenon, but rather a WL impaired phenomenon). The Discussion should further evaluate this finding and offer potential explanations (or limitations) as to why this may be. For example, the authors describe enhanced bioenergetics

(e.g. lipid cycling) in weight loss that may not be also enhanced in lean individuals compared to obese; is this directly associated with reductions in SASP?

We agree the near complete amelioration of senescence is a weight loss (or chronic calorie restriction) induced phenomenon. We also note our results show that senescence (p21 expression and scores) and SASP components (e.g. NAMPT) are consistently higher in obese than lean adipose (**Main Fig. 6 and Extended Data Fig. 10**). We do not wish to marginalise this clinically important finding, which is consistent with the existing evidence base (Smith et al.).

Irrespective, we have carried out new analysis to investigate potential reasons for senescence in lean tissue, and senescence reversal by weight loss. This shows that: i. age is a likely driver of senescence in lean tissues (**Response Fig. 2d**); ii. senescence in lean tissues increases with worsening metabolic function independent of age (**Response Fig. 2e**); iii. senescence reduction after weight loss is directly associated with (and may thus be mechanistically coupled to) enhanced bioenergetics in adipocytes (**Response Fig. 13a**). This data enhances our previous findings, offering explanations for senescence patterns in lean and weight loss tissues. More broadly, it highlights the potential to target senescence, and related bioenergetic pathways, to improve metabolic health in diverse contexts: obesity, age-related metabolic dysfunction, and lean people stratified as metabolically unhealthy.

We provide these results in **Extended Data Fig. 7d and 10g** together with enhanced discussion in the text.

Smith et al. Cellular senescence and its role in white adipose tissue. *Int J Obes* 45, 934–943 (2021). <https://doi.org/10.1038/s41366-021-00757-x>

Finally, NAMPT decreases with aging and has been shown to increase with exercise; age range of the study participants is 20-70 years old. Thus, difference in the image in 6h could be due to age rather than obesity vs weight loss.

We observe no relationship between NAMPT expression and age in our samples (**Response Fig. 13b**). We conclude the observed effects on NAMPT are age independent.

Responses Fig. 13: **a.** Correlation of delta changes in pathway scores before and after weight loss (in paired samples); calculated using the mean pathway score across adipocyte cells for each sample. **b.** The relationship between age and NAMPT expression in all study participants, coloured by trait. NAMPT level represents the average expression (log normalised and scaled) across all cells for each participant in the single nucleus dataset.

Minor:

- **The authors suggest inflammation is reduced in weight loss, but see little evidence of inflammatory macrophages in all groups. It would be beneficial to include speculation in the Discussion.**

As the reviewer states, and consistent with recent studies, we do not observe classic (M1) inflammatory macrophages in our results. We do however observe alterations in several important inflammatory genes and pathways (e.g. the NLRP3 inflammasome) in obese macrophages that improve with weight loss. We have made this distinction in the revised manuscript discussion.

- **The y-axis on extended data figure 2d may be incorrect (labeled as -log, but also presented on a log scale).**

We presented this data on a nonlinear scale to enable visualisation of significant differences in less abundant genes (e.g. TREM1/2) which otherwise may be masked. We have changed the axis labels to unlogged numbers.

- **It would be helpful if the text referred to LAM1 and LAM2 (adaptive and inflammatory) as they are presented in the figures (LAM ST1 and LAM ST2)**

We have changed this to be consistent.

- **Bar graphs (e.g figure 2C) would benefit from statistics. If these are not significant, the results may be overstated (trend vs. statistically significant finding)**

We have added significance statistics to cell composition bar graphs.

- **The section labeled “Weight loss reduces innate immune infiltration” appears to be mistitled, as it refers to specifically adaptive populations of cells.**

We have corrected this error.

- **Adipocyte area presented as a log-scale makes it hard to recognize how large of difference in adipocyte size there is in Lean -> obese -> weight loss**

We now provide the logged and unlogged areas.

- **Need clarification on whether Figure 4e is scaled expression (across all samples or per group?) vs. normalized expression.**

o **Unclear if expression is overall lower (normalized) or if there is less variance in expression (scaled).**

o **Would also be valuable to indicate the 24 genes that were used for scoring of spatial data in the extended data file (perhaps by bold or a second sheet; Supp File 12).**

This figure is not scaled. It is the logged score of the 24 conserved stress genes present in the spatial dataset within the tissue bin (50- μ m). We make this clearer in the legend. We have also added the 24 genes to Supplementary Table 12.

- **The gene AREG was noted in the methods to be added as part of the secreted SASP markers. A brief mention as to why this was added in the Results would be an improvement over mention in the Discussion.**

We added AREG to the SASP markers for accuracy because it is known to be secreted but is not currently annotated as such in the Human Protein Atlas. This was done to help readers connect the overlapping cell interaction (AREG-EGFR) and SASP results which is important for interpretability. We make this rationale clear in the figure legend of the affiliated results section, and keep the existing justification in the methods.

- The Discussion would be improved by the inclusion of a limitations section, which could elaborate on many of the following:
 - o Before weight loss, samples were collected by surgical incision, but post-operative samples were collected by needle biopsy.
 - o Discussion of non-surgical weight loss should be included. This is particularly important given the comparable weight loss induced by the new drugs on the market.
 - o All patients are not classified as T2D despite elevated insulin and HOMA-IR. This should be mentioned in the discussion/limitations.
 - o Discussion of potential differences between subcutaneous and visceral adiposity should be mentioned.
 - o Low overall numbers of some cell types limit interpretation

We have now included a limitations section in the discussion covering these points.

- It is excellent that authors are providing processed transcriptomics data in two formats as a resource; however, the Github page is not currently accessible.

We will make the data publicly accessible at publication.

NATURE_2024-05-10393, RESPONSE TO REVIEWERS 2.

Referee #1 (Remarks to the Author):

The authors have made significant efforts to respond to my concerns. The work is technically sound, and they have done more to leverage the spatial dataset. They have also performed more experiments to validate some of the predictions raised by the single cell data. I have no further substantive concerns.

We thank the reviewer.

Referee #2 (Remarks to the Author):

The revised version of Miranda et al. shows some significant improvements and the authors must be commended for their efforts. They make better use of their spatial transcriptomic data, and new hypotheses have been included as to why adipose tissue from lean people showed higher stress than that of obese individuals undergoing body weight loss interventions. Nevertheless, we think that some main aspects remain vague and would need to be further addressed:

- Although we acknowledge the attempt of the authors to better use their spatial transcriptomic results, this exciting dataset could still have been used more extensively. The different conditions (lean vs. weight loss) could have been exploited more.

We agree with the referee that this is an exciting dataset. Indeed, we believe this technology has the potential to thoroughly explore adipose tissue (AT) biology in ways that have not yet been done. We have made use of our spatial dataset to validate and contextualise findings in every section of the manuscript. This includes analyses of all 3 conditions. We have also dedicated an entire results section to this dataset, providing (i) novel biological insights and (ii) new ways to analyse the data that overcome potential confounding effects in AT such as the difference in adipocyte size.

We acknowledge a direct comparison of lean and weight loss is of interest. However, in our view the synergistic comparison of pathological AT remodelling in obesity and leanness, and obesity after marked weight loss is more biologically and clinically meaningful. Whilst we would like to do more with this technology in the future – for example we would also like to comprehensively explore the vascular compartment in a spatial context – we are limited by space in the current manuscript. Nonetheless, a major benefit of this (and other) exploratory human single cell and spatial datasets is the potential to continue to test new hypotheses and address important biological/pathobiological questions. We are providing our datasets in multiple practical formats to maximise their utility, enabling others to do this.

- One of the main take-home messages of the paper, namely that overcoming extreme obesity is seemingly more beneficial for health than simply remaining lean, is still very difficult to understand. The authors push forward a hypothesis relying on the senescent state of the tissue. This is an unexpected finding and, given that it is the cornerstone to explain a major counterintuitive phenotype, more experimental validations (e.g.: ex vivo) would be required to solidify it. Indeed, with the current level of provided support, we remain uncomfortable underwriting such a conclusion, as it is in our opinion highly provocative and may reflect latent issues with the lean cohort that is biasing the findings.

Respectfully, we have to disagree with the reviewer. We do not conclude that overcoming extreme obesity is more beneficial for health than simply remaining lean.

To the contrary, we show in **Main Fig. 1b** that although weight loss improves multiple metabolic parameters, lean counterparts have consistently better overall metabolic profiles. We also show that weight loss has apparent beneficial effects in some cellular compartments (e.g. senescence vulnerability, adipocyte metabolic flux models) but does not reverse obesity-induced changes in others (e.g. innate immune cells). This leads us to conclude:

i. leanness is healthiest (line 421);

- ii. weight loss has beneficial effects linked to the observed changes but (a) does not lead to complete metabolic recovery and (b) may promote worsening long-term outcomes in the context of weight regain (lines 432-434).
- iii. proactive prevention of obesity (as opposed to treatment with weight loss) is essential from a public health perspective (line 448).

We have now emphasised the findings of **Main Fig. 1b** in the main text (lines 99-100) as well as the discussion to avoid any potential for misinterpretation.

In this context, we do not view our findings as provocative, or indeed counterintuitive. Short term caloric restriction has been shown in multiple studies to mitigate senescence (Fontana L et al.). Thus, we might expect extended caloric restriction during marked weight loss to have similar yet more profound effects. Moreover, senescence has well-described adverse inflammatory, metabolic and disease consequences in AT, linked to functional changes in AT cell types, that can be ameliorated with senolytic interventions (Spinelli R et al., Zhuohao L et al.). In ASPC, for example, human obesity leads to increased senescence and impaired differentiation capacity ex vivo; an effect that is likely to be direct and indirect through the SASP (Gustafson B et al., Xu M et al. 2015). By contrast, human weight loss improves impaired adipogenic potential, and the secretory profile, of obese human ASPC ex vivo (Rossmeslová L et al.). These effects of senescence and the SASP on ASPC differentiation competence, as well as overall systemic homeostasis, have been experimentally verified in vitro and in vivo with: (a) diverse inducers of senescence (including saturated FA and mitochondrial stress); and (b) reversal using senolytics or SASP inhibitors (Palmer A et al., Xu M et al. 2015 and 2018, Ishaq A et al., Wiley C et al.). Importantly, comparable effects have also been shown in other AT cell compartments (Spinelli R et al., Zhuohao L et al.). We have now made better reference to this crucial supportive literature to verify that our findings not counterintuitive, making clear what we derive from our own work and the previous literature.

While of some merit, showing repeatedly that senescence improves with weight loss relative to other groups using another ex vivo modality (we have already done this with single nucleus and spatial transcriptomic and several distinct protein assays in independent samples) would provide only limited incremental value, would require de novo sample collection, and is thus not possible within the manuscript timelines. Moreover, to precisely quantify the opposing effects of senescence and persistent immune reprogramming on whole-body metabolism in leanness, obesity and weight loss, will require complex experimentation using an in vivo system. This will take many years of work, cannot reasonably be undertaken for a human atlas manuscript, but is a crucial follow-on study.

We appreciate the concern about the potential for latent variability in human patients. However, the variation in our lean cohort is within expectation compared to other non-obese cohorts. This is exemplified in **Fig. 1** below which shows the expression of Senescence/SASP-related genes in: i. AT measured with the same technology (**Fig. 1a/b**); ii. different organs with the same technology (**Fig. 1b**); iii. AT with a separate technology (**Fig.1c**). In our previous responses, we showed that stress and senescence variation in lean tissues is associated with age and fasting insulin levels. This leads us to conclude our lean findings are representative, follow known (age) and expected (fasting insulin) trait associations, and do not reflect latent cohort-intrinsic biases. We agree it will be important to understand the drivers of within group variation in more depth. However, such endeavour is beyond the scope of this manuscript, and does not impact our findings.

In summary, we provide multiple lines of evidence demonstrating the reported senescence findings are robust, representative, and not due to latent confounders. We incorporate more experimental evidence from the literature to uphold the functional consequences of senescence on AT cell types, and demonstrate our findings are not counterintuitive. We modify the text to avoid misinterpretation of our lean and weight loss results from a public health perspective.

Fig. 1: Variation in Senescence/SASP marker expression in AT and other human tissues in independent non-obese samples and cohorts, using the same droplet-based and different technologies.

Fontana L et al. Caloric restriction and cellular senescence, *Mechanisms of Ageing and Development*, Volume 176, 2018, Pages 19–23, ISSN 0047-6374, <https://doi.org/10.1016/j.mad.2018.10.005>

Gustafson B et al. Reduced subcutaneous adipogenesis in human hypertrophic obesity is linked to senescent precursor cells. *Nat. Commun.* 10, 2757 <https://doi.org/10.1038/s41467-019-10688-x>

Ishaq A et al. Palmitate induces DNA damage and senescence in human adipocytes in vitro that can be alleviated by oleic acid but not inorganic nitrate, *Experimental Gerontology*, Volume 163, 2022, 111798, <https://doi.org/10.1016/j.exger.2022.111798>

Palmer A et al. Targeting senescent cells alleviates obesity-induced metabolic dysfunction. *Aging Cell*. 2019; 18:e12950. <https://doi.org/10.1111/ace1.12950>

Rossmeslová L et al. Weight Loss Improves the Adipogenic Capacity of Human Preadipocytes and Modulates Their Secretory Profile. *Diabetes* 1 June 2013; 62 (6): 1990–1995. <https://doi.org/10.2337/db12-0986>

Spinelli R et al. Increased cell senescence in human metabolic disorders. *J Clin Invest*. 2023;133(12):e169922. <https://doi.org/10.1172/JCI169922>

Wiley C et al. Mitochondrial Dysfunction Induces Senescence with a Distinct Secretory Phenotype, *Cell Metabolism*, Volume 23, Issue 2, 2016, Pages 303–314, <https://doi.org/10.1016/j.cmet.2015.11.011>

Xu M et al. Targeting senescent cells enhances adipogenesis and metabolic function in old age. *Elife* 4, e12997 (2015) <https://doi.org/10.7554/eLife.12997>

Xu M et al. Senolytics improve physical function and increase lifespan in old age. *Nat Med* 24, 1246–1256 (2018). <https://doi.org/10.1038/s41591-018-0092-9>

Zhuohao L et al. The role of adipose tissue senescence in obesity- and ageing-related metabolic disorders. *Clin Sci (Lond)* 31 January 2020; 134 (2): 315–330. doi: <https://doi.org/10.1042/CS20190966>

- Some claims that are made in this manuscript might be misleading in terms of the scope of the current work. We previously mentioned a sentence claiming that “we propose that sustained saturated FA exposure in obese AT may activate TLR2-TREM1-mediated LAM polarization from a lipid-buffering to a pro-inflammatory phenotype.” In their rebuttal, the authors confirm that these causal mechanistic links are inferred from the literature, which should be clarified in the main text.

We thank the reviewer for the comment. We have clarified in the text where our inferences are based on existing literature.

In a second claim, the authors stated that “We infer this leads to increased metabolic flexibility in mature adipocytes, improved differentiation capacity in precursors and recovery of vascular abnormalities” which now the authors tried to address experimentally. Yet, the presented experiment

only helps to show that adipocyte precursors treated with etoposide, as a stressor causing DNA damage, fail to differentiate. This poorly relates to the claim made by the authors. Nobody would probably question that substantial DNA damage can have a significant impact on adipocyte differentiation. Could the authors at least show ex vivo that the differentiation capacity of adipocyte precursors from lean and obese patients is reduced compared to those of people after weight loss (and, therefore, less stressed)? If so, and given the main hypothesis that the authors put forward at the moment, they could examine if senolytic agents could improve the differentiation capacity of the stressed groups compared to cells from non-stressed tissues.

It was not our intention for this to be misleading in terms of what is current work and what is already known. We made this inference based on the following cumulative evidence:

- i. distinct single nucleus and spatial transcriptome and protein level evidence from independent samples showing that human weight loss significantly decreases AT senescence relative to other groups;
- ii. transcriptional changes in stressed/senescence vulnerable cells supporting impaired basal metabolic, precursor and vascular cell functions; changes that improve with weight loss;
- iii. in vitro evidence demonstrating that induction of senescence through the DNA damage response pathway impairs adipocyte precursor differentiation competence;
- iv. the extensive senescence literature in AT (summarised above) as well as the broader vascular and stromal cell compartments.

We acknowledge that the readership may not be immediately familiar with point iv. We have therefore revised our text to clarify what is derived from our results and what is known from the existing literature.

We appreciate that ex-vivo experiments and subsequent drug intervention involving cells obtained from patients within these 3 conditions could be valuable to the community. However, we believe this is beyond the scope of the current manuscript and is of limited incremental benefit on top of the multifaceted evidence already presented. First, recruiting new patients, including paired obese and weight loss samples, is not something that can be achieved in a reasonable time frame. Second, the effects of senescence (ex vivo and using diverse triggers targeting different pathways) and senotherapy (with genetic and senolytic targeting) on pre-adipocyte differentiation competence are well established.

What we show is that weight loss improves the senescence state of the tissue, which we have extensively validated. We infer this may have important functional consequences based on our own data and data from others.

- We thank the authors for providing the separated UMAPs from their own dataset and the Emont et al. one. It would have been interesting and valuable to provide the stratification UMAP plots as well, to visualize how the cell clustering is affected by the different conditions. Additionally, can the authors discuss the similarities between the Emont et al. plots with their weight-loss condition? Finally, responses Fig. 10 should be placed in the main figures.

We clustered all conditions together. Clustering each condition separately within datasets would lead to vastly inferior numbers of cells being clustered, low power, and would invariably give slightly different results, especially for rarer cell types. We do not therefore see how showing the stratification UMAPs would provide additional meaningful information.

We have already provided the UMAPs showing the distribution of the different conditions and datasets (**Extended Data Fig. 1a**). We have also provided barcharts of the cell breakdowns for each sample across groups and datasets (**Main Fig. 1d** and **Extended Data Fig. 5b**). The only information that can be gathered from the stratification UMAPs would be which cell types are present in each dataset, something we already show at individual dataset, condition and sample level.

UMAPs are primarily a tool to aid data visualization. They cannot be used to infer relationships between datasets or cell types (The specious art of single-cell genomics | PLOS Computational Biology). For this reason, it is difficult to draw meaningful conclusions when comparing the Emont and our own weight loss UMAPs, or from the data presented in previous responses **Fig. 10**.

Irrespective, every analysis we present, including cell clustering, is agnostic of UMAP dimensionality reduction.

Chari T, Pachter L (2023) The specious art of single-cell genomics. PLOS Computational Biology 19(8): e1011288.
<https://doi.org/10.1371/journal.pcbi.1011288>

Referee #3 (Remarks to the Author):

The authors have extensively revised the manuscript. All of my concerns have been addressed.

We thank the reviewer.

Referee #4 (Remarks to the Author):

The revised manuscript by Miranda et. al. presents a novel atlas of subcutaneous adipose tissue during the settings of obesity and weight loss using single nuclei transcriptomics and spatial gene expression analysis. Using nuclear and cell-boundary segmentation approaches, the authors identify regions of cellular stress occupied by large adipocytes and lipid-associated macrophages. Furthermore, the authors highlight how weight loss modifies signatures of cell stress in paired-specimens. The dataset generated for this study is a novel and valuable resource and the authors have made their code available to improve accessibility of this dataset for future work. The inclusion of paired specimens of obese and weight-loss tissue provides additional value and novelty. Revisions to improve nomenclature/labels and validate findings with external approaches (SCENITH, microscopy) greatly enhance the readability and support the authors conclusions. Finally, inclusion of a new figure evaluating the spatial cellular niches that present in lean, obese, and weight loss subcutaneous adipose tissue is a significant improvement. The authors have also improved their discussion to reduce speculative statements and include limitations.

Minor Comments:

1. In Fig. 2A, three populations are listed as MY1, MY2, and MY3 and should be renamed to MYE1, MYE2, and MYE3 in line with the rest of the figures.

We thank the reviewer for their comments and pointing this out. We have changed the labels.

2. While the limitations section is newly included, it is very brief and could be improved by providing some description as to how these limitations might affect the interpretation of the study. For example, the authors state, "Other limitations include...the abdominal subcutaneous depot and surgical weight loss...". Differences between subcutaneous and visceral are well reported and therefore the authors have a great opportunity to speculate on how visceral may differ in regard to their major findings (cell stress, signaling, spatial niches, et). Furthermore, medical weight loss is a rapidly expanding field and input from the authors on how medical weight loss may result in differences (if any) from surgical weight loss in the context of their findings would be very interesting.

We agree with the reviewer these are important areas of uncertainty. We did not address them previously because we were concerned it might be too speculative, something other reviewers were critical of. We have now included this within our discussion.

Referees' comments:

Referee #1 (Remarks to the Author):

The authors have made clear that they are not claiming that losing weight is more healthy than never having been obese in the first place. I believe further experiments are beyond the scope of the current study.

We thank the referee for their extensive efforts to review our work as well as their supportive comments.

Referee #2 (Remarks to the Author):

We thank the authors for thoughtfully replying to our comments and for better contextualizing the biomedical implications of their findings. In our opinion, further evidence would still be needed to fully demonstrate that adipocyte precursors and vascular cells from lean healthy patients display higher senescence and stress markers than those of class III obese patients after body weight loss – being still remarkably obese. Since the causes and functional implications of this rather controversial finding, at present, remain unknown, we cannot underwrite it. However, we also acknowledge that the study provides a valuable dataset and that the current version of the manuscript constitutes already a rich and dense piece of work. Therefore, we understand that a comprehensive exploration of the above topics could be beyond the scope of a single paper.

We thank the referee for their comprehensive and helpful review of our manuscript. We are grateful for their comments about the value of the work and the related datasets.

We agree it is striking that patients who remain obese (class II-III) but who have (a) already lost a significant fraction of their excess weight and (b) remain in an active weight loss phase, have markedly lower levels of senescence than weight stable lean counterparts. We are confident we have extensively replicated this finding using multiple distinct modalities (single nucleus, spatial, various protein markers) in independent samples. It is important to appreciate the reported weight loss group is profoundly and chronically calorie restricted, and that caloric restriction has a well-established mitigating effect on senescence. We accept the observed senescence reduction may revert to stable BMI equivalent levels in future, once weight loss has plateaued. Nonetheless, we would still expect, based on our lean findings, this to be lower than the pre-weight loss obese baseline. Future longitudinal studies in new patient samples obtained across multiple timepoints will be needed to address this uncertainty. We make it clear in our discussion that we cannot distinguish the respective contributions of negative energy balance, dynamic weight change and absolute fat mass, to the observed tissue and systemic effects.